# Using a nested single-model large ensemble to assess the internal variability of the North Atlantic Oscillation and its climatic implications for Central Europe

Andrea Böhnisch[1], Ralf Ludwig[1], and Martin Leduc[2, 3]

[1]Department of Geography, Ludwig-Maximilians-Universität München, Munich, Germany
[2]Ouranos, Montréal,Québec, Canada
[3]Centre ESCER, Université du Québec à Montréal, Montréal, Québec, Canada

**Correspondence:** Andrea Böhnisch (a.boehnisch@lmu.de)

**Abstract.** Central European weather and climate is closely related to atmospheric mass advection triggered by the North Atlantic Oscillation (NAO), which is a relevant index for quantifying internal climate variability on multi-annual time scales. It remains unclear, however, how large-scale circulation variability affects local climate characteristics when downscaled using a regional climate model. In this study, 50 members of a single-model initial-condition large ensemble (LE) of a nested regional climate model are analyzed for a NAO–climate relationship. The overall goal of the study is to assess whether the range of NAO internal variability is represented consistently between the driving global climate model (GCM; the CanESM2) and the nested regional climate model (RCM; the CRCM5). Responses of mean surface air temperature and total precipitation to changes in the NAO index value are examined in a Central European domain in both the CanESM2-LE and CRCM5-LE via Pearson correlation coefficients and the change per unit index change for historical (1981–2010) and future (2070–2099) winters. Results show that statistically robust NAO patterns are found in the CanESM2-LE under current forcing conditions. NAO flow pattern reproductions in the CanESM2-LE trigger responses in the high-resolution CRCM5-LE that are comparable with reanalysis data. NAO–response relationships weaken in the future period, but their inter-member spread shows no significant change. The results stress the value of single-model ensembles for the evaluation of internal variability by pointing out the large differences of NAO–response relationships among individual members. They also strengthen the validity of the nested ensemble for further impact modelling using RCM data only, since important large-scale teleconnections present in the driving data propagate properly to the fine scale dynamics in the RCM.

## 1 Introduction

One of the major sources of uncertainty regarding short-term future climate projections is internal climate variability, while model climate response and greenhouse gases concentrations scenarios become more important sources of uncertainty on a longer-term time horizon (Hawkins and Sutton, 2009, 2011). The term internal variability denotes climate variability which is not forced by external processes (either anthropogenic or natural), but arises from the chaotic properties of the climate system itself (Leduc et al., 2019; Deser et al., 2012), i.e. from varying sequences of weather events under identical external forcings.

These sequences of weather events may be altered by global atmospheric modes of variability through the linking between large-scale circulation and local weather characteristics (like surface air temperature and precipitation). Such large-scale atmospheric modes can thereby establish periods of discernible states on multi-annual time scales.

Among these modes, the North Atlantic Oscillation (NAO) is particularly important for northern hemisphere climate. Its two states, positive and negative, are evoked by planetary wave-breaking in the polar front, leading to antagonistic pressure behaviour of two centres over the North Atlantic: one located within the subtropical high pressure belt ("Azores High", AH), the second in subpolar regions ("Icelandic Low", IL) (Benedict et al., 2004). The resulting pressure gradient, which is stronger during positive and weaker during negative phases, affects large-scale extra-tropical circulation, especially the strength and position of mid-latitude westerly winds connected to the jet stream, and air mass advection during boreal winter (Deser et al., 2017; Hurrell and Deser, 2009). Compared with neutral conditions, the positive NAO state leads to warmer and moister winters in northern Europe, but cooler and drier conditions in the south, and vice versa in the negative state (e.g., Hurrell and Deser, 2009; Pokorná and Huth, 2015; Woollings et al., 2015).

The NAO is commonly quantified with an index that makes use of the air pressure or geopotential height gradient between AH and IL. The index may be calculated as a normalized difference of station measurements, spatially averaged values of pre-set regions, or the region of highest variance is obtained by principal component analysis (PCA) (Pokorná and Huth, 2015; Hurrell and Deser, 2009; Stephenson et al., 2006; Hurrell, 1995; Rogers, 1984). Each method has its advantages and limitations. For example, station-based or fixed in space indices do not reproduce shifting NAO patterns and may be affected by micro-climatic noise and other teleconnection patterns (Hurrell and Deser, 2009; Osborn, 2004). Indices based on PCA on the other hand are dependent on the chosen data domain for calculation and on the data set itself (Osborn, 2004). The different approaches, however, lead to highly similar index time series (see e.g., Pokorná and Huth, 2015, for a detailed survey of various approaches).

While the typical NAO pattern and its impacts are usually correctly reproduced in global climate models (GCMs) (Stephenson et al., 2006; Ulbrich and Christoph, 1999; Reintges et al., 2017), its fidelity in a future climate remains uncertain: the NAO is found as intensifying, but also counteracting global warming in the northern hemisphere ("global warming hiatus", Iles and Hegerl, 2017; Deser et al., 2017; Delworth et al., 2016). Similarly, the findings regarding the prevalence of future positive or negative states lack unity: Some analyses of CMIP5 models, for example, suggest more positive phases under rising greenhouse gas concentrations until 2100 (e.g., Kirtman et al., 2013; Christensen et al., 2013), others favour an increase of negative phases (Cattiaux et al., 2013).

In most of these studies it was common to rely on one simulation per model and estimate the model's performance regarding the NAO by this single run. This approach allows for comparing different models (and observations). However, it is not possible to robustly evaluate the range of NAO index values and evolution in a projected future climate, or whether the chosen simulation is a good representation of how this model simulates the phenomenon in question (Leduc et al., 2019). Relying on single realizations possibly deteriorates the assessment of a given model, as single realizations may vary considerably among themselves due to internal variability (and also deviate from the climate evolution observed in reality). One way to sample realizations is to perturb the initial conditions of the model, leading to multiple simulations with identical external forcing which only differ due to internal variability. Examples for recent GCM initial-condition large ensembles of transient simulations are

the 100-member Max Planck Institute Grand Ensemble (MPI-GE, Maher et al., 2019), the 50-member Canadian Earth System Model Large Ensemble (CanESM2-LE, e.g. Kirchmeier-Young et al., 2017; Fyfe et al., 2017) or the 40-member Community Earth System Modelling Large Ensemble (CESM-LE Kay et al., 2015) which were, among others, used for various analyses of internal variability or extreme events. Such initial condition ensembles also allow a more robust distribution of atmospheric modes to be sampled, as was done e.g. for El Niño/Southern Oscillation in Maher et al. (2018). That is why the present study is investigating the NAO pattern in a single-model large ensemble of a GCM.

However, when interested in NAO impacts on a regional scale, like Central Europe, the GCM is not sufficient for fine-scale responses. Due to their coarse spatial resolution, GCMs are poorly resolving land–water contrasts and topographic characteristics which may be highly relevant in climate impact studies over heterogeneous landscapes (Leduc et al., 2019). Thus, dynamical downscaling of the GCM members using a regional climate model (RCM) is advised (Leduc et al., 2019). The downscaling of a GCM single-model large ensemble, the CanESM2-LE, was performed within the Climate Change and Hydrological Extremes project (ClimEx, www.climex-project.org, Leduc et al., 2019).

Examples of analyses on the separation of the forced signal from internal variability within a 16-member single-model initial condition GCM-RCM ensemble of EC-EARTH and RACMO2 were performed by Aalbers et al. (2018) for various extreme precipitation indices.

Combining the driving GCM and nested RCM (i.e., driven by lateral boundary conditions of the GCM) large ensembles (LE) allows for analyzing the spread of NAO states and responses within one model chain. In doing so, it is possible to establish the range of internal variability of the NAO and find robust NAO and response patterns by significantly reducing uncertainty associated with internal variability in the ensemble.

The present study targets the research question, how global circulation variability, in this case the NAO teleconnection, affects local climate characteristics when downscaled using an RCM. It specifically aims at evaluating whether the range of internal variability is represented consistently between the driving GCM-LE and the driven high resolution RCM-LE. The latter may be important for impact modellers who work with RCM data on internal variability without taking the driving GCM into account. To answer these research questions, this study is focussing on four topics and related key questions:

(a) General performance of the model chain: Can the driving GCM resolve the NAO correctly and are climatic implications for Central Europe reproduced?

(b) Nesting approach: Does the RCM correctly incorporate the NAO pattern present in the driving data and produce realistic response patterns?

(c) Internal Variability: What is the range of possible NAO patterns and responses, expressed by the inter-member spread among the 50 members of the GCM-LE and the RCM-LE?

(d) Climate change: How do (a), (b) and (c) change in transient climate simulations that extend until 2099 using an RCP8.5 emissions scenario?

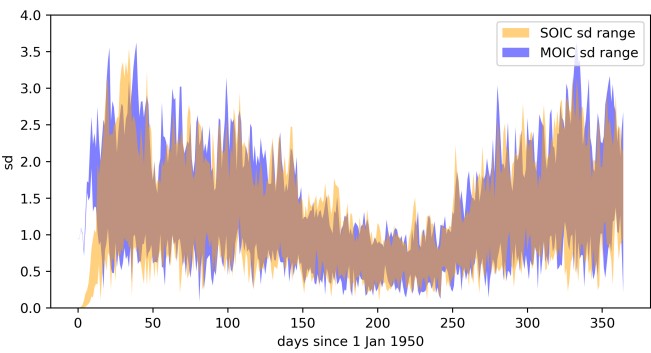

**Figure 1.** Inter-member standard deviation of a daily NAO index in the CanESM2-LE starting on 1 Jan 1950 as a function of time. The inter-member standard deviation (sd) is derived from 10 groups of 5 members with the same ocean initial conditions (SOIC) and 10 groups of 5 members with mixed ocean initial conditions (MOIC, following an approach in Leduc et al., 2019).

## 2   Data and Methodology

### 2.1   Data

Data from three different sources are employed in this study (Table 1). The major source is the RCM-LE data set of the ClimEx project which is described in detail in Leduc et al. (2019). The ClimEx project is conducted in a Québec-Bavarian cooperation and targets issues of hydrological extreme events in the time horizon of 1950–2099, using a nested high-resolution 50 member single-model initial-condition large ensemble with an RCP8.5 emissions scenario from 2006 onwards (Leduc et al., 2019). Five members of the Canadian Earth System Model version 2 (CanESM2 Large Ensemble, 2.8° spatial resolution, Fyfe et al., 2017) with different ocean inital conditions were slightly perturbed in 1950, leading to 10 members per ocean family. The members are assumed to become independent about 5 years after their initialization in 1950 (spin-up-period) (Leduc et al., 2019). Regarding the atmospheric circulation, Fig. 1 shows that owing to the chaotic nature of the atmospheric system the daily NAO index seems to lose dependence from the initial conditions within the course of one month after initialization (see Leduc et al., 2019, for a similar presentation of member independence).

As described in Leduc et al. (2019), the 50 CanESM2 members were dynamically downscaled using the Canadian Regional Climate Model version 5 (CRCM5 Large Ensemble, 0.11° spatial resolution) over two domains covering Europe and north-eastern North America, each sized $280 \times 280$ grid cells on a rotated grid. Large-scale spectral nudging of the horizontal wind field was applied during the nesting process (Leduc et al., 2019). This single-RCM 50-member ensemble allows for internal variability and extreme events to be detected in high spatial and temporal resolution within a total of 7500 modelled years (Leduc et al., 2019).

Comparing the internal variability of the CRCM5 members with the inter-member spread of a subset of the multi-model EURO-CORDEX (Coordinated Regional climate Downscaling Experiment) ensemble regarding regionally integrated European winter temperature and precipitation, von Trentini et al. (2019) showed that both ensemble spreads are of comparable magnitude. The

**Table 1.** Overview of used data sets, their spatial resolution, the number of members and the employed variables.

| data name | model type | spatial resolution | members | model output variable names | institution |
|---|---|---|---|---|---|
| ERA-I | re-analysis | $0.75° \times 0.75°$ | 1 | msl [Pa], t2m [K], tp [m] | ECMWF |
| CRCM5/ERA-I | RCM | $0.11° \times 0.11°$ | 1 | psl [Pa], tas [K], pr [kgm$^{-2}$s$^{-1}$] | Ouranos |
| CanESM2-LE | GCM | $2.8° \times 2.8°$ | 50 | psl [Pa], tas [K], pr [kgm$^{-2}$s$^{-1}$] | CCCma |
| CRCM5-LE | RCM | $0.11° \times 0.11°$ | 50 | psl [Pa], tas [°C], pr [mm] | Ouranos |

CCCma – Canadian Centre for Climate Modelling and Analysis

CORDEX ensemble consists of several GCM-RCM combinations set up in a coordinated modelling framework and aims at evaluating uncertainty due to model configuration (Giorgi et al., 2009). The comparison of the single-model and multi-model spreads suggests that a large fraction of the CORDEX ensemble spread regarding temperature and precipitation can be explained by internal variability, despite the fact that it was not explicitly sampled within the CORDEX framework (where most models provided a single simulation, von Trentini et al., 2019). At smaller regional scales, however, single-model and multi-model spreads may show considerable and in parts temporally changing differences which may partly be induced by model response uncertainties (von Trentini et al., 2019).

In the present study, model data is compared with the ERA-Interim (ERA-I) Reanalysis data set of the European Centre for Medium-Range Weather Forecasts (Dee et al., 2011, ECMWF). Additionally, a CRCM5 run driven by ERA-I is used to evaluate the CRCM5 under "perfect" (as far as ERA-I can be assumed to represent reality) lateral boundary conditions, i.e. without the potential CanESM2 data input error.

The relevant variables for this study are:

- (mean) sea level air pressure (referred to as "SLP", converted to [hPa]),

- near surface air temperature (referred to as "nSAT", converted to [K]),

- total precipitation including liquid and solid precipitation from all types of clouds (referred to as "PR", converted to [mm]).

ERA-I variables t2m, tp and msl (Table 1) are chosen as they are assumed to most accurately represent the GCM and RCM variables. As the variables derived from the three data sources are available at different temporal resolutions (three-hourly for tas and psl in RCM, hourly for pr in RCM, daily for psl, pr and tas in GCM, 6-hourly for ERA-I t2m and msl analysis, and 12-hourly for ERA-I tp forecast data), they are all aggregated to daily values first.

In the Appendix, Fig. A1 shows that the CRCM5 tends to underestimate (overestimate) mean winter nSAT mean in the northern (southern) part of the domain, regardless of the driving data (Fig. A1 (a) for ERA-I and (c) for CanESM2), whereas winter PR sums are overestimated in nearly the entire domain with strongest values in the south-eastern part (Fig. A1 (d) and (f)). Displaying opposite bias to CRCM5, the CanESM2 overestimates (underestimates) mean winter nSAT in the northern (southern) part of the domain (Fig. A1 (b)), whereas winter PR sum is underestimated in the eastern half of the domain and overestimated

on the western side of the Alps (Fig. A1 (e)). As this study will focus on responses of nSAT and PR induced by the NAO (see Section 2.2.4), aside from regions with particularly high PR sum values, it is found that such NAO responses are generally insensitive to these biases.

Commonly, NAO impact studies focus on seasonally aggregated values of the analyzed variables or extreme events (e.g.,
Stephenson et al., 2006). Yet the NAO, which accounts for variations in the mean zonal atmospheric flow towards Europe, can be assumed to not only influence winter mean values, but also their inter-annual variability. So, in addition to analyses of winter mean temperature (nSAT mean) and precipitation sums (PR sum), selected analyses are also performed on winter mean monthly standard deviations of daily mean temperature (nSAT sd) as a measure of temperature variability.

## 2.2 Methodology

### 2.2.1 Regions of interest and time horizon selection

Analyses are performed on time series of spatially averaged information (nSAT mean, PR sum for response variables and SLP for index calculation) as well as on spatially explicit data (nSAT mean, nSAT sd, PR sum). All data are provided as netCDF and most pre-processing is performed using the Climate Data Operators (CDO) of the Max-Planck-Institute for Meteorology
(Schulzweida, 2017).

The regions of interest and their names used in this study are displayed in Fig. 2. The formation of the NAO over the North Atlantic (NAR, AH, IL regions) is analyzed in the ERA-I and CanESM2-LE data, while responses over Central Europe (CEUR, NE, BY, SE regions) are evaluated in ERA-I, CRCM5/ERA-I, CRCM5-LE and CanESM2-LE data.

AH and IL regions are centered over Ponta Delgada/Azores and Reykjavik/Iceland, two commonly used stations for NAO
index calculations. To avoid micro-climatic impacts and sampling uncertainties of a single gridcell and to account for moving SLP centres (see e.g., Moore et al., 2013), both NAO core regions are extended to $3 \times 3$ GCM grid cell matrices. In preliminary analyses conducted for the present study, the NAO index has proven to be very robust towards the exact shape of the core regions.

The Central European domain (CEUR) is defined in the CanESM2-LE by selecting a $5 \times 5$ GCM grid cell matrix centered over
160 Munich/Germany. This CEUR domain extends from Denmark in the north to mid-Italy in the south and from Poland to France in east–west direction. The corresponding CEUR region within the ClimEx European domain is used to quantify the impacts of the NAO in the CRCM5-LE data. It lies downstream of the westerly flows initiated by the NAO, so the following analyses set a special focus on the incorporation of large-scale inflow from the western side into the nested RCM.

As the responses to the NAO are expected to vary over the CEUR domain, it seems favourable to analyze spatial structures
explicitly in addition to analyses of time series over several subset regions. These subset regions (see e.g., Déqué et al., 2007) denote small-scale sample areas inside the CEUR domain, sized one GCM grid cell each, with expected typical "northern European" (NE) and "southern European" (SE) NAO responses for a more detailed statistical analysis. A third GCM grid cell is chosen to represent the transition zone between NE and SE. Coincidentally, it closely represents the region of Bavaria which

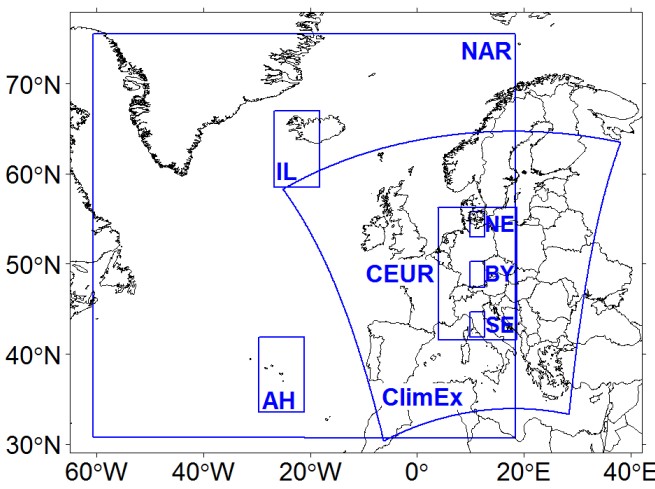

**Figure 2.** Regions of interest. Abbreviations and domain sizes in terms of GCM grid cells ($2.8°$) are as follows: AH – Azores High ($3 \times 3$); IL – Icelandic Low ($3 \times 3$); NAR – large-scale North Atlantic region ($28 \times 16$); CEUR – Central Europe ($5 \times 5$); NE – northern Europe (1); BY – Bavaria (1); SE – southern Europe (1); ClimEx – domain used in ClimEx project (extent approximately $22 \times 12$ after resampling to GCM grid).

is why the name "BY" is assigned to it. ERA-I and RCM data ($3 \times 4$ and $26 \times 26$ grid cells, respectively) is spatially aggregated to GCM resolution for this part of the analysis.

This study focuses on inter-annual analyses which are conducted for two time horizons covering 30 years each. The historical (hist; 1981–2010) period is used to establish reference statistics in the ERA-I data and the ERA-I driven CRCM5 run which are then evaluated in the GCM-LE and the RCM-LE. Links and relationships established for the historical period are also investigated in a far future horizon (fut; 2070–2099).

The chosen period length is assumed to include major fluctuations, like internal climate variations or several solar cycles, which might affect NAO phases (Andrews et al., 2015). Thus their influence can be assumed to be represented by the sampled NAO time series. Relationships between the NAO and response variables most probably vary on different time scales (Hurrell and Deser, 2009; Woollings et al., 2015; Xu et al., 2015; Hurrell and Van Loon, 1997). However, as 30 year periods are not long enough for analyses of multi-decadal (>30 years) NAO–response variability (Woollings et al., 2015), stationarity in NAO-impact relationships is assumed for simplicity reasons.

Since the NAO is known to be strongest in winter (Hurrell and Deser, 2009) and the connection between station-based indices and NAO responses tends to be best in winter (see Pokorná and Huth, 2015, for months DJF), analyses are performed for this season only. Preliminary tests within this study have shown that correlations and links between the NAO index and the climate variables are more distinct from noise, if March is included as well. That is why an extended winter season is used here (DJFM, see also Iles and Hegerl, 2017; Hurrell, 1995; Osborn, 2004).

All data (spatially explicit and subset time series) is aggregated to the seasonal time scale for further use (winter means for

nSAT and winter sums for PR).

### 2.2.2 Deriving an NAO index

The NAO index is derived from ERA-I and CanESM2-LE data, resulting in 1 ERA-I and 50 GCM realizations. As the CRCM5 ClimEx domain does not cover the AH and IL regions (see Fig. 2), the index is not derived from this data source. The NAO is quantified in this study with an index which is closest to a station based or zonally averaged index. It therefore directly represents the winter SLP gradient over the North Atlantic.

   The time series of AH and IL originate from the temporally shortened and spatially averaged SLP time series of both grid cell

matrices. Daily SLP values are averaged to monthly means (Cropper et al., 2015) and scaled to obtain mean $\mu = 0$ and standard deviation $\sigma = 1$, as outlined in Osborn (2004) and Hurrell and Van Loon (1997), by subtracting the 1981–2010 seasonal mean (overbar) and dividing by the 1981–2010 seasonal standard deviation ($s_{IL}, s_{AH}$):

$$\text{NAOIndex} = \frac{AH - \overline{AH}}{s_{AH}} - \frac{IL - \overline{IL}}{s_{IL}} \tag{1}$$

   Monthly indices are next averaged to DJFM means. This approach is similar to Woollings et al. (2015) and Jones et al. (2013).

The ERA-I NAO index caluclated this way shows high agreement with often cited NAO indices like the time series of Hurrell (Pearson correlation of $r = 0.95$ with ERA-I NAO index; index available at https://climatedataguide.ucar.edu/climate-data/hurrellnorth-atlantic-oscillation-nao-index-station-based). For further analyses it will therefore serve as a reference.

   To compare future with historical index values, the future time series of AH and IL are standardized with the historical SLP standard deviations (see also Ulbrich and Christoph, 1999; Hansen et al., 2017) and mean values. The standardization of each

GCM member is carried out individually.

### 2.2.3 Evaluation of the large-scale SLP pattern in RCM data

   To estimate whether the NAO may be seen as being correctly represented in the nested RCM data, the reproduction of inter-annual SLP pattern variations in the CRCM5 data is verified. Therefore, monthly mean SLP data of the CRCM5 (both driving

data sets) and ERA-I are linearly interpolated to GCM resolution over the ClimEx domain. During interpolation, small scales are automatically filtered such that the remaining large scales of driving data and RCM data may be compared. As a next step, a root-mean-square difference (RMSD) of the difference time series between monthly mean driving and RCM data over the hist and fut time periods is obtained across all members and winter months:

$$\text{RMSD}(i,j) = \left\langle \left\langle \frac{\sqrt{\langle D_m(i,j,t,n)^2 \rangle_t}}{\sqrt{\text{VarDrive}_m(i,j,n)}} \right\rangle_n \right\rangle_m \tag{2}$$

$$\text{VarDrive}_m(i,j,n) = \left\langle (\text{Drive}_m(i,j,t,n) - \langle \text{Drive}_m(i,j,t,n) \rangle_t)^2 \right\rangle_t \tag{3}$$

where $\langle \cdot \rangle$ is the averaging operator over a given index, $D_m$ is the difference between monthly mean driving data and RCM data; $\text{Drive}_m$ is driving SLP data; $\text{VarDrive}_m$ is the variance of SLP driving data over the 30 year periods; $i, j$ are spatial grid coordinates, $m$ are months 12, 1–3, $n$ are ensemble members 1–50 for CanESM2 and 1 for ERA-I, and $t$ are years in 1981–2010 and 2070–2099. The normalization by the square root of the temporal variance of the driving data provides a measure relative to the inter-annual variability of the SLP pattern in a given location.

### 2.2.4 Climatic Changes Associated with NAO

All data sources (Table 1) are used to obtain response patterns of the variables nSAT and PR. Climatic changes associated with the NAO are evaluated using Pearson correlation coefficients and a slope parameter obtained by linear regression. ERA-I and CRCM5/ERA-I nSAT and PR data are correlated with the ERA-I index, CanESM2 and CRCM5 members are correlated with the CanESM2 index calculated for the corresponding member.

The correlation analysis assumes (symmetric) linear relationships between the NAO index and nSAT or PR. The associated response of the variables to NAO changes may then be quantified by a linear equation (Iles and Hegerl, 2017; Stephenson et al., 2006; Hurrell, 1995):

$$Y = \alpha_1 X + \alpha_0 + \varepsilon_Y \tag{4}$$

with $Y$ being the (response) variable at a given grid cell that is partly explained by the NAO ($X$, the predictor) and by any other influences ($\varepsilon_Y$; Stephenson et al., 2006; von Storch and Zwiers, 2003). The coefficient $\alpha_1$ is estimated on each grid cell using ordinary least squares regression with the R function lm (www.rdocumentation.org). It represents mean change in nSAT or PR that accompanies unit index change during the time period under consideration (Iles and Hegerl, 2017). The line offset $\alpha_0$ in Eq. (4) equals the long-term mean. The $\alpha_1$ coefficients may be computed with respect to normalized index series (von Storch and Zwiers, 2003), but in this study the non-normalized index time series is preferred in order to take into account the member-specific index units. The NAO–response relationship is analyzed individually for each GCM and RCM member (as is done e.g. in Woollings et al., 2015).

### 2.2.5 Addressing Internal Variability

In this study, the GCM-RCM combination allows to set a focus on the internal variability of an RCM ensemble and the driving GCM ensemble. Climate modes tend to show high internal variability (see e.g. Maher et al., 2018, for an analysis of ENSO internal variability in CMIP5 models and two single-model large ensembles). The present study targets the NAO-related internal variability within a single GCM-RCM combination.

In general, natural internal variability may be understood from different angles. When looking into single realizations of time series of a given variable, internal variability may be seen as represented by the oscillation around the long-term mean evolution, i.e. the residuals (Frankcombe et al., 2015; Hawkins and Sutton, 2011, 2009). In this case, the amplitude of internal variability is usually calculated as a time-invariant quantity for the period under consideration (Hawkins and Sutton, 2009, 2011).

Another way is investigating transient internal variability in initial-condition ensembles, like e.g. in Maher et al. (2019). In this case, the ensemble establishes ranges of possible weather event sequences by superposing single realizations which are equally likely by construction of the ensemble.

In the present study, the latter approach is used within the 50-members CanESM2-LE and CRCM5-LE. This allows to sample internal variability at single points in time as the range of the members' values, i.e. across members (e.g., Maher et al., 2018). While internal variability is assumed to be stationary within both 30-year periods for this study, the use of a LE allows to detect potential changes in internal variability between both analysis periods.

Internal variability is expressed as the across-member standard deviation, i.e. the inter-member spread of the CanESM2-LE and the CRCM5-LE (see also Leduc et al., 2019; Déqué et al., 2007; Aalbers et al., 2018) among the 30-year means, rather than computing a transient internal variability at each time step as was done e.g. in Maher et al. (2019). Aggregations to ensemble means (like in Deser et al., 2017; Aalbers et al., 2018) of NAO responses are only performed for illustrating purposes in order to avoid masking model internal variability (Zwiers and von Storch, 2004).

# 3   Results

The result section is structured in two large parts: Section 3.1 deals with the representation of the NAO and climatic responses in the GCM and RCM and Section 3.2 targets internal variability in the GCM and RCM.

## 3.1   NAO within the ClimEx Data Set

Naturally, the first step when evaluating the NAO in a model ensemble is to analyse its representation and index distribution in the model data of interest.

### 3.1.1   NAO index and SLP conditions

The CanESM2-LE produces NAO index values which follow a distribution comparable to the ERA-I data (similar to a normal distribution with $\mu = 0, \sigma = 1$, Fig. 3 (a)), but the CanESM2-LE distribution appears smoother due to a larger sample size ($n = 1500$ for CanESM2-LE and $n = 30$ for ERA-I). Maximum and minimum index values (x-axis in Fig. 3 (a)) of some of the 50 members exceed those of the ERA-I realization; thus, ERA-I which serves as a reference realization lies well within the ensemble inter-member spread. The future NAO index shows a similar distribution of values, but with slightly less positive and more negative values (red curve in Fig. 3 (a)).

For the following analyses independence of the 50-member ensemble is critical to interpreting the inter-member spread as a proxy for internal variability. In evaluating this, it is important to recall that the 50-member CanESM2-LE was constructed in two steps (Fyfe et al., 2017; Leduc et al., 2019): First, independent atmosphere/ocean states in 1850 were used to launch 5 historical simulations integrated forward until 1950. Second, in 1950, each of these 5 ensemble members were used to launch 10 individual simulations by applying a small perturbation to the atmosphere and integrated forward until 2099, thereby pro-

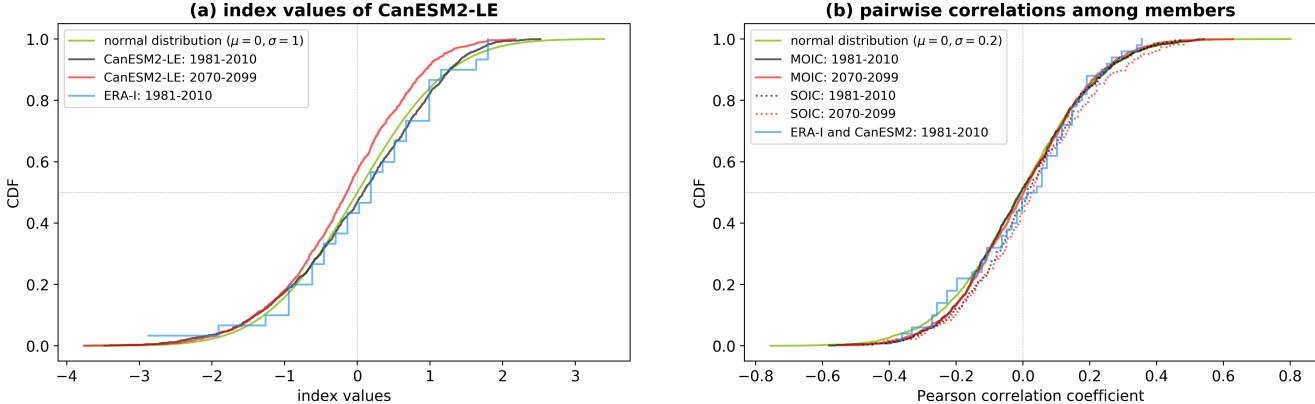

**Figure 3.** Cumulative density functions (CDFs) of NAO index values. (a) distribution of all CanESM2-LE ($n = 50 \times 30$ per period) and ERA-I ($n = 30$) NAO index values. (b) pairwise correlations among member NAO index time series from the same ocean families (SOIC – same ocean initial conditions, dotted lines, $n = 225$), from different ocean families (MOIC – mixed ocean initial conditions, solid lines, $n = 1000$) and between ERA-I and all CanESM2 members ($n = 50$). Black: 1981–2010 CanESM2-LE, red: 2070–2099 CanESM2-LE, blue: 1981–2010 ERA-I, green: normal distribution with $\mu = 0$ and $\sigma = 1$ in (a) and $\sigma = 0.2$ in (b).

ducing the 50-member large ensemble.

As a consequence, for this study, members between each of the 5 groups of 10 are expected to be independent. However, members within each group of 10 are highly correlated in 1950 and progressively increase their independence beyond their 1950 starting point. To evaluate whether the 10 members within each of the 5 groups have become sufficiently independent by the two 30-year periods of interest (1981-2010 and 2070-2099), correlations among member time series are applied to two groups following Leduc et al. (2019): (i) correlations among the 10 members from the same group (same ocean initial conditions –

SOIC, $n = 225$ cases, dotted lines in Fig. 3 (b)) and (ii) correlations between each member and the 40 members from the 4 other groups (mixed ocean initial conditions – MOIC, $n = 1000$ cases, solid lines in Fig. 3 (b)).

These correlations approximately follow a normal distribution with $\mu = 0$ and $\sigma = 0.2$. There is a slight surmount of low positive correlations in the SOIC group compared with the MOIC group which is (not significantly) stronger in the fut time horizon (see red and black dotted lines in Fig. 3 (b)). Although zero correlations do not necessarily imply independence, clear

correlations among members would contradict the assumption of independence. In general, the members are thus not seen as being dependent.

As will be discussed below, the SLP pattern over the North Atlantic changes slightly in the future period. So the direct comparison between historical and future SOIC and MOIC correlations remains difficult. The members also show no systematic correlation with the ERA-I NAO index despite similar statistics (see also Fig. 9). Thus, the ERA-I and GCM indices can be

seen as not dependent realizations drawn from the same distribution.

In order to further evaluate the NAO representation in the CanESM2-LE, Fig. 4 presents the large-scale SLP patterns in the

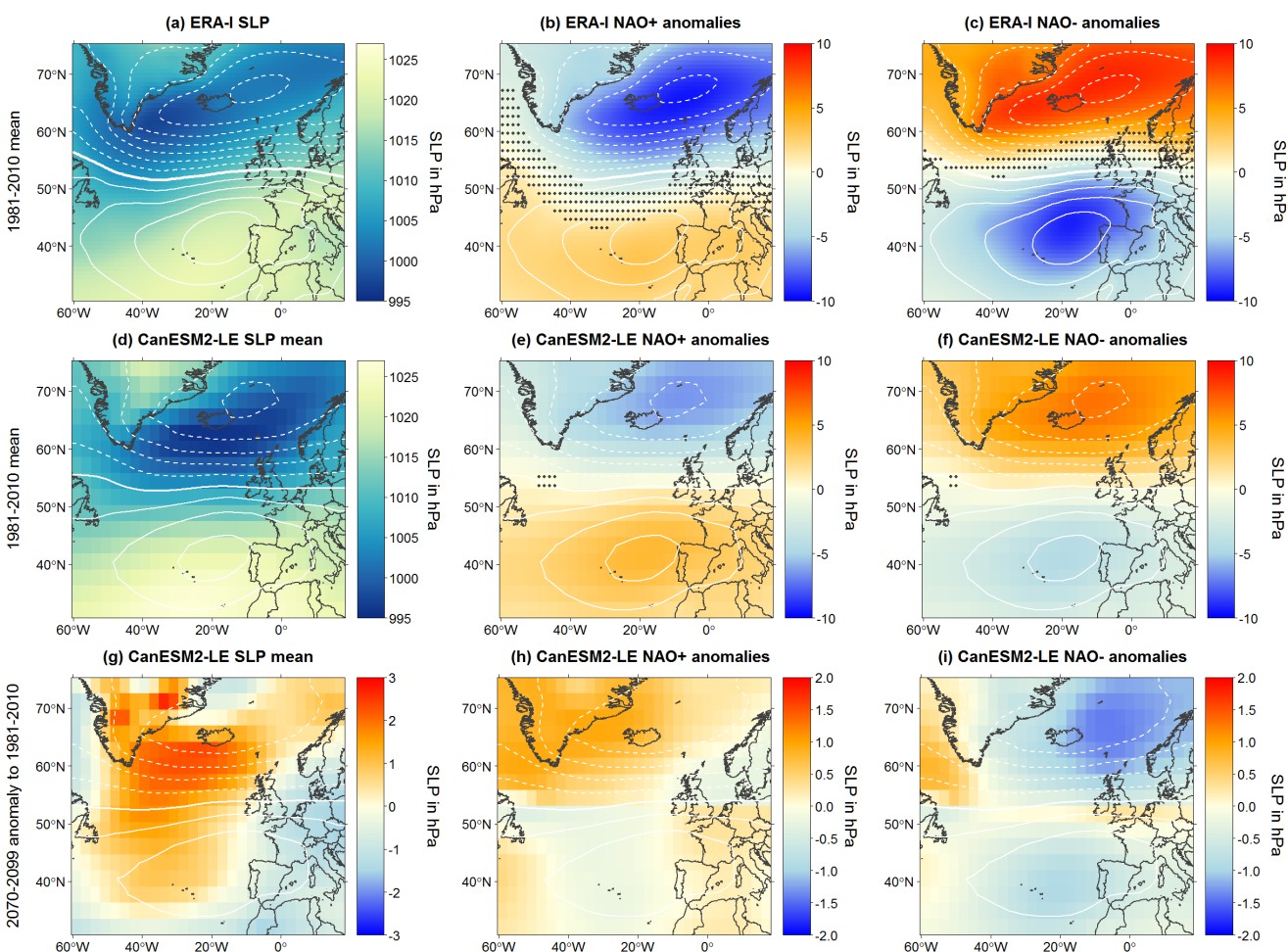

**Figure 4.** NAR winter mean SLP [hPa] composites in ERA-I ((a)–(c)) and CanESM2-LE ((d)–(i)) data showing long-term neutral conditions (left column), NAO positive (mid column) and negative anomalies (right column). (a)–(f): for 1981–2010, (g)–(i): 2070–2099 changes with respect to 1981–2010 in GCM data. White isolines: difference between positive and negative anomalies by a step of 2.50 hPa, as e.g. in Hurrell (1995), solid: positive, dashed: negative, bold line: zero. Grey stippling in subpanels (b)–(c) and (e)–(f): regions where the anomaly is smaller than the standard error of the composite samples.

NAR region during neutral, positive and negative NAO conditions. Positive (negative) index years are chosen, if the respective index value exceeds $1$ ($-1$) as in Rogers (1984). The neutral conditions refer to the 30 year SLP average. Regions with strong sampling uncertainties, i.e. where the standard error is larger than the anomaly, are indicated with stippling in panels (b)–(c) and (e)–(f).

Under neutral NAO conditions, the North Atlantic region is characterized by a pressure dipole. This structure is intensified and tilted clockwise in the CanESM2-LE ensemble mean (Fig. 4 (d)) compared with ERA-I (Fig. 4 (a)). The mean SLP difference

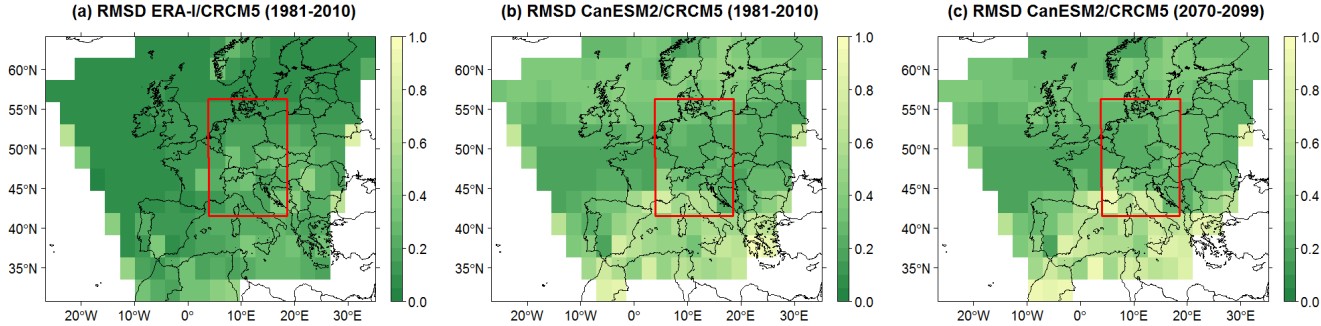

**Figure 5.** RMSD of monthly SLP differences between driving data and CRCM5 members, calculated following Eq. (2). Colouring: RMSD $\leq 1$ significant at $p \leq 0.05$ with a false detection rate smaller than 0.1 (see Wilks, 2016). (a) for driving data ERA-I (1981–2010, one realization), (b) for driving data CanESM2-LE (1981–2010, 50 members), (c) for driving data CanESM2-LE (2070–2099, 50 members). Red box: position of CEUR domain.

between the CanESM2-LE mean and ERA-I reaches up to 10 hPa in both directions. SLP values are higher over Greenland and lower over the North Sea in the CanESM2-LE compared with ERA-I (Fig. 4 (a), (d)). Long-term neutral states of both driving data sources show robust signals in the entire NAR region (i.e., no stippling). This suggests that the different patterns in GCM and reference data are not singularly artefacts arising from different sample sizes, but rather robust features.

The GCM multi-member composites of positive and negative phases show less pronounced SLP anomalies than the reference data (Fig. 4 (b)–(c) and (e)–(f)). Transition regions between the AH and IL nodes are marked by high uncertainty in ERA-I, whereas the SLP anomalies at the NAO centres of action show less uncertainty. The GCM patterns are more robustly assessed (i.e., less prone to sampling uncertainty) as can be seen by the very small area with stippling in which the sign of the anomaly may not be assessed robustly in Fig. 4 (e)–(f). So the difference between CanESM2-LE and ERA-I NAO anomalies may be due to the fact that ERA-I composites are derived from 3 negative and 4 positive years whereas the GCM data provides 264 negative and 263 positive years during 1981–2010.

The difference between SLP anomalies in positive and negative years representing the pressure variability is indicated by white lines. These NAO centres of action reach GCM (ERA-I) SLP differences between positive and negative conditions of about 12.5 (17.5) hPa in the IL region and 7.5 (10.0) hPa in the AH region. They do not coincide with the highest and lowest SLP values in the neutral state, but are situated near the $3 \times 3$ GCM grid cell matrices used for index calculation (see Fig. 2). This supports the choice of these SLP centres for index calculation.

Under projected future climate conditions, SLP rises over large parts of the North Atlantic and shows less variability (see Fig. 4 (g)–(i)). Future positive phases tend to be weaker as SLP shows a marked increase in the northern NAO node region. Negative phases exhibit SLP decreases in both node regions, although with larger changes near IL, resulting in negative phases to become slightly weaker as well.

Having established a reasonably plausible representation of the NAO in the driving data, the next step is to evaluate the large-scale NAO pattern in the RCM data. This is achieved by analysing the deviations of RCM and driving data SLP variability.

Figure 5 maps the RMSD between driving data and RCM SLP during 1981–2010 for driving data ERA-I (a) and CanESM2-LE (b), and CanESM2-LE in 2070–2099 (c). An $O(1)$ value of RMSD would indicate a poor reproduction of the SLP signal in the RCM because the RMSD between the RCM and driving data SLP is of the same order as the variability of the SLP in the driving data itself. Values of RMSD $<< 1$, on the other hand, would indicate a good reproduction of the SLP signal in the RCM because it suggests that the RCM is tracking the variability in the driving data. With this understanding it can be seen that the large-scale SLP pattern is reasonably well represented in most parts of the entire ClimEx domain for both driving data sets and both periods (significant at $p \leq 0.05$ using a t-test with a false detection rate $< 0.1$ to account for multiple hypothesis testing, see Wilks, 2016). All subpanels in Fig. 5 show an RMSD increase towards the south, indicating that in these regions the control exerted by the lateral boundary conditions on the CRCM5 internal solution appears to be weaker. The RMSD is larger in the CanESM2/CRCM5 combination than in the ERA-I/CRCM5 combination, and slightly increases in the future period in the southern parts (Fig. 5 (c)). The differences of the spatial patterns are most likely due to different large-scale SLP patterns in both driving data sets which are in parts visible in Fig. 4 (a) and (d). In the CEUR domain (red box in Fig. 5), however, errors are low in general and therefore the NAO pattern of the driving data may be assumed to be correctly incorporated there. It is thus reasonable to continue with the evaluation of nSAT and PR responses in the CEUR domain.

### 3.1.2 Local climate response to the NAO

nSAT and PR spatial responses as revealed in the ERA-I data are generally reproduced under current climate conditions in the CanESM2-LE and CRCM5-LE (see Figs. 6–8). Highest magnitudes of the NAO-responses (i.e., the slope of the regression line, $\alpha_1$, introduced in Eq. (4)) occur in the CRCM5/ERA-I run for all variables. In general, the CRCM5 produces stronger $\alpha_1$ response values at the local scale than the driving data. Regarding the absolute $\alpha_1$ values, the CRCM5-LE mean meets the ERA-I better than the CRCM5/ERA-I run.

Positive NAO conditions are accompanied by winters with warmer temperatures (up to $+2$ K per unit index change, Fig. 6) and less day-to-day nSAT variability compared to neutral conditions (Fig. 7). The mainly positive relationship between nSAT mean and the NAO (Fig. 6) is strongest in the north-eastern parts of the domain. Regionally, the NAO explains up to 40–60 % of nSAT mean variability (see also Fig. A2 where the nSAT mean $\alpha_1$ share of the entire winter standard deviation of daily temperature values is shown). Explained variance is highest in the CRCM5/ERA-I run and lowest in the CanESM2-LE.

The reduction of nSAT variability reaches up to 0.4–0.6 K in the northeastern continental section while it is near zero in the southern part of the domain (Fig. 7).

In comparison to the neutral state, positive phases are also accompanied by more humid conditions in the north, and drier conditions in the south of the CEUR domain (Fig. 8). The strength of the NAO–PR relationship, expressed by a Pearson correlation coefficient $r$, is not affected by topography in any of the models within the domain; only the pivotal line crossing Europe is following the Alpine ridges (see solid dark line in Fig. 8, panels (a)–(d)). The change between positive and negative $r$ and $\alpha_1$ occurs within a very narrow region. Within the CanESM2-LE, this zero-line is shifted northwards compared with ERA-I, CRCM5/ERA-I and CRCM5-LE. As is visible in Fig. 8, higher $\alpha_1$ values in mountainous regions indicate strong NAO responses related to orography. Regionally, the NAO accounts for 40–50 % of total PR sum variance, in both positively

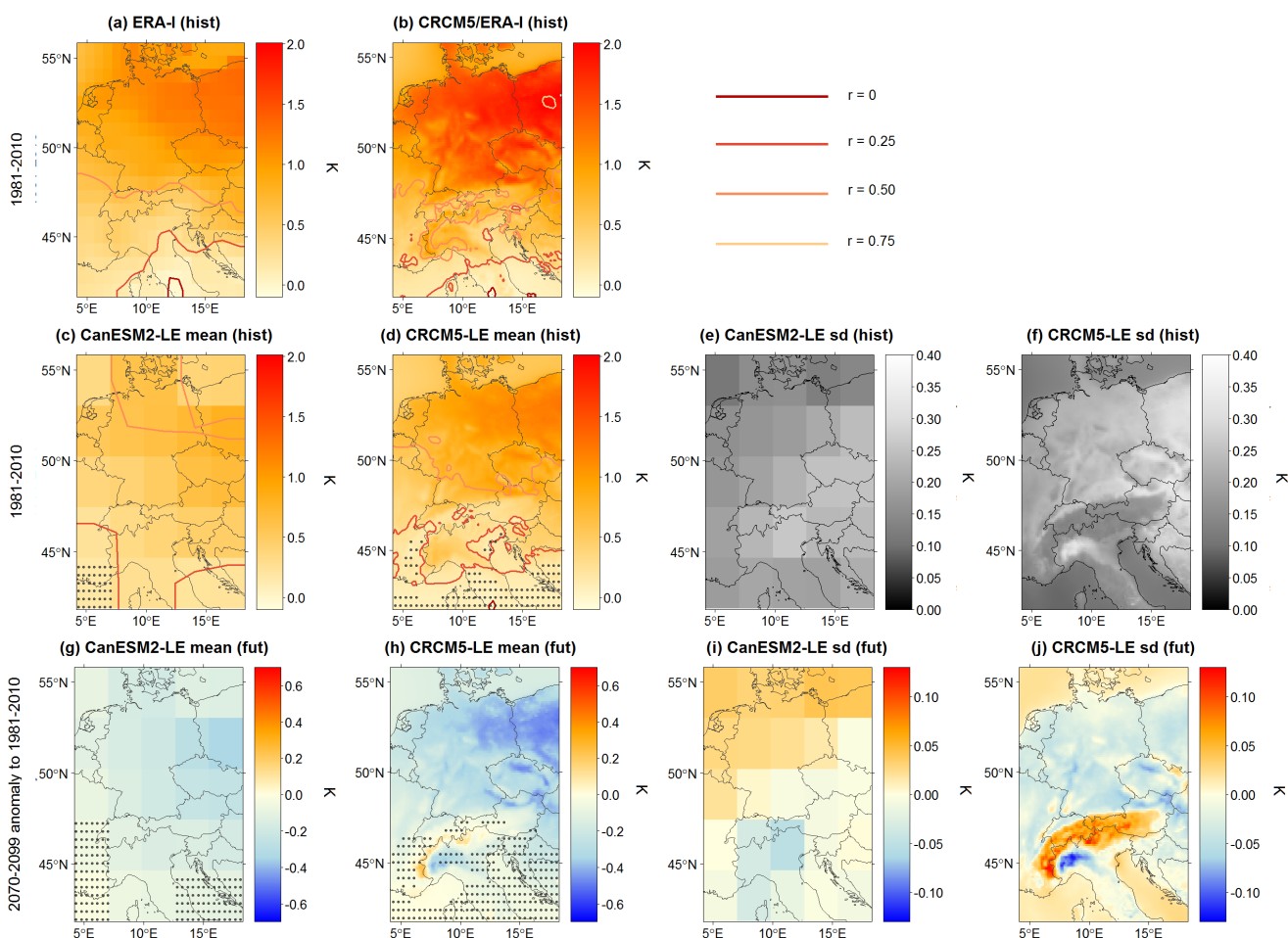

**Figure 6.** Spatial patterns of change in nSAT mean ($\alpha_1$ in [K]) for a unit change in the NAO index for ERA-I, CRCM5/ERA-I, CanESM2-LE and CRCM5-LE in 1981–2010 ((a)–(f)) and the change in 2070–2099 with respect to 1981–2010 ((g)–(j)). Both 50-member ensembles are represented with ensemble mean ((c)–(d), (g)–(h)) and standard deviation (sd, (e)–(f), (i)–(j)) representing the inter-member spread. Reddish lines in the ensemble mean maps represent the Pearson correlation between nSAT mean and the NAO index at an increment of 0.25; red shadings see legend in upper right panel. Grey stippling in the ensemble mean maps show regions where $\mathrm{SNR} < 1$, SNR being the signal-to-noise ratio between the 30 year ensemble mean and sd of GCM and RCM LEs in both time periods. Stippling will be explained in more detail in Section 3.2.2.

and negatively correlated regions. In the CRCM5-LE, single spots in mountainous regions (e.g., in the Dinaric Alps) show extremely high PR sum $\alpha_1$ values (up to $\pm 220\,\mathrm{mm}$ per unit index change). In these parts the long-term mean PR sums are also very high. This stresses the more detailed production of geographical features, but also the tendency to evolve local extreme values in the high-resolution RCM (see similar results for local daily extreme precipitation in Leduc et al., 2019) which may even be noted in the (spatially aggregated) bias towards the GCM (see Fig. A1 (f)). PR sum shows only weak correlations in

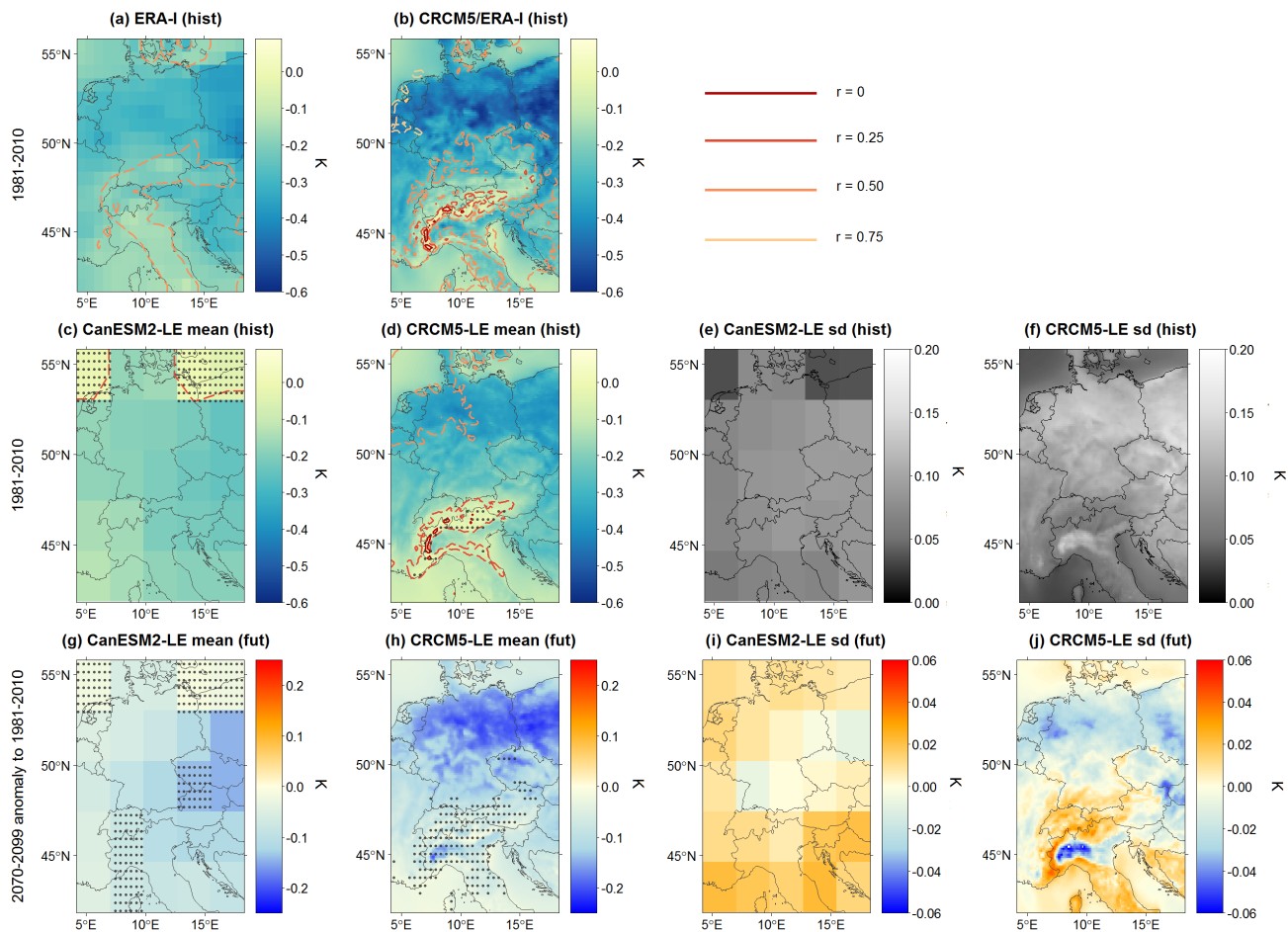

**Figure 7.** Like Fig. 6, but for nSAT sd ($\alpha_1$ in [K]). Dashed lines of correlation coefficients indicate negative values. Note that the difference maps for CanESM2-LE and CRCM5-LE mean are calculated using absolute values.

the central region of the CEUR domain.

The mean state of nSAT and PR changes in the transient climate simulation towards warmer and moister conditions with less intra-seasonal variability of nSAT. For a detailed description of the future climate evolution (though for 2080–2099) in Europe within the CRCM5-LE see Leduc et al. (2019). Future NAO–climate relationships weaken in general compared with the historical ones for all variables as can be inferred from the ensemble mean changes in panels (g)–(h) of Figs. 6–8. The spatial patterns of NAO-induced change do not change considerably between both periods. The response to the NAO, $\alpha_1$, is clearly

reduced in nSAT mean as well as nSAT sd, and there is also a reduction in PR sum change (panels (g)–(h) in Figs. 6–8).

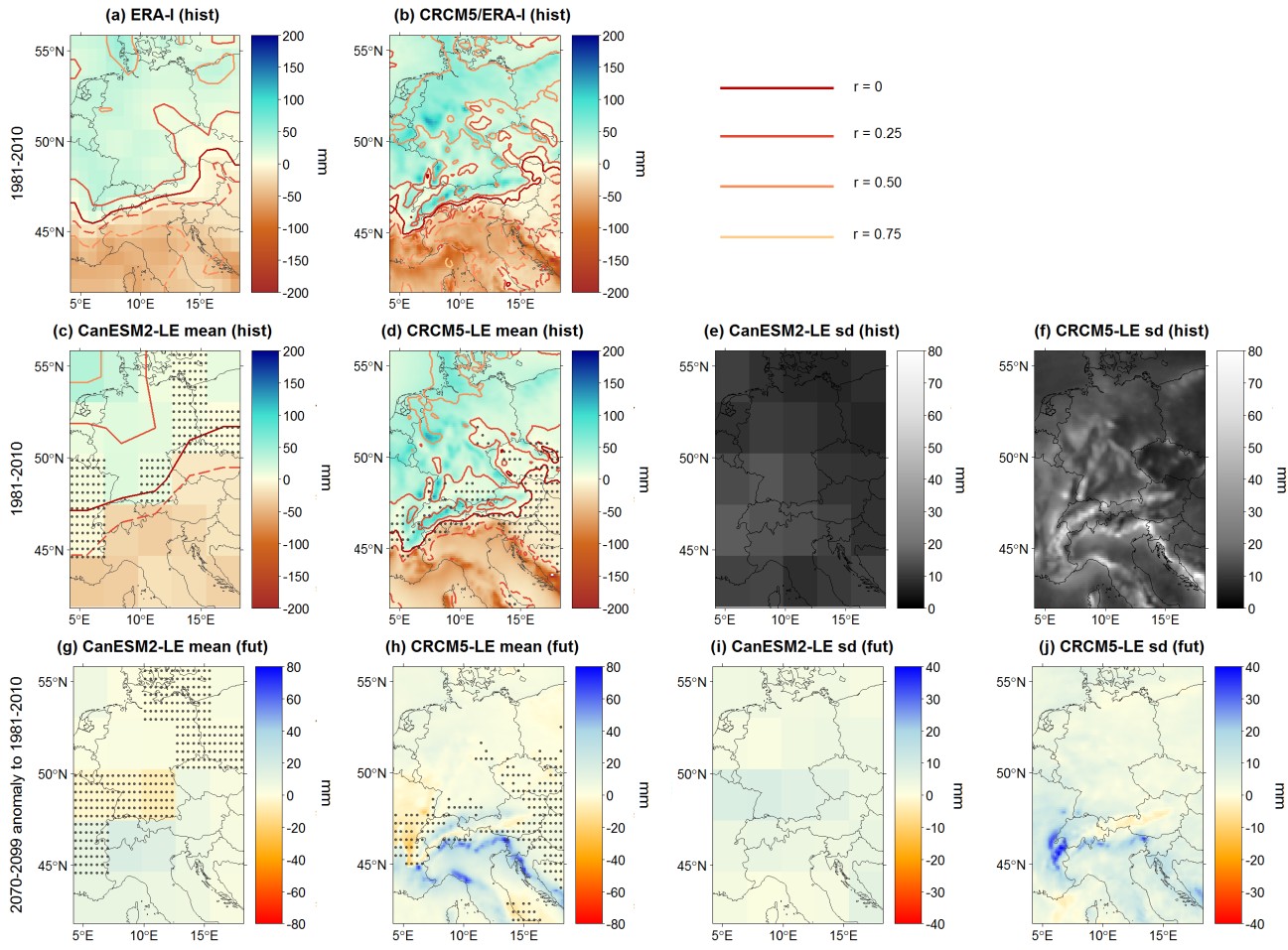

**Figure 8.** Like Figs. 6–7, but for PR sum ($\alpha_1$ in [mm]). Dashed lines of correlation coefficients indicate negative values. Note that the difference maps for CanESM2-LE and CRCM5-LE mean are calculated using absolute $\alpha_1$ values and that the colour bar in the bottom row is flipped compared with Figs. 6–7.

## 3.2 Internal Variability at the GCM and RCM scale

The next section focuses less on the ensemble mean changes, but rather their internal variability. The representation of internal variability in the GCM and RCM regarding the responses to the NAO in CEUR and subset regions NE, BY, SE is assessed via 375 the inter-member spreads of the CRCM5-LE and the CanESM2-LE, and their differences.

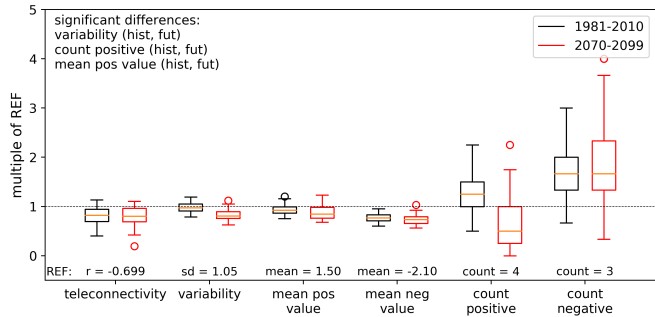

**Figure 9.** Several index statistics of all 50 CanESM2-LE members expressed as multiples of the respective ERA-I value (ERA-I value set to 1.0): teleconnectivity (Pearson correlation between AH and IL time series), index variability (expressed as temporal standard deviation of index time series), mean value of all positive (negative) phases and count of all positive (negative) phases per realization. Positive (negative) years are defined by an absolute index value exceeding 1. Text denotes combinations of which the differences are significant at $p \leq 0.05$ using an unpaired Mann-Whitney/U-test. Orange line in boxplots: median.

### 3.2.1 Multi-member ensemble

The CanESM2-LE reproduces typical NAO index characteristics: Fig. 9 summarizes several statistics for all 50 GCM members as multiples of the reference, i.e. ERA-I, value. Generally, the ensemble meets the ERA-I value in all aspects of the NAO index. However, some GCM members only reach half of the ERA-I teleconnectivity values (minimum correlation between AH/IL time series: $r = -0.281$, not significantly different from zero at $p \leq 0.05$ using a t-test; ERA-I $r = -0.699$). This finding is especially interesting as this metric quantifies the strength of the NAO within the individual members. The inter-member spread of the teleconnection strength does not change significantly over time, in spite of the SLP changes over the North Atlantic. The 2070–2099 NAO index exhibits less inter-annual variability, less positive phases, more neutral phases and a relative increase of negative phases but with reduced mean values (see also Fig. 3 (a)).

The spatial NAO-responses also show a considerable degree of internal variability. Its spatial distribution expressed by diverging ensemble members can be derived from Figs. 6–8 (e)–(f) presenting spatially distributed ensemble standard deviation (sd) as a measure of inter-member spread. Locally, the RCM shows considerably higher spreads than the GCM. Largest deviations for nSAT mean are found in continental regions of CEUR, but they do not simply correspond to high or low $\alpha_1$ (see also Fig. A3 (a)–(d)). Low inter-member spread corresponds mostly to Alpine and sea regions. The stippling in Figs. 6–8 (c)–(d) and (g)–(h) indicates regions where the variability among the members is larger than the ensemble mean response, i.e. where the signal-to-noise ratio (SNR) between ensemble mean and sd lies below 1. For nSAT mean, the SNR exceeds 1 in most regions north of the Alps (Fig. 6 (c)–(d)). nSAT sd shows SNR< 1 in the northern parts of the CanESM2-LE data (Fig. 7 (c)) and in the Alpine region of the CRCM5-LE data (Fig. 7 (d)). This variable shows a strong linear relationship between LE mean and sd (Fig. A3 (e)–(h)). Regarding PR sum, RCM members vary most in regions with highest absolute $\alpha_1$ values and altitudes, but there is no clear dependence in GCM (Fig. A3 (i)–(l)). An east-west corridor of SNR values below 1 accompanies rather

low $\alpha_1$ values of PR sum (Fig. 8 (c)–(d), (g)–(h)).

In addition to future changes in the NAO responses ensemble means, there is also a change in the spatial distribution of the inter-member spread expressed as ensemble sd (subpanels (i)–(j) in Figs. 6–8).

To further investigate the inter-member spread, Fig. 10 illustrates the Pearson correlation coefficients $r$ between the NAO index and subset regions for nSAT mean or PR sum in GCM and RCM LEs separately. In these boxplots, the variability among the members, is illustrated by the boxsize, i.e. the inter-quartile distance. Both ensemble inter-member spreads generally envelope the ERA-I value (dashed line) of the given region, apart from GCM hist in Fig. 10 (b). This general finding does not change in the projected future climate: most boxes and whiskers keep their size, only GCM nSAT in the NE region is characterized by

a larger range in the future (significant at $p \leq 0.05$, using an F-test for comparison of variances). Some of the ensemble mean values exhibit a significant shift towards lower $r$ values in the future for both models for nSAT mean and PR sum (see text insertions CanESM2(hist, fut) and CRCM5(hist, fut)). An unpaired Mann-Whitney/U-test is applied here as the samples from hist and fut are seen as being drawn from different climates (since the null hypothesis of independence between hist and fut periods could not be rejected at $p \leq 0.05$ using a $\chi^2$-test).

### 3.2.2 Change of scales

Having analyzed GCM and RCM separately so far, the next step is to compare both ensembles. A $\chi^2$-test reveals that GCM and RCM samples of $r$ can be seen as significantly dependent in both time frames. The amount of variance explained by the NAO is generally higher in the ERA-I reference than in the RCM ensemble mean. The CRCM5-LE enhances the relationship showing higher $r$ and $\alpha_1$ values than the CanESM2-LE (see Fig. 10 for $r$ and Figs. 6–8 for $\alpha_1$). This enhancement by the

415 CRCM5 is notably independent of the driving data: for both variables, the CRCM5/ERA-I $r$ value (dotted lines in Fig. 10) is also found to be higher than the ERA-I value in most regions (dashed lines in Fig. 10). In all subset regions, the CRCM5/ERA-I $r$ value lies in the upper part (stronger correlations) of the CRCM5-LE ensemble values.

Figure 10 shows that mean $r$ values of RCM (grey filling) and GCM (white) members are significantly different in all subset regions for nSAT mean in both time horizons, but only in the NE and BY regions for PR sum; in the SE region, only weak

differences between GCM and RCM PR sum $r$ distributions are visible. In NE and BY regions this difference is expressed by higher $r$ values in RCM data, whereas in the SE region lower $r$ values are found in the RCM data (only for nSAT mean). Apart from PR sum in the NE region (both time horizons), no significant difference between the spread amplitudes of GCM and RCM is visible ($p \leq 0.05$, F-test). The inter-member spread of the correlation between NAO and response variables is not generally altered during the nesting process.

To evaluate the co-variability of CanESM2 and CRCM5 data in the subset regions, time series of the response variables originating from both data sources are correlated member-wise (see Fyfe et al., 2017, for a similar approach). As can be seen in Fig. 11, highest accordance on average is reached for nSAT mean in both periods, indicating that CanESM2-LE and CRCM5-LE show very similar temporal variability for this variable. The co-variability of GCM and RCM time series is weaker for PR sum (Fig. 11 (b)) and nSAT sd (Fig. 11 (c)) than for nSAT mean (Fig. 11 (a)) in both periods. Also, the inter-member spread is larger

for PR sum and nSAT sd than for nSAT mean. This finding suggests that there is a larger discrepancy in portraying PR sum and

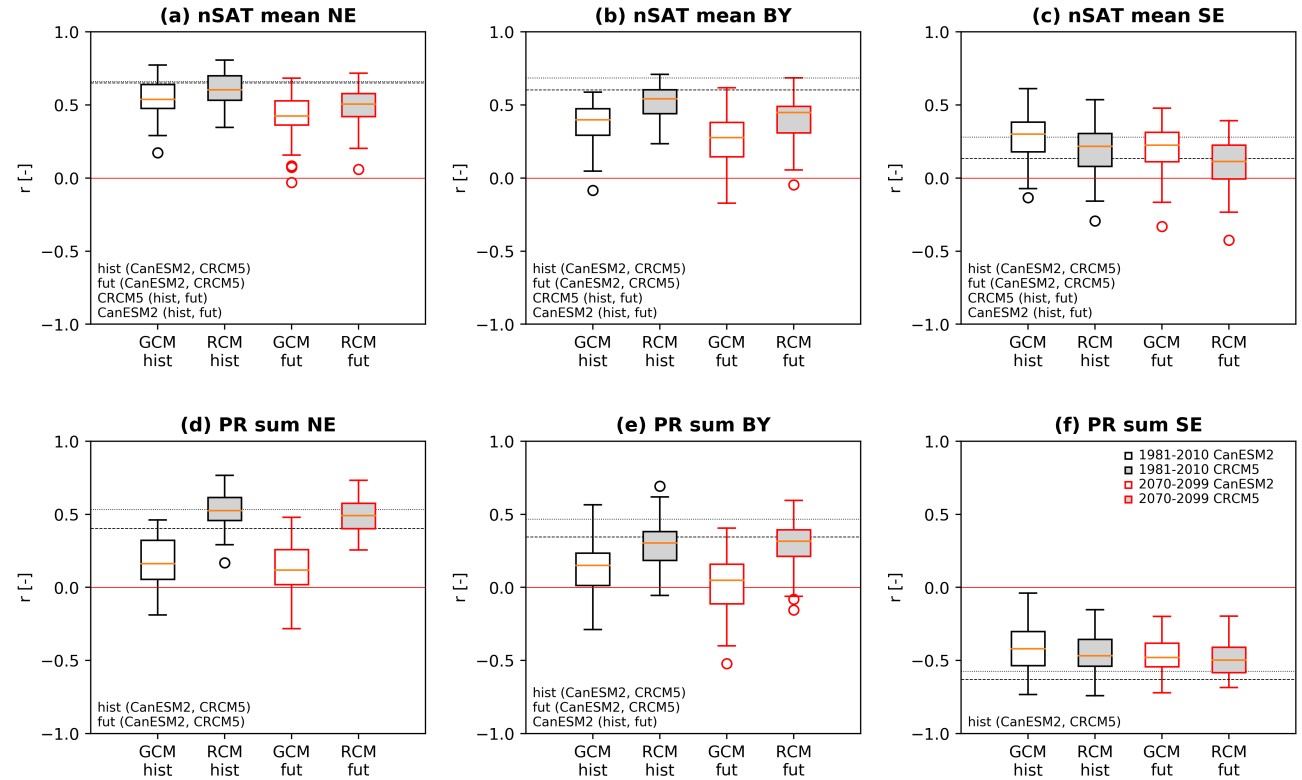

**Figure 10.** Boxplots of nSAT mean ((a)–(c)) and PR sum ((d)–(f)) showing Pearson correlation ($r$) with the NAO index of 50 CanESM2-LE (white filling) and CRCM5-LE (grey filling) realizations for three regions (NE, BY, SE) in historical (black outlines) and future (red outlines) time horizons. Dashed (dotted) horizontal lines indicate the ERA-I (CRCM5/ERA-I) value; text denotes combinations of which the differences are significant at $p \leq 0.05$ using an unpaired Mann-Whitney/U-test for the comparison between hist and fut periods and a paired Wilcoxon test for the comparison between CanESM2-LE and CRCM5-LE. Orange line in boxplots: median. For regions NE, BY, SE see Fig. 2.

nSAT sd in the RCM with respect to the GCM compared with nSAT mean, i.e. the RCM does not generally track the variability induced by the GCM for these variables. The correlations of nSAT mean and PR sum between CanESM2 and CRCM5 subset regions are in general significantly lower under future climate conditions compared with the historical ones, apart from nSAT mean in the BY region and PR sum in the SE region (see text in Fig. 11). For nSAT sd a significant shift of the distribution of $r$ towards higher values is visible, apart from nSAT sd in the BY region. All variables exhibit a future inter-member spread increase, but not all subset regions are affected (e.g., nSAT mean in BY or nSAT sd in SE, Fig. 11). This suggests that under future climate conditions a potential reduction of GCM–RCM co-variability needs to be considered, at least for PR sum and (weaker) for nSAT mean.

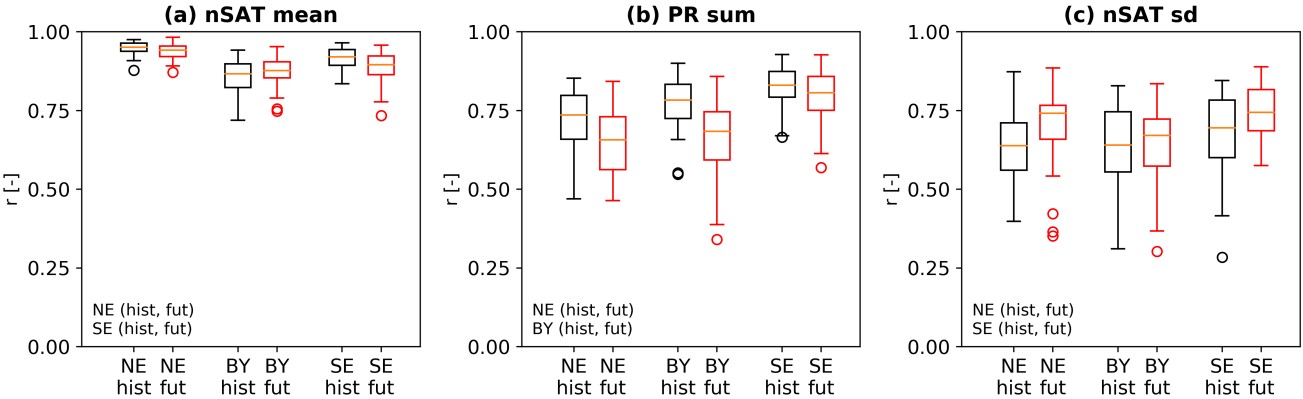

**Figure 11.** Temporal co-variability of CanESM2-LE and CRCM5-LE subset regions in all 50 members. Each boxplot represents 50 Pearson correlation coefficients of the time series of variables nSAT mean (a), PR sum (b) and nSAT sd (c) in the subset regions between CanESM2-LE members and the corresponding CRCM5-LE members. Time periods used for correlations: 1981–2010 (hist, black), 2070–2099 (fut, red). Text denotes combinations of which the differences are significant at $p \leq 0.05$ using an unpaired Mann-Whitney/U-test. Orange line in boxplots: median. For regions NE, BY, SE see Fig. 2.

## 4    Discussion

### 4.1    General performance of the model chain

The ClimEx climate data ensemble is able to reproduce an NAO-like pattern with realistic temporal and spatial characteristics over the North Atlantic and corresponding response patterns in Central Europe. Ensemble mean information aggregates several realizations and so differences towards the single ERA-I realization are to be expected. However, results show that the ERA-I pattern may in general be seen as being "embedded" in the RCM or GCM inter-member spread, implying that GCM, RCM and the reference data share comparable climate statistics.

Regarding temperature, Europe is commonly seen as divided into a region with positive NAO–response correlations in the north and negative correlations in the south (see e.g., Woollings et al., 2015). The first is found in the here presented results, the latter is not clearly visible in the chosen domain. nSAT sd is correlated negatively with the NAO, pointing towards less temperature variability in winters with positive NAO phases, and a higher variability during negative phases. Correlations of PR sums and NAO are in accordance with the prevalence of large-scale (frontal) precipitation in winter which might be affected if the large-scale circulation is altered due to the NAO.

The strong SLP gradient under neutral NAO conditions over the North Atlantic noted in the CanESM2-LE suggests an overestimation of the local atmospheric circulation with too strong westerlies. Similar model biases are widely reported (see e.g., Ruprich-Robert and Cassou, 2015; Stephenson et al., 2006; Reintges et al., 2017; Ulbrich et al., 2008). Since the NAO index was obtained from raw SLP data, it contains the contribution of the NAO, but possibly also of micro-climatic noise or other teleconnection patterns like the East Atlantic (EA) and the Scandinavian Pattern (SCA) which interact with the NAO and exert

a notable control on the North Atlantic SLP gradient according to Moore et al. (2013). These authors investigated the contributions of the North Atlantic teleconnections NAO, EA and SCA in reanalysis data by separating them with empirical orthogonal functions. The authors found that the "pure"' NAO accounts for about one third of winter SLP variability, and the second and third leading modes for roughly 20 % and 15 %, respectively (see also Comas-Bru and McDermott, 2014). Thus the results shown here may be seen as representing the superposition of these atmospheric modes.

The fidelity of NAO responses further depends on two aspects: (i) the goodness of representation of the large-scale NAO-related SLP pattern in CEUR and (ii) the strength of the linear relationship between the NAO and the response variables. The first point is addressed by a good representation of the SLP pattern in RCM data (see Fig. 5). The second point may be targeted by a combination of the strength of the responses (correlations $r$) and the response values themselves ($\alpha_1$): NAO responses in the CEUR domain of all data sets are most reliable in regions where a strong linear relationship between the NAO and the response variable may be assumed. This may be the case if the correlation coefficient between the NAO index and the variable time series on the given grid cells is significantly different from zero. However, linearity does not apply under all conditions. For example, particularly strong negative NAO phases with low-ice conditions in the Arctic coincide with cooling in Europe that is weaker than expected from a linear relationship due to an accompanying warming over Siberia (Screen, 2017). Low correlation values may also suggest that climate variability in these regions is only to a small fraction influenced by the NAO in this data set and period under consideration. In these cases, the NAO as expressed by the North Atlantic SLP gradient in this study is not the most important contributor and the noise, $\varepsilon_Y$ in Eq. (4), is dominant.

Historical changes induced by the NAO ($\alpha_1$, all data sources) are generally in accordance with observed composite anomalies (see also Fig. A4), but most so in regions with significant correlations. Thus, the future change of nSAT and PR per unit index change is most valid where correlations are high and where the NAO related responses emerge from internal variability (i.e., SNR > 1). Of course, $\alpha_1$ and composite maps are not identical, as on the one hand the average index value that accompanies nSAT and PR anomalies is not the same ($\pm 1$ for $\alpha_1$, but $+1.498$ and $-2.103$ for ERA-I composites, see Fig. 9). On the other hand, $\alpha_1$ estimates a change which is singularly generated by the NAO index in a linear relationship, while composite maps originate from raw data which might include further influences.

## 4.2 Nesting approach

NAO response patterns are similar within the CanESM2-LE and CRCM5-LE, but some deviations remain due to differences in model parameterization and spatial resolution. Another possible explanation could be that the control exerted by CanESM2 through the CRCM5 lateral boundary conditions is insufficient, but this is unlikely given the relatively small CRCM5 domain implying stronger lateral boundary conditions control (Leduc and Laprise, 2009), in addition to the strong spectral nudging of large scales that was applied in the production of the CRCM5-LE (Leduc et al., 2019). Also, the large-scale SLP pattern over CEUR shows no large errors in the CRCM5-LE with respect to its driving data sources (see Fig. 5) and temporal correlation of GCM and RCM time series are generally high. Nevertheless, the influence of the lateral boundary conditions regarding SLP appears to vary over the CRCM5 domain, being a bit weaker in the southern part. It is worth noting that this feature is less

pronounced when CRCM5 is driven by ERA-I as compared with CanESM2, highlighting the importance to investigate further the interactions between global atmospheric circulation, surface forcings (e.g., topography and land-sea contrasts) and local feedbacks.

The CRCM5 reproduces the response structures much finer than the CanESM2 and adds some robust high resolution geographical features which are clearly visible within the ensemble mean.

Apart from the coarser pattern resolution, there is also a shift in the spatial climate patterns in the CanESM2-LE within the CEUR domain with respect to ERA-I data which is not found in the CRCM5-LE: for example, typical continental climate features, such as high nSAT variability (as indicated by Fig. 6), are shifted southwards in the CanESM2-LE with respect to

500 CRCM5-LE data (or ERA-I). This shift may be explained by the fact that due to coarser spatial resolution the GCM topography shows land grid cells where the Mediterranean or the Baltic Sea extend in ERA-I and CRCM5; thus, in the GCM, the continent Europe also occupies a region which is sea in ERA-I. Assuming that the land–sea distribution affects the climate evolution, the GCM also experiences a geographical shift of climatic characteristics (such as continental properties) compared with the ERA-I and RCM data within the study domain. Another example is the dividing line for NAO–PR sum relations (see Fig. 8)

which shows a displacement in the GCM compared with the RCM. This displacement is related to the GCM orography which deviates due to the coarser spatial resolution in shape, position and height from the RCM orography. These findings suggest that similar responses of GCM and RCM to the NAO may not be visible at the same geographical location (i.e., coordinates), but under similar geographical conditions (exposition, altitude, distance to sea). Continuing this thought, the RCM reproducing the spatial climatic patterns in the "correct" location is another expression of the RCM added value for regional or local scale

analyses. However, for general statements on this issue, analyses on a larger domain would be necessary.

On the regional scale, the correlations in the CRCM5 are significantly stronger in several regions than in the CanESM2 (Fig. 6–8). These differences are not evened out by spatial aggregation. Thus, in the CRCM5-LE, more variance is explained by the NAO (i.e., by large-scale circulation) than in the CanESM2-LE. Explained variance is also higher in the single realizations of ERA-I and CRCM5/ERA-I than in the ensemble mean of GCM and RCM.

### 4.3 Internal Variability

In general, the 50 NAO signals from the atmospheric "inflow" as given by the GCM boundary conditions are correctly translated into 50 regional responses of the RCM regarding the range of internal variability.

The large ensemble internal variability favours a smoothing of structures in the ensemble mean. Nevertheless, as the ensem-

520 ble mean (GCM and RCM) reproduces patterns very similar to the observed ones, the atmospheric dynamics behind can be regarded as correctly reproduced in all members.

When looking at spatially explicit ensemble sd maps (see Figs. 6–8 and A3), the RCM-LE exhibits higher ensemble sd values than the GCM. This is in accordance with Giorgi et al. (2009) who stated that internal variability at finer scales tends to be larger compared with coarser scales. However, the amplitude of the inter-member spread of NAO–response correlations in

the aggregated RCM and GCM subset regions is similar. Thus, the range of internal variability regarding the strength of the

NAO–response relationship is transferred during nesting and the CRCM5 added internal variability (Leduc et al., 2019) does not significantly alter it. However, the ensemble values are shifted towards significantly higher $r$ values in the RCM compared with the GCM in both time frames, but not in the SE region.

## 4.4 Climate Change

The results show that historical and projected future climate statistics deviate such that the comparison of relationships in both periods remains difficult: the NAO pattern changes, NAO index variability and nSAT and PR responses are reduced in the future climate simulation. Also the uncertainty range of the signals does not change significantly in the future horizon. With the here presented results, it can be argued that the internal variability of more complex parameters (such as the NAO–response relationship quantified via Pearson correlation) shows no significant changes between historical and future periods. When looking at the spatial distribution of $\alpha_1$ ensemble sd however, several regions show slight future increases or decreases which are not necessarily consistent between GCM and RCM.

It has to be added that this study evaluated two 30-year blocks rather than continuous time series, treating the NAO–response relationship and the inter-member spread as stationary during these blocks such that the inter-member spread of both periods represents generalized conditions for 1981–2010 and 2070–2099.

According to Comas-Bru and McDermott (2014), potential non-stationarity in NAO–response relationships can at least partly be attributed to influences of the EA/SCA patterns on the NAO, and especially the geographical position of the North Atlantic SLP gradient.

The relative prevalence of negative index phases in the future period occurs in correspondence to a generally strengthened high pressure ridge over the North Atlantic and especially Greenland (see Fig. 4 (g)). The latter feature is supposed to be related with the emergence of negative index phases (Hanna et al., 2015; Woollings et al., 2010; Gillett and Fyfe, 2013; Cattiaux et al., 2013; Screen, 2017). Another relationship ties the emergence of negative NAO index phases to reduced sea ice extents: Warner (2018) found that particularly October sea ice extent over the Barents/Kara Sea is positively correlated with the NAO in that it leads to strengthened IL and AH. Consequently, a reduced sea ice extent is associated with negative NAO phases, but this relationship is not simply linear (Warner, 2018). For example, Screen (2017) notes that negative NAO events tend to be stronger during winters with low sea ice extents. The NAO-sea ice relationship may follow from sea ice effects on the stratospheric polar vortex or from tropospheric Arctic amplification which reduces the meridional temperature gradient leading to a weakened, more wavy jetstream in the mid-latitudes (Warner, 2018). The CanESM2-LE is known to show a low bias regarding Arctic sea ice in all seasons compared with observations (Kushner et al., 2018), but it follows quite correctly the observed downward trend (Kirchmeier-Young et al., 2017) and leads to a clear reduction of sea ice in the 2070–2099 horizon compared with 1981–2010 in the entire Arctic and also the Barents/Kara Sea as is verified with the CanESM2 variable "sea ice concentration" for this study (not shown).

An increasing frequency (relative to positive phases) of negative NAO events as noted in the presented results favours more cold and harsh winters in theory due to the advection of continental Eurasian air masses (Screen, 2017) which is in great contradic-

560 tion to projected future background conditions (warmer, moister, see Leduc et al., 2019) that would rather, likewise following from theory, accompany positive phases. On the other hand, the response to NAO impulses is clearly reduced for nSAT mean, PR sum and nSAT sd. A coherent explication for this discrepancy might be that as correlations weaken, the Eurasian influence (advection of cold, dry airmasses) during negative phases may be repressed or weaker in its occurrence than now or, as indicated by Screen (2017), is actually increasing warmer air mass advection. As less nSAT and PR variance is explained by the 565 NAO in the future climate projections than in the historical period, the influence of this climate mode on CEUR climate may be seen as potentially reduced.

## 5 Conclusions

In this study, a RCM single-model initial condition large ensemble is analyzed with a special focus on the downscaled responses 570 to a teleconnection, the NAO, that is present in the driving data. For proper assessment, the driving GCM ensemble is also included in the study. With regard to the key questions raised in the introduction, it can be stated that:

(a) The ClimEx RCM-LE and its driving GCM-LE are able to depict a robust NAO pattern under current forcing conditions. Each member represents a distinct climate evolution while sharing comparable statistics with all other 49 realizations and producing NAO and response patterns that are more robust than patterns of individual realizations. The ensemble also 575 shows climate statistics that are comparable with the reference time series and patterns. The clearly visible connection of the NAO with nSAT mean and PR sum follows well-known patterns and allows to derive robust information on the influence of the NAO on nSAT variability (nSAT sd).

(b) The RCM is able to reproduce the large-scale SLP pattern and realistic response patterns in the analyzed domain. Clearly more topographic features are visible in the CRCM5-LE than in the CanESM2-LE which suggests added value by the 580 RCM regarding the evaluation of small-scale NAO impacts. Deviations of nSAT and PR responses between members vary spatially within the domain and are found mostly in regions with strongest NAO responses.

(c) Internal variability of the NAO pattern is expressed very well within the 50 member single-model ensembles, and easily spans the observations regarding various indicators. The range of NAO responses is represented consistently between the driving GCM and the nested RCM. The spread is shifted towards stronger NAO–nSAT/PR relations in the RCM 585 compared with the GCM in both time horizons.

(d) Concerning climate change, several changes go hand in hand: the winter index variability is reduced, the overall winter variability of nSAT and PR and also the fraction of NAO-explained nSAT is reduced, the relationship between NAO and response variables is weakened, and the co-variability of CanESM2 and CRCM5 subset regions for all variables is reduced.

While these results are especially valid for the analyzed GCM-RCM combination, they allow drawing some general conclusions. The results strengthen the validity of this GCM-RCM combination for further applications, as important large-scale

teleconnections only present in the GCM propagate properly to the fine-scale dynamics in the RCM. The RCM does not alter the spread of driving GCM data which is a valuable information for impact modelling with a focus on internal variability. The results also stress the importance of single-model ensembles for evaluating and estimating internal variability since single
realizations show considerable variations among themselves and also deviations from the ensemble mean. So the ensemble mean and the ensemble spread together are needed for robust assessment of climate modes and whether a given model is able to reproduce the phenomenon of interest.

*Data availability.* Data used in this study may be retrieved from the following sources:
CanESM2-LE data is available via https://open.canada.ca/data/en/dataset/aa7b6823-fd1e-49ff-a6fb-68076a4a477c.
CRCM5-LE data can be retrieved at https://climex-data.srv.lrz.de/Public/
ERA-Interim Reanalysis data set was obtained at https://apps.ecmwf.int/datasets/data/interim-full-daily/levtype=sfc/

**Appendix A**

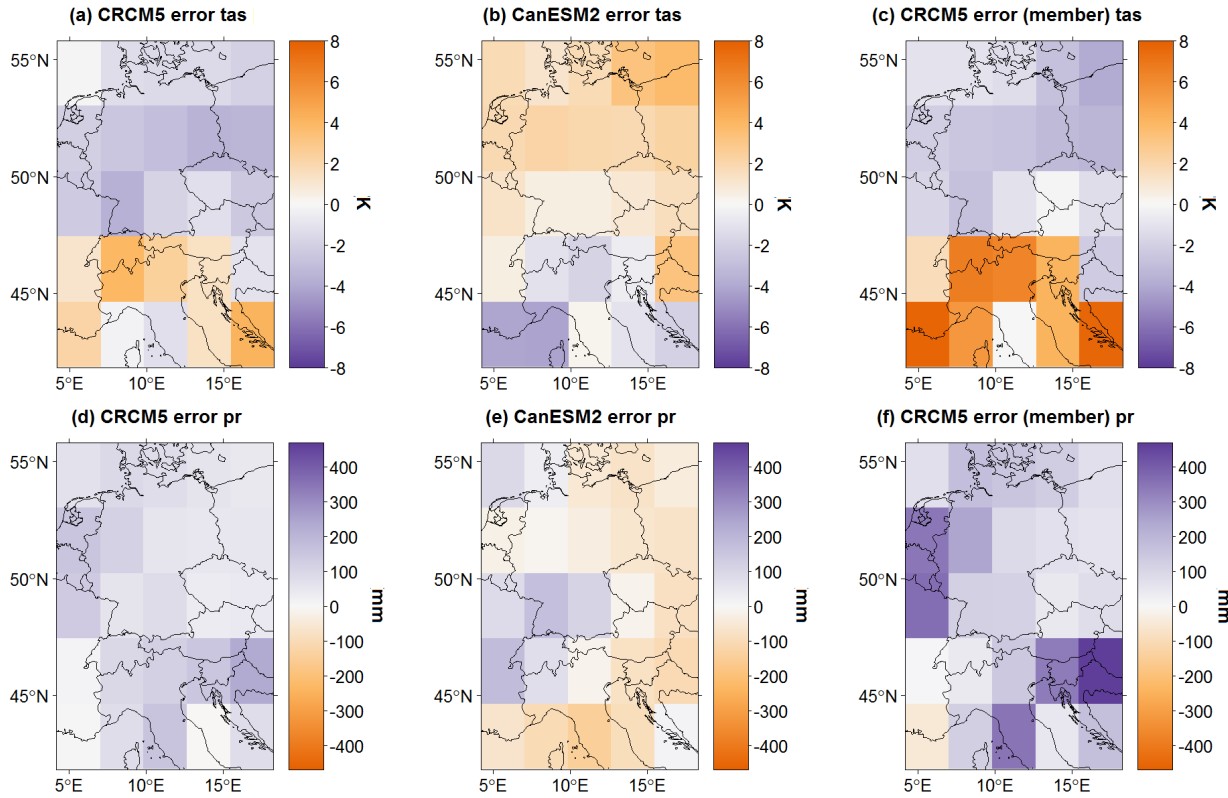

**Figure A1.** Model deviation for the 1981–2010 winter mean nSAT mean ((a)–(c)) and winter mean PR sum ((d)–(f)) in GCM resolution (2.8°). First column: error of CRCM5 under ERA-I boundary conditions (difference between CRCM5/ERA-I and ERA-I). Second column: error of CanESM2-LE towards ERA-I data (ensemble mean of differences between CanESM2-LE members and ERA-I). Third column: CRCM5 error under CanESM2-LE boundary conditions (ensemble mean of differences between CRCM5 members and corresponding CanESM2 members).

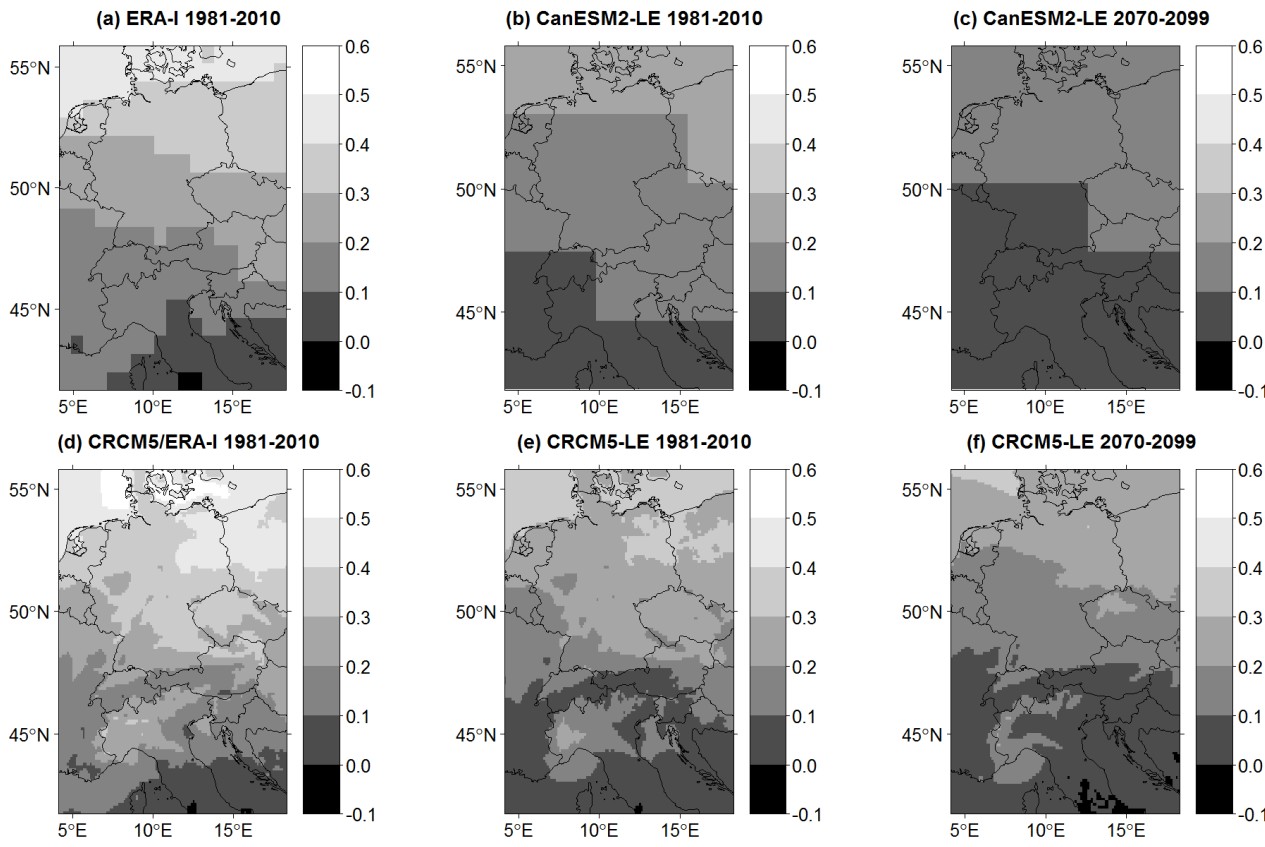

**Figure A2.** Ratio of nSAT $\alpha_1$ and winter mean daily standard deviation of nSAT for driving data ((a)–(c)) and RCM data ((d)–(f)) during historical ((a)–(b), (d)–(e)) and future ((c), (f)) conditions. The panels show the proportion of nSAT $\alpha_1$ in winter mean daily standard deviation of nSAT.

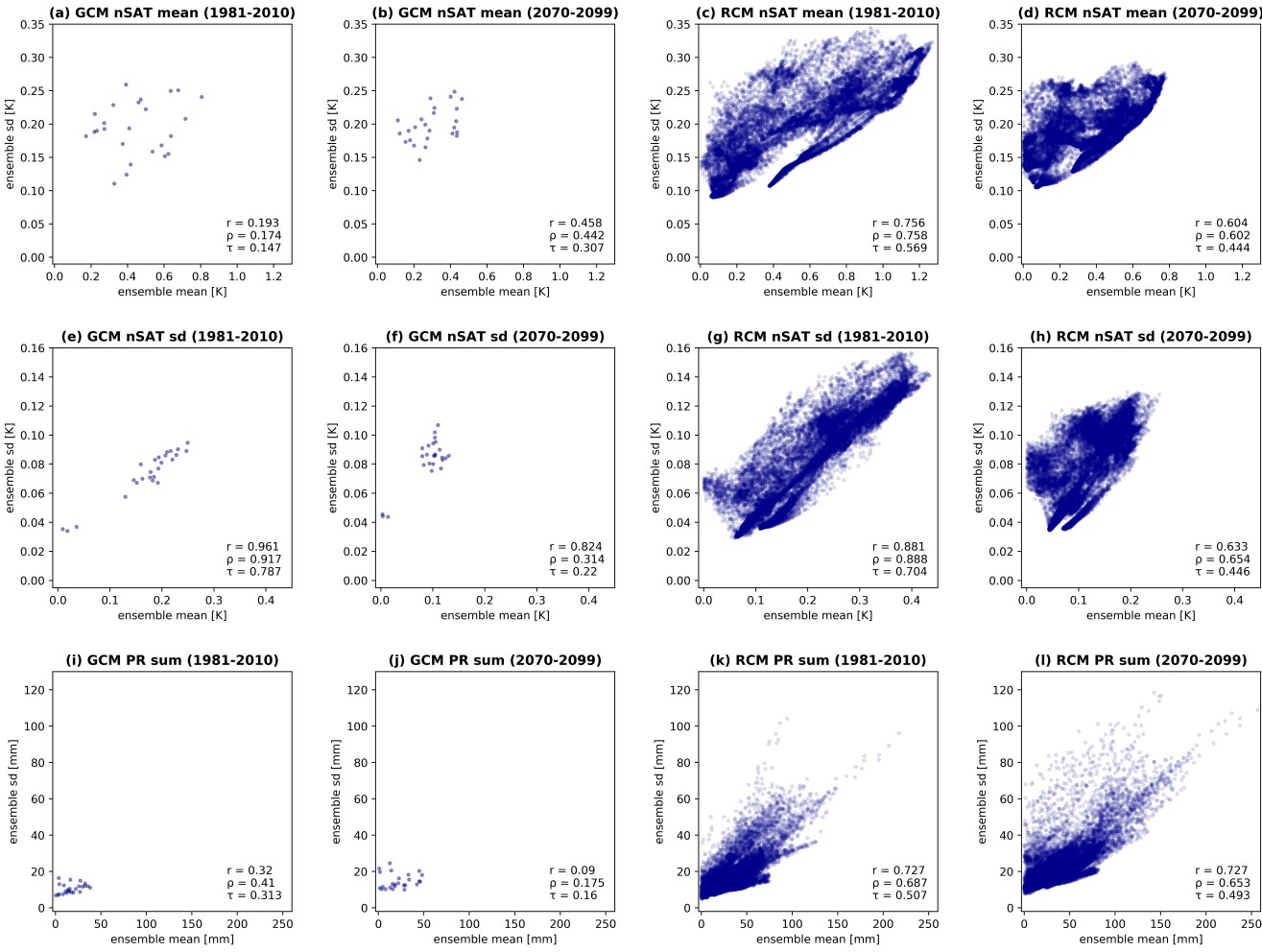

**Figure A3.** Relationship between LE mean and sd values of CanESM2-LE (GCM) and CRCM5-LE (RCM) for variables nSAT mean (a)–(d), nSAT sd (e)–(h), PR sum (i)–(l) for hist and fut periods. Lower right corner: $r$ – Pearson correlation coefficient, $\rho$ – Spearman rank correlation coefficient, $\tau$ – Kendall's Tau.

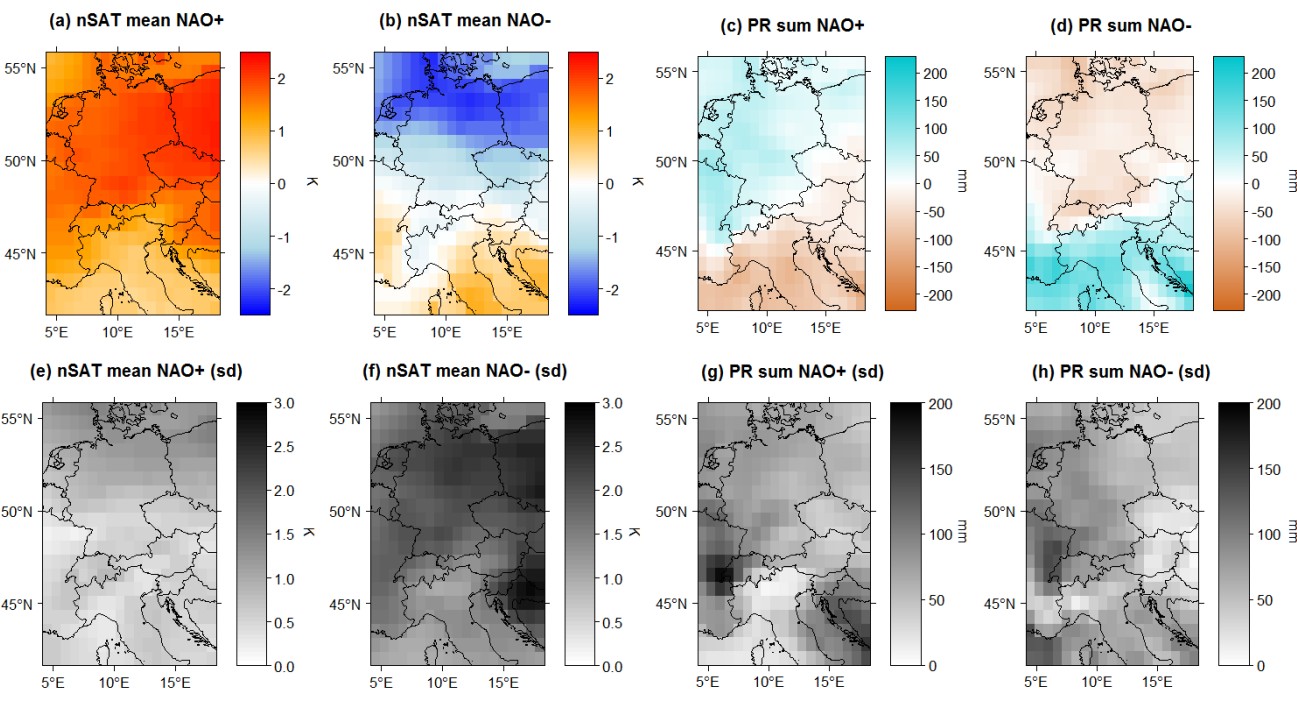

**Figure A4.** ERA-I anomalies from the long-term mean of nSAT mean in [K] and PR sum in [mm] in NAO positive (1989, 1990, 1994, 1995) and negative (1996, 2001, 2010) winters. Mean index value for positive (negative) NAO phases is $+1.498$ $(-2.103)$.

*Author contributions.* This study was conceptualized by AB under supervision of RL. Formal analysis, visualization of results and writing of the original draft was performed by AB. All authors contributed to the interpretation of the findings and revision of the paper.

*Competing interests.* The authors declare that they have no competing interests.

*Acknowledgements.* The authors would like to thank two anonymous reviewers for their valuable input which helped to considerably improve this work. The production of ClimEx was funded within the ClimEx project by the Bavarian State Ministry for the Environment and
Consumer Protection. The CRCM5 was developed by the ESCER centre of Université du Québec à Montréal (UQAM; www.escer.uqam.ca) in collaboration with Environment and Climate Change Canada. We acknowledge Environment and Climate Change Canada's Canadian Centre for Climate Modelling and Analysis for executing and making available the CanESM2 Large Ensemble simulations used in this study, and the Canadian Sea Ice and Snow Evolution Network for proposing the simulations. Computations with the CRCM5 for the ClimEx project were made on the SuperMUC supercomputer at Leibniz Supercomputing Centre (LRZ) of the Bavarian Academy of Sciences and
Humanities. The operation of this supercomputer is funded via the Gauss Centre for Supercomputing (GCS) by the German Federal Ministry of Education and Research and the Bavarian State Ministry of Education, Science and the Arts.

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
