# Peer review of "Using a nested single-model large ensemble to assess the internal variability of the North Atlantic Oscillation and its climatic implications for Central Europe"

_Earth System Dynamics, 2019_

## Referee Comment (RC1) · Anonymous Referee #1 · 28 Oct 2019

**Review of "Using a nested single-model large ensemble to assess the internal variability of the North Atlantic Oscillation and its climate implications for Central Europe"**
**by A. Bohnisch et al.**

The manuscript presents an analysis of changes in the North Atlantic Oscillation (NAO) under a global warming scenario, using two 50-member model ensembles: an ensemble of a global general circulation model, and an ensemble of a high-resolution nested regional climate model. The large ensemble size allows the authors to not only analyze the change in the mean NAO, but also in its variability. The authors also show the impact of the NAO and its variability on European climate.

This manuscript presents an interesting study that combines two state-of-the-art techniques: very large ensembles to estimate transient change of internal climate variability, and a high-resolution regional climate model. The results are novel and relevant. However, I think there is some unused potential in the study that should be harvested (see my specific comments below), and the presentation of the results could be improved.

I think the manuscript is a good fit for Earth System Dynamics and should be published. That being said, the manuscript requires structural clarification that warrants a major rewrite, so that I recommend **major revisions** to the manuscript before publication can be considered.

Specific Comments:
l. 2 *"...(NAO) which is a relevant index for quantifying natural variability..."* I find this sentence to be ambiguous. What is a relevant index? As it stands now, it seems to be the mass advection triggered by the NAO. I suspect that the authors mean the NAO itself. If this is the case, I think this ambiguity can be avoided by introducing a comma between "(NAO)" and "which".

l. 4 Is the link to the CORDEX project really needed in the abstract? Please consider removing it.

ll. 4-6 This sentence is missing the crucial information that the "LE" model is a nested regional climate model.

l. 9 I do not see how the word "strength" in brackets on its own relates to "pearson correlation coefficient". Please re-evaluate whether "strength" adds any meaning at this point.

l. 11 What is a "correct response" to NAO forcing? How is that defined? If it's based on the global model simulation (which I assume it is) I am not sure that "correct" is the right word here.

l. 12 Which relationships weaken in the future? Also, what does it mean and why is it important to show that the amplitude of inter-member spread does not change with anthropogenic forcing?

Introduction I find the introduction confusing and hard to follow. For example, the first paragraph (ll. 16-22) seems to set the reader up for a follwing paragraph on ensembles, but instead global and regional climate and the NAO are introduced in the next paragraph (ll. 23-32). For another example, the reader expects a discussion of advantages and limitations of different methods to quantify the NAO index after paragraph 3 (ll. 33-37), but paragraph 4 (ll. 38-42) introduces the reader to NAO impacts and its interactions with other modes of climate variability. Moreover, this interaction with other modes of variability is in my opinion not important to the study presented in this manuscript. Both the missing storyline and the lack of focus on the important information for this study are an issue throughout the entire introducion. I therefore recommend that the authors rewrite the introduction with particular attention to the storyline and focusing on the important information, so that the reader can follow the reasoning more easily.

l. 38 There is no mention of a positive state before. I believe the authors are referring to a positive NAO state, but that needs to be made explicit, especially so at the very beginning of a paragraph.

ll. 75-76 Please consider omitting the "table of contents" at the end of the introduction. It does not add to the story and takes focus off the nice overview of key questions that will be addressed in the paper just before.

ll. 80-86        I think somewhere here it would be important to mention which region the regional model covers. Please consider adding this crucial information.

l. 91        The implications of this sentence would be much easier to understand, if the CORDEX ensemble was introduced very briefly. Please consider adding a few words on what the CORDEX ensemble is, as well as a literature reference.

ll. 95-96        I am not sure that I agree with the conclusion, that "the most important" modes of climate variability are captured by the ClimEx model, as this conclusion is here based on a comparison to another model ensemble. I agree that it is reasonable to assume from this comparison that the ClimEx model produces reasonable climate variability, but I do not think such a comparison warrants a judgment on which mode of variability is important or not. Please consider rephrasing.

ll. 100-103        The most commonly used acronyms for sea level pressure and surface air temperature are SLP and SAT, respectively. Why did the authors decide to use different abbreviations? This is not a huge issue, but interrupts the flow when reading. Also, t2m and tas are usually not the same in model output. The manuscript would benefit from clarification as to which of the two is used in this study – this is currently not clear.

l. 120        The text says that there are two regions of interest, while table 2 specifies seven regions and the remaining manuscript references those seven regions. I suggest omitting the "two regions" phrase, as it is more confusing than helpful at this point.

l. 140        The authors use past tense to describe the present study here, and this appears to be the dominant choice of tense. Elsewhere, however, present tense is used (e.g. l. 120 "...there are two separated regions..."). This inconsistency can be found throughout the entire manuscript. To improve readability, I suggest the authors decide on one tense and stick to it throughout the manuscript.

ll. 141-142        The word "representative" is lacking a reference here. The 30-year time horizon leads to an NAO distribution that is representative of what? Please elaborate briefly.

ll. 144-145        This is an important caveat. I like that this is mentioned here, but missed it in the discussion section. I suggest taking it up again there to make sure this (perfectly acceptable) limitation of the study can be appreciated.

ll. 150-154        I think this bit would be easier to understand if the order of the phrases was altered to first explain why March can be included and then say that DJFM is used for winter. Please consider making this change.

ll. 159        I suggest refraining from the statement that a station-based NAO index is "easy" to interpret – its reference is arbitrary (easy for whom?) and it is not a very scientific expression. Please rephrase.

ll. 189-195        This section appears to already present results. Please consider moving it to the results section.

l.200        In lines 97-98, the authors define REF as the ERA-Interim data set. Here, REF appears to refer to the NAO index within the ERA-I data set. Please define REF only once and unambiguously.

ll. 205-206        I am not sure I agree that figure 1a shows that REF (the blue bars) lies "comfortably" within the ensemble spread (grey & red). Particularly negative extremes, but to some degree also positive ones, seem to be underrepresented in the model. Can you please comment on this and possible implications for this study?

l. 214        *"…original data in**to** three subsets..."*

l. 214        Please consider changing "indifferent" to "neutral" or "average" here and throughout the document.

l. 214        Are the "average psl conditions" referenced here the same as the "MSLP mean" in figure 2? If so, I highly recommend using coherent names (i.e. "mean" or "average" in both cases) to avoid confusion. I had to read this paragraph several times before I understood it.

ll. 216-217    Which difference is referenced here? Also, what do over- and underestimation refer to? If this is based on a comparison of figs. 2a and d, I cannot follow the argumentation – actually, it appears to me that the model overestimates mean SLP over the North Sea and underestimates SLP over Greenland. Can you please clarify?

l. 218    *"...phases also show less pronounced..."* Weren't the anomalies more pronounced in the model than in REF for the mean state? If so, please omit the "also".

ll. 239-240    *"…the spatial patterns of ERA-I and CRCM5/ERA-I differ* **more** *strongly than in Fig. 3,..."*

l. 241    What is the reference for the "more humid conditions"? The lack of a reference for relative statements is an issue that needs addressing throughout the manuscript.

l. 256    The NAO explains less variance than what?

l. 257    tas std decreases less than what?

l. 259    While I am sure the inconsiderable change of spatial patterns compares the historical to the projected period, I think it would help to give this information here again.

ll. 259-260    Could you please give a figure reference for the claims made here?

l. 264    Is there a particular area for which the transfer of internal variability from GCM to RCM is assessed?

l. 277    If large tas deviations do not correspond to high or low α, what do they correspond to?

l. 284    I find the presentation of this reference to figs. 3, 4 and 5, h & i ambiguous. Do you refer to panels h & i of all those plots, or just 5?

ll. 301-302    This sentence is difficult to understand due to the many parentheses and different references therein. I highly recommend splitting this sentence in at least two.

l. 306    I think the "matching subset region time series" warrant a more detailed explanation. As it stands, I am not sure what these are and how to interpret them. As a result I cannot follow the text. Please introduce this metric at least shortly.

ll. 308-309    I am not sure I fully agree with this statement. While correlations indeed appear to be generally lower for pr sum (fig. 8b), 1/3 regions for tas mean (fig. 8a) and 2/3 regions for tas std (fig. 8c) show an increase towards the later period. I think the manuscript could benefit from a more detailed discussion here.

ll. 309-310    I do not quite understand the last sentence of the "results" section. As a result, I struggle to see what its consequences are. I recommend adding some more explanation here, as this might be a crucial point.

ll. 314-315    What does it tell us that one realization shows a good correlation to REF? Why are the two so highly correlated? I am not sure why this is mentioned here. As in the introduction, this (apparently) irrelevant information might cause the reader to loose track of what is important. Please consider omitting this sentence or, if you deem it relevant enough, elaborate to illustrate its relevance.

l. 316    It is not clear about which strong psl gradient the authors are writing here.

ll. 318-319    NAO+ and NAO- are weaker within CanESM2-LE than which reference?

l. 320    The very limited sample size of n=7 (or rather n=3 and n=4) in REF is an important issue that is worrisome. It should be discussed further! How robust are the results presented here? What could maybe be learned about observations from the model?

l. 326        At this point, I somewhat expected a discussion on the influence of other teleconnection patterns. I think the authors should at least provide some indication (from the literature) about how large these teleconnections' influence on this study can be expected to be.

l. 335        The latter is not as clear in the chose domain as what?

ll. 338-339        I think the observation is missing a reference in this sentence: Is it NAO+ or NAO-? And are these observations derived from reanalysis or the literature or a model? As it stands, this is quite ambiguous.

l. 350        Omit the comma between "region" and "which".

ll. 352-353        This is an intriguing thought. What are its consequences/implications? Please consider to elaborate a bit.

l. 361        What does it mean for the findings presented here that the GCM overestimates T and pr? Does this limit the conclusions that can be drawn?

l. 367        Since the patterns are "only" very similar, I find the statement "atmospheric dynamics are correctly implemented" a bit too strong. Please consider rephrasing to, e.g., "...can be regarded as correctly implemented".

l. 378        As stated before (comment lines 205-206), I do not agree that the observations lie comfortably within the model spread, so I also have an issue with the statement "...the same climate statistics". Please either explain where I went wrong or rephrase.

ll. 382-383        Maybe rephrase to *"...with highest change in CRCM5-LE, but not* **necessarily** *in CasESM2-LE."*?

l. 391        Less tas and pr variation is explained by NAO than by what?

Conclusions        I think the reference to the questions raised in the introduction could be made clearer. While the references are there, I think it would make this part clearer if it was structured in bullet points, like the questions raised in the introduction. Please consider making this change.

ll. 397-399        This is a long sentence that is hard to understand because it takes up two different points. Please consider splitting the sentence in two.

l. 404        I find the word "proves" very strong. I agree that the clearly visible topographic features are nice to look at and encouraging for the model presented here, but I disagree with the notion that the mere notice of more pronounced topographic features "proves" the added value of anything. High resolution does not always equal added value. Please rephrase.

Fig. 2 caption        *"(g)-(i): 2070-2099 changes* **with respect to** *1981-2010"*

Figs. 3-5 caption        What are the correlations show in blue isolines? What is correlated to what? Also, this is a confusing figure, partly due to the ambiguous headers for the subpanels (which are identical for, e.g., c and g). Please think about a more intuitive way to convey this very interesting information.

Fig. 6        Some of the indices named in the upper left corner have slightly different names than those found on the x-axis. It could help the clarity of the (otherwise very nice and interesting!) figure if those names were the same. Please consider changing the figure accordingly.

Fig. 8        Please explain a, b and c in the caption. Also, I do not quite understand what is displayed. What is a "similarity of matching regions"?

Fig. A2 caption        Please explain the subpanels in the caption.

---

## Referee Comment (RC2) · Anonymous Referee #2 · 6 Nov 2019

In this study a regional climate model (CRCM5) is employed to dynamically downscale a single global climate model (CanESM2) large ensemble of climate change simulations to investigate the nature of downscaled responses to the modeled North Atlantic Oscillation (NAO) and its influence on future European climate. By employing a large ensemble, the authors are able to evaluate future downscaled responses associated NAO inter-annual variability in addition to mean changes. The authors set out four key questions related to, documenting the properties and fidelity of the modeled NAO in both the GCM and RCM; the associated screen temperature and precipitation responses in both models; and how such properties change under future external forcings (following the future CMIP5 pathway RCP8.5).

This is an interesting paper and ultimately worthy of publication. The authors present the problem from the perspective of downscaling teleconnections that exist in the driving data (ie the NAO). This is a subtle but critically important shift in focus for the dynamical downscaling community. The proper communication of teleconnection patterns/relationships from driving data to the RCM is essential for credible downscaled results. The use of a large GM/RCM ensemble pair positions the authors to say something definitive about this problem and offer guidance to the community.

The four key questions represent a clear and sensible plan for the paper. However, I found it difficult at times to cleanly connect a particular analysis performed by the authors with an answer to some of these questions. Specifically, I do not think that the authors addressed the first part of their question 3, "Do GCM NAO impulses propagate correctly into the RCM realizations" (l. 71). Perhaps a better way of stating this is, does the RCM faithfully represent the NAO pattern present in the driving data? This is a critical question in the authors' "model chain" (l. 65) that needs to be addressed before one moves on to evaluate the NAO responses. That is, if the large-scale NAO pattern is not faithfully represented in the RCM domain in some location, the downscaled responses in that location would be less credible. The increased resolution and potentially improved physical processes present in the RCM themselves cannot correct the large-scale NAO pattern within the RCM domain. As the authors discuss, the NAO pattern is governed by "planetary wavebreaking in the polar front" (Benedict et al., 2004), which is intern influenced by external factors such as sea-ice, snow cover, sea-surface temperatures, ENSO, stratospheric circulation variability, solar variability, volcanic eruptions and the Quasi-Biennial Oscillation (eg Hall et al. 2014 https://doi.org/10.1002/joc.4121).

Given that the European domain is relatively small, and the experimental design employs spectral nudging in the RCM, the NAO pattern, and its interannual variability,

should on balance be reasonably represented in the RCM. For the authors' stated plan, however, this needs to be verified. Given that the authors employ a large ensemble in their study, they are in the unique position to definitively address this issue and provide an example to the community of the type of analysis that is required to support the credibility of downscaled results in such complex problems. It is my recommendation that, prior to publication, the manuscript undergo major revision to address this issue and to improve its overall clarity. My detailed comments follow.

Major Comment:

RCM reproduction of NAO teleconnection in driving data

As part of the authors' model chain, it is essential to verify that the large-scale year-to-year variations of the NAO pattern in surface pressure are faithfully reproduced (each year) in CRCM5 when driven by both ERA-I and CanESM2. Inspired by Fig. 2, the sort of analysis required would be as follows:

- interpolate monthly-mean timeseries of sea-level pressure (SLP) in the driving dataset onto the RCM grid (such interpolation is already done for the driving-data winds used for spectral nudging). Call this field SLP_Drive.

- take the difference of the RCM and driving data monthly-mean SLP on the RCM grid

SLP_RCM - SLP_Drive,

and then smooth the result retaining large scales that are representative of the driving data resolution:

$D\_m(i,j,t,n) = [SLP\_RCM - SLP\_Drive]\_LRG$

Here, i,j are lateral spatial coordinates of the RCM grid, t is time in units of years, n is ensemble member, and the subscript m corresponds to month (1-12). The smoothing operation, represented by the operator [ ]_LRG, can be performed with the same double-cosine transform used for the spectral nudging.

- derive a normalized root-mean-square difference map for extended winter, over the two 30-year periods displayed in Fig 2, over all ensemble members:

RMS (i,j) = Ave_(m=12,1-3){Ave_n { SQRT[Ave_t {D_m(i,j,t,n)^2 }] / Var_Drive_m(i,j,n)}}

where, Ave_x is a simple averaging operators for the quantity x and Var_Drive_m is the variance in time of the driving data for each month and each ensemble member:

Var_Drive_m(i,j,n) = Ave_t {[ SLP_Drive_m (i,j,t,n) - Ave_t {SLP_Drive_m (i,j,t,n) } ]^2}.

Normalization by Var_Drive_m is important as it indicates the size of an rms difference relative to the interannual variability in the NAO pattern at that location. Such an RMS map would provide a sensible measure of the difference in the driving data and RCM SLP patterns associated with the NAO, which need to be faithfully reproduced in each year. If RMS « 1 at a given location, then the large-scale NAO pattern is well represented there and one can conclude that the downscaling is consistently being performed on the "correct" large-scale flow. The larger RMS is, towards O(1) values, the more suspect the downscaled responses are at that location (ie a large-scale flow disconnected from the NAO in the driving data was being downscaled in these regions). One should also do a significance test and indicate this by, say, filling in contours by color in only those regions that are significant at the 5% level. Given the size of the GCM/RCM ensemble, this should be quite robust (ie much of the canvas should be colored) and definitive statements could be made.

This test would seem to be most well posed for the case of observational driving of the RCM (ie ERA-I driving of CRCM5 over the historical period 1981-2010). The large scales in that data are well observed and, because they came from the real system, they were influenced realistically by all processes and scales. Significant deviations in RMS(i,j) for ERA-I (ie RMS_ERA-I) would necessarily indicate a degradation of the NAO teleconnection in those regions of the CRCM5 domain. If regions of NAO deviation in RMS_ERA-I were consistent with regions of NAO deviation in RMS_CanESM2 (in the historical and even the future periods), then this would indicate a systematic

issue with the reproduction of the NAO pattern in the European domain in these locations and care should be taken in the interpretation of the downscaled responses in this, and possibly other RCM studies using the same domain.

Minor Comments:

l.2 - "natural variability". Later it seems, "internal variability" (l. 16) is used to refer to the same phenomenon. It would be helpful to be consistent throughout.

l. 5-6. "its transfer from the driving model CanESM2 into the driven model CRCM5." Perhaps better wording might be "its representation in the driven model CRCM5 relative to the driving model CanESM2."

l.11 "(b) impulses from the NAO in the CanESM2-LE produce" The use of the word impulses implies causality, which may be true for the one-way nesting/spectral nudging methodology but is not for the NAO itself. To avoid confusion perhaps say, "(b) reproduction of the CanESM2-LE NAO flow patterns in the CRCM5-LE produce"

l. 21 "is to apply slight differences in" -> "is to perturb"

ll.21-22 "with similar long-term climate statistics" This refers to a response rather than an experimental setup. I think it might be more correct to say "under identical external forcings"

l. 44 "its dynamics in a future climate" -> "its fidelity in a future climate"

l.61 "is transferred correctly from the driving GCM into the driven RCM". Inter-member spread is not "transferred" from the driving model to the RCM. It would be clearer to say, "is represented consistently between the driving GCM and the driven RCM". Also, from my major comment, representation of NAO inter-member spread is a necessary condition from credible downscaled responses.

ll. 65-66 "finding robust NAO patterns which exceed the uncertainty due to internal variability in the ensemble." The phrase, "exceed the uncertainty due to internal variability"

is confusing in this context. Perhaps say, "finding robust NAO patterns by significantly reducing sampling uncertainty associated with internal variability"

l. 71 "Do GCM NAO impulses propagate correctly into the RCM realizations" perhaps better stated as, "Does the RCM correctly represent the NAO pattern present in the driving data" (ie my major comment)

ll. 68-74. These are excellent focal points/topics for the paper. It would be very helpful if these were better referred back to in the analysis, discussion, and summary sections so the reader can more easily keep track of which of these you are addressing and what progress you have made on each.

ll.101-103. two names are presented for each of three variables (eg msl/psl, t2m/tas, and tp/pr). I did not see a reason for this. If there is a reason it should be stated. If there isn't, then it would be clearer if just one name was presented for each and used throughout the paper.

ll.120-139. It would be very helpful here to provide a schematic, say of the range/extent displayed in Fig.2, where the RCM domain is indicated and where all of the regions discussed in this section were labeled . Not until I got to Fig 2 did the layout of things become clearer to me. Even then I had to look up Leduc (2019) to understand the relative positioning of the RCM domain.

l. 200 Fig.1 This figure is very faint and it is very hard to distinguish between the three cases being presented here. The authors should work on making these results clearer by using more vivid colours and/or fills.

l. 208 "Pairwise correlations between the members". As discussed in Leduc et al. (2019), The CanESM2-LE was spawned in 1950 from 5 independent historical realizations (separated by 150 years of coupled integration each - including 50 years of preindustrial simulation between the launch of each ensemble member). As such, each of the 5 groups of 10 are highly independent of each other. The question of independence applies to the members within each group of 10 which has only 30years of coupled integration to develop independence prior to the 1981-2010 analysis period. Wouldn't a better check of independence be to form two correlation groups? The first would involve pairwise correlations between each member and the 40 other members from the 4 other groups that were spawned from a different CanESM2 realization in 1950. This first group would form a control assumed to be highly independent. The second group would involve pairwise correlations between each member and the 9 other members of the same group spawned from the same CanESM2 realization in 1950. Plots like figure 1b for this latter group could be compared to similar plots of the control group to assess the independence of the ensemble members most likely to have residual correlations during the 1981-2010 period.

l. 211 " They are not systematically related to the ERA-I (the \u201creference\u201d) realization." Why would they be? I don't understand the reasoning behind this correlation. If you are looking for a control group, a much larger group could be formed by the suggestion immediately preceding this point.

l. 214 "positive, negative and indifferent index values" -> "positive, negative and neutral index values"

l. 223 "it backs the choice" -> "it supports the choice"

ll.312-390 Discussion section. The references and discussion here are quite detailed and require constant back-and-forth reference to the earlier sections. For example, the opening statement of the second paragraph states, "The strong psl gradient suggests an overestimation of the local atmospheric circulation with too strong westerlies over the North Atlantic in the background state within the CanESM2-LE." What gradient? Where? The reader has to stop to review the previous sections to determine the context of this statement. This extends to the use of quantities that were defined in previous sections. For example, "Concerning NAO responses, they are most reliable in regions where r is significant (i.e. $|r| > 0.361$ for $p \leq 0.05$,...". "r" may have been

define earlier but the reader must stop here to find where that was to understand this context. (Also "Historical $\u03b11$ values" l. 327.) This discussion needs to be elevated somewhat out of the details of the previous section, summarize those outcomes and their implications, and connect back to the 4 key issues outlined in the introduction.

l. 321 "less prone to incidental fluctuations of single realizations" -> "less prone to sampling uncertainty"

ll. 323-325 "On the other hand, lower correlation values ($|r| < 0.361$) suggest that climate variability at the local scale evolves differently from the global teleconnection. In these cases, the NAO is not the most important contributor and $\u03b5Y$ in Eq. (2) is dominant. Since the index was obtained from raw psl data, it contains the NAO contribution, but possibly also of other teleconnection patterns and noise." There is also the possibility that the large-scale NAO pattern in these regions was not reproduced correctly in the RCM. See my major comment.

ll 341-343 "Another possible explanation could be that the control exerted by CanESM2 through the CRCM5 lateral boundary conditions (LBC) is insufficient, but this is unlikely given the relatively small CRCM5 domain". Adopting the suggestion in my major comment would explicitly address this key issue.
* * *

---

## Author Comment (AC1) · 12 Dec 2019

**Detailed Responses to Referee Comment No. 1**

Dear Reviewer No. 1,

We would like to thank you very much for your constructive and valuable comments. We think that they will considerably improve the scientific quality of our manuscript. Below, we answer to all points raised in the Referee Comment (comments in grey, answers in black). Please consider also the responses to Reviewer No. 2 as there are some cross-overs regarding the comments.

With kind regards,
Andrea Böhnisch on behalf of all co-authors
12 December 2019

The manuscript presents an analysis of changes in the North Atlantic Oscillation (NAO) under a global warming scenario, using two 50-member model ensembles: an ensemble of a global general circulation model, and an ensemble of a high-resolution nested regional climate model. The large ensemble size allows the authors to not only analyze the change in the mean NAO, but also in its variability. The authors also show the impact of the NAO and its variability on European climate. This manuscript presents an interesting study that combines two state-of-the-art techniques: very large ensembles to estimate transient change of internal climate variability, and a high-resolution regional climate model. The results are novel and relevant. However, I think there is some unused potential in the study that should be harvested (see my specific comments below), and the presentation of the results could be improved. I think the manuscript is a good fit for Earth System Dynamics and should be published. That being said, the manuscript requires structural clarification that warrants a major rewrite, so that I recommend major revisions to the manuscript before publication can be considered.

*Thanks for the generally positive reception of the manuscript. We agree that the manuscript needs further structural improvement which was mentioned by both reviewers. We will work on the presentation of the results in order to harvest the unused potential; please see responses to the specific comments below. We also intend to include an analysis of the large-scale SLP pattern in the RCM data (see Major Comment of Reviewer No. 2) as we think that this may help us to interpret NAO responses in central Europe.*

Specific Comments:

l. 2         *"...(NAO) which is a relevant index for quantifying natural variability..."* I find this sentence to be ambiguous. What is a relevant index? As it stands now, it seems to be the mass advection triggered by the NAO. I suspect that the authors mean the NAO itself. If this is the case, I think this ambiguity can be avoided by introducing a comma between "(NAO)" and "which".

*Yes, the second part of this sentence is referring to the NAO. We will introduce a comma to clarify.*

l. 4         Is the link to the CORDEX project really needed in the abstract? Please consider removing it.

*As the link to the ClimEx project is also included in the Data section, it is not needed in the abstract. We will remove it.*

ll. 4-6      This sentence is missing the crucial information that the "LE" model is a nested regional climate model.

*This is true. The missing information will be included.*

l.9          I do not see how the word "strength" in brackets on its own relates to "pearson correlation coefficient". Please re-evaluate whether "strength" adds any meaning at this point.

*The word "strength" refers to "strength of linear relationship". As it does not add any meaning in the abstract, we may remove it here.*

l. 11        What is a "correct response" to NAO forcing? How is that defined? If it's based on the global model simulation (which I assume it is) I am not sure that "correct" is the right word here.

*We agree that "correct" is ambiguous in this context. To underline the intended meaning, we will rephrase to "… responses which are comparable with reference reanalysis data".*

l. 12          Which relationships weaken in the future? Also, what does it mean and why is it important to show that the amplitude of inter-member spread does not change with anthropogenic forcing?

*This sentence refers to the relationships between the NAO and corresponding responses. The finding that the amplitude of the inter-member spread does not change suggests that internal variability of responses and uncertainty of response assessment are similar in both time periods. – We will include the reference and add a corresponding explanation.*

Introduction     I find the introduction confusing and hard to follow. For example, the first paragraph (ll. 16-22) seems to set the reader up for a follwing paragraph on ensembles, but instead global and regional climate and the NAO are introduced in the next paragraph (ll. 23-32). For another example, the reader expects a discussion of advantages and limitations of different methods to quantify the NAO index after paragraph 3 (ll. 33-37), but paragraph 4 (ll. 38-42) introduces the reader to NAO impacts and its interactions with other modes of climate variability. Moreover, this interaction with other modes of variability is in my opinion not important to the study presented in this manuscript. Both the missing storyline and the lack of focus on the important information for this study are an issue throughout the entire introducion. I therefore recommend that the authors rewrite the introduction with particular attention to the storyline and focusing on the important information, so that the reader can follow the reasoning more easily.

*We agree that the introduction is not written as clearly as it should be. So we will restructure the section in the attempt to focus on the four major topics of interest – internal variability, the NAO, nesting and ensemble approaches. We will open our introduction with the explanation of internal variability, introduce the NAO as a mode of internal climate variability, continue with the NAO representation in various climate models, introduce the ensemble approach with which one can assess potential NAO shapes (i.e. the model internal variability of the NAO) and close with the necessity of regional climate models when analyzing NAO responses in heterogeneous regions. We will also attempt to clarify the study goals within the four topics. The key questions will integrate the four topics and represent the structure of the following analyses.*
*We will also remove the information on interactions of the NAO with other modes of climate variability, as it indeed is not crucial for the study.*

l. 38          There is no mention of a positive state before. I believe the authors are referring to a positive NAO state, but that needs to be made explicit, especially so at the very beginning of a paragraph.

*Yes, that is true. The "positive state" refers to the positive NAO state introduced in lines 26-27. To improve the readability, we will change the beginning of this paragraph to: "Compared to the neutral state, the positive NAO state leads to warmer …"*

ll. 75-76          Please consider omitting the "table of contents" at the end of the introduction. It does not add to the story and takes focus off the nice overview of key questions that will be addressed in the paper just before.

*It is true that the "table of content" does not add important information. We will remove it.*

ll. 80-86          I think somewhere here it would be important to mention which region the regional model covers. Please consider adding this crucial information.

*Currently, the information on the region follows in lines 87-90. We will change the order of some phrases to provide the information on cover regions earlier.*

l. 91          The implications of this sentence would be much easier to understand, if the CORDEX ensemble was introduced very briefly. Please consider adding a few words on what the CORDEX ensemble is, as well as a literature reference.

*Thanks, this is a good hint. Other than the ClimEx ensemble, the CORDEX ensemble consists of several GCM-RCM combinations set up in a coordinated modelling framework, and aims at evaluating model variability. We will include a short summary on the CORDEX ensemble, based e.g. on Giorgi et al. 2009.*

ll. 95-96          I am not sure that I agree with the conclusion, that "the most important" modes of climate variability are captured by the ClimEx model, as this conclusion is here based on a comparison to another model ensemble. I agree that it is reasonable to assume from this comparison that the ClimEx model produces reasonable climate variability, but I do not think such a comparison warrants a judgment on which mode of variability is important or not. Please consider rephrasing.

*We agree that the focus of this paragraph should not be set on the judgement of importance of modes. We will rephrase the sentence accordingly.*

ll. 100-103    The most commonly used acronyms for sea level pressure and surface air temperature are SLP and SAT, respectively. Why did the authors decide to use different abbreviations? This is not a huge issue, but interrupts the flow when reading. Also, t2m and tas are usually not the same in model output. The manuscript would benefit from clarification as to which of the two is used in this study – this is currently not clear.

*Thanks for this hint. We will change the names from psl → SLP, tas → nSAT (near surface air temperature) and pr → PR whenever the variable is meant. Table 1 introduces the model output variable names, which is why we will keep psl, tas, pr etc. in there.*

*In CanESM2 and CRCM5 "tas" refers to near surface air temperature, and ERA-I variable "t2m" is 2-m temperature. We assumed that t2m is the ERA-I variable that is most similar to the model variable. We will place an explanation in the manuscript.*

l. 120    The text says that there are two regions of interest, while table 2 specifies seven regions and the remaining manuscript references those seven regions. I suggest omitting the "two regions" phrase, as it is more confusing than helpful at this point.

*Originally, the "two regions" in this phrase refer to the NAO formation (1) and response (2) regions. We agree that the mention of seven analysis regions in Table 2 does not fit the "two regions" phrase, so we will remove it. Following a suggestion of Referee Comment 2, we will also replace Table 2 with a labeled map indicating the size and position of the regions of interest.*

l. 140    The authors use past tense to describe the present study here, and this appears to be the dominant choice of tense. Elsewhere, however, present tense is used (e.g. l. 120 "...there are two separated regions..."). This inconsistency can be found throughout the entire manuscript. To improve readability, I suggest the authors decide on one tense and stick to it throughout the manuscript.

*Thanks, we will fix this.*

ll. 141-142    The word "representative" is lacking a reference here. The 30-year time horizon leads to an NAO distribution that is representative of what? Please elaborate briefly.

*As stated in the sentence before (lines 140-141), major fluctuations of the natural climate system on several temporal scales are assumed to be included in the 30-year time horizon. Their potential influence on the NAO may thus be seen as represented within the sampled NAO time series. – We will rephrase these sentences (lines 140-142) in order to clarify.*

ll. 144-145    This is an important caveat. I like that this is mentioned here, but missed it in the discussion section. I suggest taking it up again there to make sure this (perfectly acceptable) limitation of the study can be appreciated.

*Thanks, we will include this caveat in the discussion section.*

ll. 150-154    I think this bit would be easier to understand if the order of the phrases was altered to first explain why March can be included and then say that DJFM is used for winter. Please consider making this change.

*Thanks, we will change the order of the sentences.*

ll. 159    I suggest refraining from the statement that a station-based NAO index is "easy" to interpret – its reference is arbitrary (easy for whom?) and it is not a very scientific expression. Please rephrase.

*We agree that "easy to interpret" is not an appropriate expression in this context. We will rephrase the paragraph to justify our choice of calculation method.*

ll. 189-195    This section appears to already present results. Please consider moving it to the results section.

*This paragraph was included to explicitly mention the way internal variability was addressed in this study. It is not intended to present results. We will change the phrasing accordingly to clarify.*

l.200    In lines 97-98, the authors define REF as the ERA-Interim data set. Here, REF appears to refer to the NAO index within the ERA-I data set. Please define REF only once and unambiguously.

*Yes, reference/REF is defined to be anything derived from the ERA-I data set, but this sentence uses REF confusingly. Therefore, we will change the sentence to "… a reference NAO index was derived from ERA-I …"*

ll. 205-206     I am not sure I agree that figure 1a shows that REF (the blue bars) lies "comfortably" within the ensemble spread (grey & red). Particularly negative extremes, but to some degree also positive ones, seem to be underrepresented in the model. Can you please comment on this and possible implications for this study?
*This sentence refers primarily to the x-axis of the histogram in Fig. 1a, not the frequency of occurrences: The index values of the ERA-I NAO index may be found within the CanESM2-LE, that is, between the minimum and maximum index values. It is true, that the distribution of ERA-I index values shows differences towards the distribution of the CanESM2-LE. These differences may partly be explained by the different sample sizes (n_ERA-I = 30, n_CanESM2-LE =1500, we will stress this information in the text); the ERA-I sample is only one realization which is compared with 50 realizations of the ensemble, so deviations between the distributions may occur.*
*A better way to display these results may be the usage of CDFs which is why we will change the type of this figure.*

l. 214     *"...original data in**to** three subsets..."*
*Thanks, we will fix it!*

l. 214     Please consider changing "indifferent" to "neutral" or "average" here and throughout the document.
*Thanks, we will change it to "neutral".*

l. 214     Are the "average psl conditions" referenced here the same as the "MSLP mean" in figure 2? If so, I highly recommend using coherent names (i.e. "mean" or "average" in both cases) to avoid confusion. I had to read this paragraph several times before I understood it.
*Yes, both refer to the same. We will adopt your suggestion and change the wording in both text and Fig. header to "neutral SLP conditions".*

ll. 216-217     Which difference is referenced here? Also, what do over- and underestimation refer to? If this is based on a comparison of figs. 2a and d, I cannot follow the argumentation – actually, it appears to me that the model overestimates mean SLP over the North Sea and underestimates SLP over Greenland. Can you please clarify?
*"Difference" refers to the mean SLP difference between CanESM2-LE and ERA-I (Figs. 2a and 2d, respectively). MSLP over Greenland rises to about 1025 hPa in CanESM2-LE and about 1015 hPa in ERA-I data (hence overestimation in CanESM2 with respect to ERA-I); over the North Sea, MSLP reaches 1000 hPa in the CanESM2-LE and 1010 hPa in ERA-I (hence underestimation in CanESM2 with respect to ERA-I).*
*We will clarify the wording and change the coloring of the MSLP maps to better visualize the differences.*

l. 218     *"...phases also show less pronounced..."* Weren't the anomalies more pronounced in the model than in REF for the mean state? If so, please omit the "also".
*That is true, we will remove the "also".*

ll. 239-240     *"...the spatial patterns of ERA-I and CRCM5/ERA-I differ **more** strongly than in Fig. 3,..."*
*Thanks! We will fix this.*

l. 241     What is the reference for the "more humid conditions"? The lack of a reference for relative statements is an issue that needs addressing throughout the manuscript.
*In this case, the reference is the neutral NAO state. Where necessary, we will include the respective references to clarify (see also following comments regarding the same issue).*

l. 256     The NAO explains less variance than what?
*In this case, a temporal comparison was drawn. We will change the sentence to: "...less variance under future conditions compared to the historical period".*

l. 257     tas std decreases less than what?
*This is also a comparison between both time horizons. We will include: "...in the projected future climate".*

l. 259     While I am sure the inconsiderable change of spatial patterns compares the historical to the projected period, I think it would help to give this information here again.
*We will add this information.*

ll. 259-260    Could you please give a figure reference for the claims made here?

*Of course: the claims refer to panels (g) & (i) in Figs. 3-5. We will include the figure reference in the manuscript.*

l. 264    Is there a particular area for which the transfer of internal variability from GCM to RCM is assessed?

*We assessed the "transfer" in the response regions – that is, spatially explicit in CEUR (see std.dev50 maps/subpanels (d), (f) in Figs. 3-5) and spatially aggregated in NE, BY, SE (see Fig. 7). A corresponding note will be placed in the manuscript.*

l. 277    If large tas deviations do not correspond to high or low α, what do they correspond to?

*Thank you for this question; this sentence is not as clear and detailed as it should be. We will include the correlations between the $\alpha_1$ and std.dev50 values to give a more differentiated impression of the relationship between $\alpha_1$ values and the inter-member spread.*

l. 284    I find the presentation of this reference to figs. 3, 4 and 5, h & i ambiguous. Do you refer to panels h & i of all those plots, or just 5?

*Yes, you are right; the reference is ambiguous. Looking in figs. 3-5, we also noted that there is a mistake; it should be (j), not (i). To clarify that we meant panels (h) & (j) of figs. 3-5, we will change the reference to "panels (h), (j) in Figs. 3-5".*

ll. 301-302    This sentence is difficult to understand due to the many parentheses and different references therein. I highly recommend splitting this sentence in at least two.

*Thank you, we will rephrase this sentence and also the preceding sentences that suffer from the same problem.*

l. 306    I think the "matching subset region time series" warrant a more detailed explanation. As it stands, I am not sure what these are and how to interpret them. As a result I cannot follow the text. Please introduce this metric at least shortly.

*This is a good point. The idea was to compare the variability of tas mean, pr sum and tas std time series of the CRCM5 with the CanESM2 in the subset regions NE, BY, SE. Therefore, we correlated the time series of, e.g. tas mean, derived from the spatially aggregated subset region in CRMC5 with the time series derived from the CanESM2 subset region. These correlations were calculated member-wise, leading to 50 correlation coefficients per subset region. High (low) correlation coefficients indicate a strong (weak) co-variability of the CRMC5 and CanESM2 in the respective member. We will add a brief explanation in the manuscript. Therefore we will rephrase this paragraph (306-310) and also incorporate changes following the next two comments.*

ll. 308-309    I am not sure I fully agree with this statement. While correlations indeed appear to be generally lower for pr sum (fig. 8b), 1/3 regions for tas mean (fig. 8a) and 2/3 regions for tas std (fig. 8c) show an increase towards the later period. I think the manuscript could benefit from a more detailed discussion here.

*These findings are certainly true. We will review and correct the statements, include a more detailed discussion and elaborate on the implications of these findings.*

ll. 309-310    I do not quite understand the last sentence of the "results" section. As a result, I struggle to see what its consequences are. I recommend adding some more explanation here, as this might be a crucial point.

*The last sentence is not as precise as it should be. The results presented in Fig. 8. suggest that there is a larger discrepancy in portraying pr sum and tas std in the RCM with respect to the GCM than for tas mean. We will rewrite the paragraph concerning the "matching subset region time series" (see also comments to lines 306, 308-309) and thereby also elaborate on this finding.*

ll. 314-315    What does it tell us that one realization shows a good correlation to REF? Why are the two so highly correlated? I am not sure why this is mentioned here. As in the introduction, this (apparently) irrelevant information might cause the reader to loose track of what is important. Please consider omitting this sentence or, if you deem it relevant enough, elaborate to illustrate its relevance.

*This realization was mentioned to show that the ensemble may incidentally produce very "realistic" looking realizations. However, we agree that it might seem irrelevant and distracting, so we will remove the sentence.*

l. 316          It is not clear about which strong psl gradient the authors are writing here.

*Yes, this information is missing here. We refer to the SLP gradient over the North Atlantic within the CanESM2 under neutral SLP conditions as seen in Fig. 2 (d). The sentence will be updated with the corresponding information.*

ll. 318-319        NAO+ and NAO- are weaker within CanESM2-LE than which reference?

*The reference (which is indeed missing) is the ERA-I data set. We will include it in the revised manuscript.*

l. 320          The very limited sample size of n=7 (or rather n=3 and n=4) in REF is an important issue that is worrisome. It should be discussed further! How robust are the results presented here? What could maybe be learned about observations from the model?

*We agree that the small sample size is problematic. We will conduct an uncertainty assessment of the samples for positive and negative NAO composites (referring to Fig. 2, panels (b)-(c) and (e)-(f)). For example, we may estimate the standard error of the arithmetic mean on each grid cell for the ERA-I data and compare it with the CanESM2 samples (which are considerably larger). This analysis should show where the anomaly patterns can be seen as robust in both models.*

*However, it may be difficult to learn about observations from the model. Learning from the model about observations would imply that the model internal variability can be seen as "correct" as the observed internal variability which is not easy to estimate as there is only a single realization of observations.*

l. 326          At this point, I somewhat expected a discussion on the influence of other teleconnection patterns. I think the authors should at least provide some indication (from the literature) about how large these teleconnections' influence on this study can be expected to be.

*This is an interesting suggestions, thanks! We will include a short survey on the influence of the East Atlantic Pattern and the Scandinavian Pattern, as we based our NAO index on the SLP gradient over the North Atlantic which occasionally is affected by these teleconnection patterns (see Moore et al. 2013 and Comas-Bru and McDermott 2014).*

l. 335          The latter is not as clear in the chose domain as what?

*The sentence was intended to say: " … is not as clear as the first in the chosen domain". We will rephrase accordingly.*

ll. 338-339        I think the observation is missing a reference in this sentence: Is it NAO+ or NAO-? And are these observations derived from reanalysis or the literature or a model? As it stands, this is quite ambiguous.

*We agree that this sentence is ambiguous. It is meant to refer to the fact that the Jetstream position it altered during the NAO+/NAO- phases and therefore associated air mass advection is displaced (see e.g. Woollings et al. 2015). However, as we do not further refer to the Jetstream in the text and thus the sentence does not add to the argumentation in the discussion, we will not necessarily keep it in the revised manuscript.*

l. 350          Omit the comma between "region" and "which".

*We will fix this.*

ll. 352-353        This is an intriguing thought. What are its consequences/implications? Please consider to elaborate a bit.

*We will include a more detailed discussion on this subject. The GCM reproduces strongest variability in (geographically) other regions than ERA-I, but in the RCM the positions are "correct"; so for example, we may also see added RCM value for regional scale analysis in this.*

l. 361          What does it mean for the findings presented here that the GCM overestimates T and pr? Does this limit the conclusions that can be drawn?

*This information is given as background information. The overestimation of average T and pr does not affect the findings regarding the correlation coefficient and $\alpha_1$ as these are based on the changes/variability. The results in the study thus represent the differences in the model-specific variability related to the NAO (which show also some kind of bias regarding stronger/weaker variability in the RCM/GCM). We will update the text accordingly.*

l. 367    Since the patterns are "only" very similar, I find the statement "atmospheric dynamics are correctly implemented" a bit too strong. Please consider rephrasing to, e.g., "...can be regarded as correctly implemented".
*Thanks, we will rephrase the statement accordingly.*

l. 378    As stated before (comment lines 205-206), I do not agree that the observations lie comfortably within the model spread, so I also have an issue with the statement "...the same climate statistics". Please either explain where I went wrong or rephrase.
*Thanks for your concerns. We agree that "the same climate statistics" sounds too strong. As shown in Fig. 6, the CanESM2 ensemble generally encompasses the REF realization regarding several statistics, e.g. inter-annual variability or number and mean values of positive/negative phases. We will insert clearer references in this paragraph and also rephrase to: "The ensemble also shows comparable climate statistics with the REF time series and patterns".*

ll. 382-383    Maybe rephrase to *"...with highest change in CRCM5-LE, but not* **necessarily** *in CasESM2-LE."*?
*Thanks, we will adopt the wording suggestion.*

l. 391    Less tas and pr variation is explained by NAO than by what?
*"Less" is referring to a comparison between historical and future time frames. We will rephrase accordingly.*

Conclusions    I think the reference to the questions raised in the introduction could be made clearer. While the references are there, I think it would make this part clearer if it was structured in bullet points, like the questions raised in the introduction. Please consider making this change.
*Thanks for this idea. We will put the answers to key questions (a)-(d) in bullet points.*

ll. 397-399    This is a long sentence that is hard to understand because it takes up two different points. Please consider splitting the sentence in two.
*Thank you; we will consider this suggestion.*

l. 404    I find the word "proves" very strong. I agree that the clearly visible topographic features are nice to look at and encouraging for the model presented here, but I disagree with the notion that the mere notice of more pronounced topographic features "proves" the added value of anything. High resolution does not always equal added value. Please rephrase.
*Thanks for this concern. We agree that "proves" sounds rather strong. We will rephrase to "suggests".*

Fig. 2 caption    *"(g)-(i): 2070-2099 changes* **with respect to** *1981-2010"*
*Thanks, we will fix it.*

Figs. 3-5 caption  What are the correlations show in blue isolines? What is correlated to what? Also, this is a confusing figure, partly due to the ambiguous headers for the subpanels (which are identical for, e.g., c and g). Please think about a more intuitive way to convey this very interesting information.
*The blue isolines correspond to lines of equal correlations between the NAO index and the tas mean/tas std/pr sum time series on the grid cells by increments of 0.1. We agree that the bare presentation of blue isolines is rather confusing. We will change the increments to 0.25 (in order to picture less lines) and indicate the correlation strengths by different grey scales (and a legend). We think that figures 3-5 will gain more clarity in doing so. Also, headers and captions will be changed to better indicate what is shown by the single subpanels. For example, headers may be updated with information on the time frame (1981-2010 for first and second row, 2070-2099 for third row).*

Fig. 6    Some of the indices named in the upper left corner have slightly different names than those found on the x-axis. It could help the clarity of the (otherwise very nice and interesting!) figure if those names were the same. Please consider changing the figure accordingly.
*Thank you. We will correct the names in the text box.*

Fig. 8         Please explain a, b and c in the caption. Also, I do not quite understand what is displayed. What is a "similarity of matching regions"?

*These figures display the temporal co-variability of the corresponding CanESM2 and CRCM5 members in the three subset regions (NE, BY, SE) for tas mean (a), pr sum (b) and tas std (c). Thus "matching" refers to the same member in the GCM and RCM.*

*We will include a detailed description of the metric in the text (see also response to comment line l. 306), and change the caption accordingly.*

Fig. A2 caption    Please explain the subpanels in the caption.

*Figure A2 shows the ratio of tas mean $\alpha_1$ and winter tas std for the data sets employed in the study: (a) CRCM5/ERA-I and (b) ERA-I under historical conditions, and CanESM2-LE ((c)-(d)) andCRCMR5-LE ((e)-(f)) under historical and future conditions. – We will add a description of the subpanels in the caption.*

References in this response:

Comas-Bru, L., McDermott, F. (2014): Impacts of the EA and SCA patterns on the European twentieth century NAO-winter climate relationship. Quaterly Journal of the Royal Meteorological Society, 140, 354-363.

Giorgi, F., Jones, C. Asrar, G. R. (2009): Addressing climate information needs at the regional level: the CORDEX framework. WMO Bulletin, 58 (3), 175-183.

Moore, G., Renfrew, I., Pickart, R. (2013): Multidecadal Mobility of the North Atlantic Oscillation. Journal of Climate, 26, 2453-2466.

Woollings, T., Franzke, C., Hodson, D., Dong, B., Barnes, E., Raible, C., and Pinto, J. (2015): Contrasting interannual and multidecadal NAO variability, Climate Dynamics, 45.

---

## Author Comment (AC2) · 12 Dec 2019

**Responses to Referee Comment No. 2**

Dear Reviewer No. 2,

We would like to thank you very much for your constructive and valuable comments, particularly the very interesting suggestions for further analyses. We think that they will considerably improve the scientific quality of our manuscript. Below, we answer to all points raised in the Referee Comment (comments in grey, answers in black). Please consider also the responses to Reviewer No. 1 as there are some cross-overs regarding the comments.

With kind regards,
Andrea Böhnisch on behalf of all co-authors
12 December 2019

In this study a regional climate model (CRCM5) is employed to dynamically downscale a single global climate model (CanESM2) large ensemble of climate change simulations to investigate the nature of downscaled responses to the modeled North Atlantic Oscillation (NAO) and its influence on future European climate. By employing a large ensemble, the authors are able to evaluate future downscaled responses associated NAO inter-annual variability in addition to mean changes. The authors set out four key questions related to, documenting the properties and fidelity of the modeled NAO in both the GCM and RCM; the associated screen temperature and precipitation responses in both models; and how such properties change under future external forcings (following the future CMIP5 pathway RCP8.5).

This is an interesting paper and ultimately worthy of publication. The authors present the problem from the perspective of downscaling teleconnections that exist in the driving data (ie the NAO). This is a subtle but critically important shift in focus for the dynamical downscaling community. The proper communication of teleconnection patterns/relationships from driving data to the RCM is essential for credible downscaled results. The use of a large GM/RCM ensemble pair positions the authors to say something definitive about this problem and offer guidance to the community.

The four key questions represent a clear and sensible plan for the paper. However, I found it difficult at times to cleanly connect a particular analysis performed by the authors with an answer to some of these questions. Specifically, I do not think that the authors addressed the first part of their question 3, "Do GCM NAO impulses propagate correctly into the RCM realizations" (l. 71). Perhaps a better way of stating this is, does the RCM faithfully represent the NAO pattern present in the driving data? This is a critical question in the authors' "model chain" (l. 65) that needs to be addressed before one moves on to evaluate the NAO responses. That is, if the largescale NAO pattern is not faithfully represented in the RCM domain in some location, the downscaled responses in that location would be less credible. The increased resolution and potentially improved physical processes present in the RCM themselves cannot correct the large-scale NAO pattern within the RCM domain. As the authors discuss, the NAO pattern is governed by "planetary wavebreaking in the polar front" (Benedict et al., 2004), which is intern influenced by external factors such as sea-ice, snow cover, sea-surface temperatures, ENSO, stratospheric circulation variability, solar variability, volcanic eruptions and the Quasi-Biennial Oscillation (eg Hall et al. 2014 https://doi.org/10.1002/joc.4121).

Given that the European domain is relatively small, and the experimental design employs spectral nudging in the RCM, the NAO pattern, and its interannual variability, should on balance be reasonably represented in the RCM. For the authors' stated plan, however, this needs to be verified. Given that the authors employ a large ensemble in their study, they are in the unique position to definitively address this issue and provide an example to the community of the type of analysis that is required to support the credibility of downscaled results in such complex problems. It is my recommendation that, prior to publication, the manuscript undergo major revision to address this issue and to improve its overall clarity. My detailed comments follow.

*Thank you very much for this generally positive assessment of the study scope, but also for the concerns regarding key question 3. This question originally targeted the question whether the combination of NAO indices from the GCM and response variables from the RCM produces realistic looking NAO responses in the RCM. The suggested formulation changes its meaning towards the nesting of the NAO/SLP pattern itself. However, in light of the fact that indeed the assessment of large-scale SLP patterns in the RCM data is relevant but missing so far, this change of formulation is justifiable. We will adopt the suggestion in the major comment (see our point-by-point responses) and include the results in our assessment of the NAO and NAO responses.*

*The lack of (structural) clarity in the manuscript is also criticized by Reviewer No. 1, and we will work on the manuscript to address this issue.*

Major Comment:

RCM reproduction of NAO teleconnection in driving data

As part of the authors' model chain, it is essential to verify that the large-scale year-to year variations of the NAO pattern in surface pressure are faithfully reproduced (each year) in CRCM5 when driven by both ERA-I and CanESM2. Inspired by Fig. 2, the sort of analysis required would be as follows:

- interpolate monthly-mean timeseries of sea-level pressure (SLP) in the driving dataset onto the RCM grid (such interpolation is already done for the driving-data winds used for spectral nudging). Call this field SLP_Drive.

- take the difference of the RCM and driving data monthly-mean SLP on the RCM grid SLP_RCM - SLP_Drive, and then smooth the result retaining large scales that are representative of the driving data resolution:

$$D\_m(i,j,t,n) = [SLP\_RCM - SLP\_Drive]\_LRG$$

Here, i,j are lateral spatial coordinates of the RCM grid, t is time in units of years, n is ensemble member, and the subscript m corresponds to month (1-12). The smoothing operation, represented by the operator [ ]_LRG, can be performed with the same double-cosine transform used for the spectral nudging.

- derive a normalized root-mean-square difference map for extended winter, over the two 30-year periods displayed in Fig 2, over all ensemble members:

$$RMS (i,j) = Ave\_(m=12,1-3)\{Ave\_n \{ SQRT[Ave\_t \{D\_m(i,j,t,n)^2 \}] / Var\_Drive\_m(i,j,n)\}\}$$

where, Ave_x is a simple averaging operators for the quantity x and Var_Drive_m is the variance in time of the driving data for each month and each ensemble member:

$$Var\_Drive\_m(i,j,n) = Ave\_t \{[ SLP\_Drive\_m (i,j,t,n) - Ave\_t \{SLP\_Drive\_m (i,j,t,n) \} ]^2\}.$$

Normalization by Var_Drive_m is important as it indicates the size of an rms difference relative to the interannual variability in the NAO pattern at that location. Such an RMS map would provide a sensible measure of the difference in the driving data and RCM SLP patterns associated with the NAO, which need to be faithfully reproduced in each year. If RMS « 1 at a given location, then the large-scale NAO pattern is well represented there and one can conclude that the downscaling is consistently being performed on the "correct" large-scale flow. The larger RMS is, towards O(1) values, the more suspect the downscaled responses are at that location (ie a large-scale flow disconnected from the NAO in the driving data was being downscaled in these regions).

One should also do a significance test and indicate this by, say, filling in contours by color in only those regions that are significant at the 5% level. Given the size of the GCM/RCM ensemble, this should be quite robust (ie much of the canvas should be colored) and definitive statements could be made. This test would seem to be most well posed for the case of observational driving of the RCM (ie ERA-I driving of CRCM5 over the historical period 1981-2010). The large scales in that data are well observed and, because they came from the real system, they were influenced realistically by all processes and scales. Significant deviations in RMS(i,j) for ERA-I (ie RMS_ERA-I) would necessarily indicate a degradation of the NAO teleconnection in those regions of the CRCM5 domain.

If regions of NAO deviation in RMS_ERA-I were consistent with regions of NAO deviation in RMS_CanESM2 (in the historical and even the future periods), then this would indicate a systematic issue with the reproduction of the NAO pattern in the European domain in these locations and care should be taken in the interpretation of the downscaled responses in this, and possibly other RCM studies using the same domain.

*Thanks for this very detailed suggestion! It is true that the original analysis did not include an assessment of the large-scale RCM SLP pattern. We will thus adopt this suggestion with some slight modifications, the first one being that we will interpolate the RCM data (and also the ERA-I driving data) to the GCM grid. This will be done in order to not create additional errors during the interpolation onto the high resolution RCM grid. By aggregating the data, we will also filter the small scales, retaining only the large-scale patterns. The RMS error*

*will be calculated on the entire ClimEx domain. We expect the RMS to be quite low in the entire domain, and we will conduct a significance test on H0: RMS(i,j) >=1 (maybe also 0.5 or even lower).*

Minor Comments:
l.2                 "natural variability". Later it seems, "internal variability" (l. 16) is used to refer to the same phenomenon. It would be helpful to be consistent throughout.
*This is true, thank you. We will use "internal variability" throughout the study.*

l. 5-6.            "its transfer from the driving model CanESM2 into the driven model CRCM5." Perhaps better wording might be "its representation in the driven model CRCM5 relative to the driving model CanESM2."
*Thank you, we will adopt the wording suggestion.*

l.11                "(b) impulses from the NAO in the CanESM2-LE produce" The use of the word impulses implies causality, which may be true for the one-way nesting/spectral nudging methodology but is not for the NAO itself. To avoid confusion perhaps say, "(b) reproduction of the CanESM2-LE NAO flow patterns in the CRCM5-LE produce"
*Thanks, we will clarify this phrase accordingly.*

l. 21               "is to apply slight differences in" -> "is to perturb"
ll.21-22          "with similar long-term climate statistics" This refers to a response rather than an experimental setup. I think it might be more correct to say "under identical external forcings"
l. 44              "its dynamics in a future climate" -> "its fidelity in a future climate"
*Thanks, we will consider the phrasing suggestions for lines 21, 21-22, 44.*

l.61               "is transferred correctly from the driving GCM into the driven RCM". Inter-member spread is not "transferred" from the driving model to the RCM. It would be clearer to say, "is represented consistently between the driving GCM and the driven RCM". Also, from my major comment, representation of NAO inter-member spread is a necessary condition from credible downscaled responses.
*Thanks for your explanation. We will rephrase the sentence accordingly.*

ll. 65-66         "finding robust NAO patterns which exceed the uncertainty due to internal variability in the ensemble." The phrase, "exceed the uncertainty due to internal variability" is confusing in this context. Perhaps say, "finding robust NAO patterns by significantly reducing sampling uncertainty associated with internal variability"
*The suggested formulation is certainly clearer than the original one. We will adopt the suggestion.*

l. 71              "Do GCM NAO impulses propagate correctly into the RCM realizations" perhaps better stated as, "Does the RCM correctly represent the NAO pattern present in the driving data" (ie my major comment)
*We agree that key question (c) is better stated in this way as the suggested wording also encompasses the additional analyses regarding the large-scale SLP pattern. We will rephrase the sentence.*

ll. 68-74.        These are excellent focal points/topics for the paper. It would be very helpful if these were better referred back to in the analysis, discussion, and summary sections so the reader can more easily keep track of which of these you are addressing and what progress you have made on each.
*Thanks! We will structure sections 3—5 accordingly. Please have also a look at Referee Comment 1 regarding the Introduction and Conclusions sections.*

ll.101-103.       two names are presented for each of three variables (eg msl/psl, t2m/tas, and tp/pr). I did not see a reason for this. If there is a reason it should be stated. If there isn't, then it would be clearer if just one name was presented for each and used throughout the paper.
*Thanks for this note. Please have also a look at the responses to Referee Comment 1 where we address a similar issue. The two names refer to different model variable output names (e.g. msl, t2m, tp were derived from ERA-I, psl, tas, pr from CanESM2 and CRCM5). We will also change the analysis variable names like psl → SLP, tas mean/std → nSAT mean/std, pr sum → PR sum in the text.*

ll.120-139. It would be very helpful here to provide a schematic, say of the range/extent displayed in Fig.2, where the RCM domain is indicated and where all of the regions discussed in this section were labeled . Not until I got to Fig 2 did the layout of things become clearer to me. Even then I had to look up Leduc (2019) to understand the relative positioning of the RCM domain.

Thanks for this hint. We will replace Table 2 with a map which will indicate all domains employed in the study. We think this is a better way to illustrate the position and extent of the domains than listing the boundary coordinates.

l. 200 Fig.1 This figure is very faint and it is very hard to distinguish between the three cases being presented here. The authors should work on making these results clearer by using more vivid colours and/or fills.

*Yes, this is certainly true. In combination with the following suggestion we will change the colors and possibly also the diagram style (CDFs instead of histograms).*

l. 208 "Pairwise correlations between the members". As discussed in Leduc et al. (2019), The CanESM2-LE was spawned in 1950 from 5 independent historical realizations (separated by 150 years of coupled integration each - including 50 years of preindustrial simulation between the launch of each ensemble member). As such, each of the 5 groups of 10 are highly independent of each other. The question of independence applies to the members within each group of 10 which has only 30years of coupled integration to develop independence prior to the 1981-2010 analysis period. Wouldn't a better check of independence be to form two correlation groups? The first would involve pairwise correlations between each member and the 40 other members from the 4 other groups that were spawned from a different CanESM2 realization in 1950. This first group would form a control assumed to be highly independent. The second group would involve pairwise correlations between each member and the 9 other members of the same group spawned from the same CanESM2 realization in 1950. Plots like figure 1b for this latter group could be compared to similar plots of the control group to assess the independence of the ensemble members most likely to have residual correlations during the 1981-2010 period.

*This is a very nice idea. We will consider an analysis following these steps. In order to better discriminate the different groups (and periods) we will also switch from histograms to CDFs. Names of the two groups may be SOIC – "same ocean initial conditions" (looking at members from the same family), and MOIC – "mixed ocean initial conditions" (looking at members from different ocean families) as in Leduc et al. 2019.*

l. 211 "They are not systematically related to the ERA-I (the "reference") realization." Why would they be? I don't understand the reasoning behind this correlation. If you are looking for a control group, a much larger group could be formed by the suggestion immediately preceding this point.

*When correlating the ERA-I realization with the 50 CanESM2 members we were not so much looking for a control group. The idea was to evaluate whether ERA-I can be seen as another independent realization, but with comparable climate statistics. We will include a short explanation on the reasoning in the text.*

l. 214 "positive, negative and indifferent index values" -> "positive, negative and neutral index values"

*Thanks, we will change "indifferent NAO states/index values" to "neutral" in the text. Please have also a look at Referee Comment 1.*

l. 223 "it backs the choice" -> "it supports the choice"
*Thanks, we will change it.*

ll.312-390 Discussion section. The references and discussion here are quite detailed and require constant back-and-forth reference to the earlier sections. For example, the opening statement of the second paragraph states, "The strong psl gradient suggests an overestimation of the local atmospheric circulation with too strong westerlies over the North Atlantic in the background state within the CanESM2-LE." What gradient? Where? The reader has to stop to review the previous sections to determine the context of this statement. This extends to the use of quantities that were defined in previous sections. For example, "Concerning NAO responses, they are most reliable in regions where r is significant (i.e. $|r| > 0.361$ for p $\leq$ 0.05,...". "r" may have been define earlier but the reader must stop here to find where that was to understand this context. (Also "Historical $\Delta^1$ values" l. 327.) This discussion needs to be elevated somewhat out of the details of the previous section, summarize those outcomes and their implications, and connect back to the 4 key issues outlined in the introduction.

*We agree that there is a lot "back and forth" which is related to the fact that we tried to respect the strict separation of the results and discussion sections. In order to improve readability though and following some comments of Referee Comment 1, we will restructure the discussion with respect to the four key questions, integrate the results and remove too detailed repeats of results.*

l. 321 "less prone to incidental fluctuations of single realizations" -> "less prone to sampling uncertainty"
*Thanks, we will change the formulation.*

ll. 323-325 "On the other hand, lower correlation values ($|r| < 0.361$) suggest that climate variability at the local scale evolves differently from the global teleconnection. In these cases, the NAO is not the most important contributor and $\nu_Y$ in Eq. (2) is dominant. Since the index was obtained from raw psl data, it contains the NAO contribution, but possibly also of other teleconnection patterns and noise." There is also the possibility that the large-scale NAO pattern in these regions was not reproduced correctly in the RCM. See my major comment.
*This is an interesting connection. We will look into the data to assess it. However, we have already established that low correlations (weak NAO relationships) do not occur in the same locations for all variables (e.g. for pr sum, they are mostly located near the Alps; but for tas mean they are found in the southern part of CEUR, where pr sum shows relatively strong correlations with the NAO).*

ll 341-343 "Another possible explanation could be that the control exerted by CanESM2 through the CRCM5 lateral boundary conditions (LBC) is insufficient, but this is unlikely given the relatively small CRCM5 domain". Adopting the suggestion in my major comment would explicitly address this key issue.
*We will also analyze the RMS results to assess the SLP representation between GCM and RCM.*

Reference:

Leduc, M., Mailhot, A., Frigon, A., Martel, J.-L., Ludwig, R., Brietzke, G., Giguère, M., Brissette, F., Turcotte, R., Braun, M., and Scinocca, J. (2019): The ClimEx Project: A 50-Member Ensemble of Climate Change Projections at 12-km Resolution over Europe and Northeastern North America with the Canadian Regional Climate Model (CRCM5), Journal of Applied Meteorology and Climatology, 58, 663–693.

---

## Author Response (AR1)

Dear reviewers, dear editor,

We would like to thank you for your constructive advice and suggestions. We are confident that these truly improved the quality of our manuscript.
Please find our point-by-point responses to the issues raised in the Referee Comments below. The structure of this document is as follows:

1) Responses to Referee Comment No. 1

2) Responses to Referee Comment No. 2

3) Responses to Editor Comment

4) Revised manuscript with track changes

There are some general remarks that do not apply to any specific comment, but arose during the revision of the manuscript:

- Appendix Figures: We removed (old) Figs. A3, A4 and A6. The information included in Figs. A3 and A4 was inserted in new Figs. 6 and 8 (i.e. spatial distribution of α1; stippling now indicates regions with SNR < 1). Fig. A6 (i.e. 50 GCM index time series for 1981-2010) was not regarded as adding meaningful information to the discussion in the manuscript.

- Additional Figures: In order to clarify that GCM members may be regarded as independent in the early time frame, we included a figure showing the inter-member standard deviations among five-member groups with the same ocean initial conditions and among five-member groups with mixed ocean initial conditions. We also removed Table 2 (position and size of study regions) and present the information in a map.

- Additional abbreviation: special attention was paid to the use of words like "inter-member spread", "internal variability", "noise", "std.dev50" which were used somewhat interchangeably and imprecise in the discussion paper. In order to clarify, we tried to use "inter-member spread" whenever we meant the range (maximum to minimum) of members in the ensemble (new abbreviation: IMS). When referring to the spatially distributed IMS expressed as the standard deviation among the 50 members, we used "ensemble sd" as it does not mean exactly the same as IMS. We also decided to drop the term "std.dev50" which was not used consistently in the manuscript.

- Additional analyses: following a major comment of reviewer No. 2, we included an analysis on the large-scale RCM SLP pattern within the CEUR domain.

- Information on the lines with changes and figure/table references refer to the updated manuscript if not stated otherwise.

- Correspondence: we changed the e-mail address from a.boehnisch@iggf.geo.uni-muenchen.de to a.boehnisch@lmu.de

The comments raised by the referees are marked in blue, responses in black (explanations in italics).

With kind regards,

Andrea Böhnisch on behalf of all co-authors
17 February 2020

**Responses to Referee Comment No. 1**

The manuscript presents an analysis of changes in the North Atlantic Oscillation (NAO) under a global warming scenario, using two 50-member model ensembles: an ensemble of a global general circulation model, and an ensemble of a high-resolution nested regional climate model. The large ensemble size allows the authors to not only analyze the change in the mean NAO, but also in its variability. The authors also show the impact of the NAO and its variability on European climate. This manuscript presents an interesting study that combines two state-of-the-art techniques: very large ensembles to estimate transient change of internal climate variability, and a high-resolution regional climate model. The results are novel and relevant. However, I think there is some unused potential in the study that should be harvested (see my specific comments below), and the presentation of the results could be improved. I think the manuscript is a good fit for Earth System Dynamics and should be published. That being said, the manuscript requires structural clarification that warrants a major rewrite, so that I recommend major revisions to the manuscript before publication can be considered.

*Thanks for the generally positive reception of the manuscript. We worked on the presentation and text structure in order to increase readability. For example, the introduction and discussion sections now clearly follow the key questions raised at the end of the introduction. Additional analyses regarding the large-scale SLP pattern present in the RCM data within the European domain were performed, and we aimed at better assessing the uncertainty within the analyses.*

Specific Comments:

l. 2 *"...(NAO) which is a relevant index for quantifying natural variability..."* I find this sentence to be ambiguous. What is a relevant index? As it stands now, it seems to be the mass advection triggered by the NAO. I suspect that the authors mean the NAO itself. If this is the case, I think this ambiguity can be avoided by introducing a comma between "(NAO)" and "which".

*We included the comma in order to avoid the mentioned ambiguity.*

l. 4 Is the link to the CORDEX project really needed in the abstract? Please consider removing it.

*In the submitted discussion manuscript, the link to the ClimEx project was also included in the data section, such that it was indeed not needed in the abstract. When rewriting the introduction, we decided to move it to the first mention of the project. The text now reads:*

l. 61-63: "Such downscaling of a GCM single-model large ensemble was performed within the Climate Change and Hydrological Extremes project (ClimEx, www.climex-project.org, Leduc et al., 2019)."

ll. 4-6 This sentence is missing the crucial information that the "LE" model is a nested regional climate model.

*We updated the sentence accordingly. It now reads:*

l. 4-5: "In this study, 50 members of a single-model initial condition large ensemble (LE) of a nested regional climate model were analyzed for a NAO-climate relationship."

l.9 I do not see how the word "strength" in brackets on its own relates to "pearson correlation coefficient". Please re-evaluate whether "strength" adds any meaning at this point.

*"Strength" refers to the strength of the linear relationship expressed by the Pearson correlation coefficient, but it does indeed not add additional information. We rewrote this part of the abstract, thereby removing the word "strength":*

l. 7-10: "Responses of mean surface air temperature and total precipitation to changes in the index value are expressed for a Central European domain in both the CanESM2-LE and CRCM5-LE via Pearson correlation coefficients and the change per unit index change for historical (1981–2010) and future (2070–2099) winters."

l. 11 What is a "correct response" to NAO forcing? How is that defined? If it's based on the global model simulation (which I assume it is) I am not sure that "correct" is the right word here.

*We agree that "correct" is ambiguous in this context. To underline the intended meaning, we rephrased:*

l. 11-12: "Reproductions of the NAO flow patterns in the CanESM2-LE trigger responses in the high-resolution CRCM5-LE that are comparable with reference reanalysis data."

l. 12        Which relationships weaken in the future? Also, what does it mean and why is it important to show that the amplitude of inter-member spread does not change with anthropogenic forcing?
*This sentence refers to the relationships between the NAO and corresponding responses. The finding that the amplitude of the inter-member spread does not change suggests that internal variability of responses and uncertainty of response assessment are similar in both time periods. – We included the reference and added a corresponding explanation.*

l. 12-13: "NAO–response relationships weaken in the future period, but their inter-member spread shows no significant change."

l. 488-490: "When comparing present and future values, a vertical shift of the boxes in Fig. 10 indicates that r is reduced in the future, but the inter-quartile distance of the r distributions (box size) stays nearly the same for GCM and RCM. This shows that the uncertainty range of the signals does not change significantly in the future horizon."

Introduction        I find the introduction confusing and hard to follow. For example, the first paragraph (ll. 16-22) seems to set the reader up for a follwing paragraph on ensembles, but instead global and regional climate and the NAO are introduced in the next paragraph (ll. 23-32). For another example, the reader expects a discussion of advantages and limitations of different methods to quantify the NAO index after paragraph 3 (ll. 33-37), but paragraph 4 (ll. 38-42) introduces the reader to NAO impacts and its interactions with other modes of climate variability. Moreover, this interaction with other modes of variability is in my opinion not important to the study presented in this manuscript. Both the missing storyline and the lack of focus on the important information for this study are an issue throughout the entire introducion. I therefore recommend that the authors rewrite the introduction with particular attention to the storyline and focusing on the important information, so that the reader can follow the reasoning more easily.
*We restructured the introduction in an attempt to focus on the four major topics of interest – internal variability, the NAO, nesting and ensemble approaches. We now open our introduction with the explanation of internal variability, introduce the NAO as a mode of internal climate variability, continue with the NAO quantification and representation in various climate models, introduce the ensemble approach (in order to assess NAO internal variability) and close with the necessity of regional climate models when analyzing NAO responses in heterogeneous regions.*
*Finally, we integrate the aforementioned topics to present our research question, which in turn leads to our four key questions. Among the key questions, we changed the order (switch (b) and (c)) as this seemed more consistent with the analyses in the study.*
*We also removed the information on interactions of the NAO with other modes of climate variability, as it is not crucial for the study. Instead, we included a short paragraph on the interactions between NAO and the East Atlantic/Scandinavian Pattern in the discussion section (see below, response to RC1 l. 326)*

l. 38        There is no mention of a positive state before. I believe the authors are referring to a positive NAO state, but that needs to be made explicit, especially so at the very beginning of a paragraph.
*The "positive state" refers to the positive NAO state introduced in (discussion paper) lines 26-27. To improve the readability, we rephrased:*

l. 25-28: "Its two states, positive and negative, are evoked by planetary wave-breaking in the polar front, leading to antagonistic pressure behavior of two centers over the North Atlantic: one located within the subtropical high pressure belt ("Azores High", AH), the second in subpolar regions ("Icelandic Low", IL) (Benedict et al., 2004)."

l. 31-33: "Compared to neutral conditions, the positive NAO state leads to warmer and moister winters in northern Europe, but cooler and drier conditions in the south, and vice versa in the negative state (e.g., Hurrell and Deser, 2009; Pokorná and Huth, 2015; Woollings et al., 2015)."

ll. 75-76        Please consider omitting the "table of contents" at the end of the introduction. It does not add to the story and takes focus off the nice overview of key questions that will be addressed in the paper just before.

*Thank you! We removed the "table of contents".*

 I think somewhere here it would be important to mention which region the regional model covers. Please consider adding this crucial information.

*We re-structured this paragraph. It now starts with a more detailed explanation of the GCM LE before moving to the RCM LE. The information on the domains covered by the regional model follows immediately after:*

l. 91-93: "As described in Leduc et al. (2019), these 50 GCM members were dynamically downscaled using the Canadian Regional Climate Model version 5 (CRCM5 Large Ensemble, 0.11° spatial resolution) over two domains covering Europe and north-eastern North America."

l. 91 The implications of this sentence would be much easier to understand, if the CORDEX ensemble was introduced very briefly. Please consider adding a few words on what the CORDEX ensemble is, as well as a literature reference.

*Thanks, this is a good hint. Other than the ClimEx ensemble, the CORDEX ensemble consists of several GCM-RCM combinations set up in a coordinated modelling framework, and aims at evaluating model uncertainty. We included a short comparison between both model ensembles after the first mention of the CORDEX ensemble:*

l. 96-103: "Comparing the internal variability of the CRCM5 members with the IMS of a subset of the multi-model EURO-CORDEX (Coordinated Regional climate Downscaling Experiment) ensemble regarding winter temperature and precipitation, von Trentini et al. (2019) showed that both ensemble spreads are of comparable magnitude. The CORDEX ensemble consists of several GCM-RCM combinations set up in a coordinated modelling framework and aims at evaluating uncertainty due to model configuration (Giorgi et al., 2009). The similarity of the single-model and multi-model spreads suggests that a large fraction of the CORDEX ensemble spread can be explained by internal variability, despite the fact that it was not explicitly sampled within the CORDEX framework (where most models provided a single simulation, von Trentini et al., 2019)."

ll. 95-96 I am not sure that I agree with the conclusion, that "the most important" modes of climate variability are captured by the ClimEx model, as this conclusion is here based on a comparison to another model ensemble. I agree that it is reasonable to assume from this comparison that the ClimEx model produces reasonable climate variability, but I do not think such a comparison warrants a judgment on which mode of variability is important or not. Please consider rephrasing.

*We agree that the focus of this paragraph should not be set on a judgement of importance of modes. We rephrased the statement:*

l. 103-104: "Therefore, the GCM and RCM ClimEx ensemble can be expected to capture the range of winter temperature and precipitation internal variability despite the set up with a single model."

ll. 100-103 The most commonly used acronyms for sea level pressure and surface air temperature are SLP and SAT, respectively. Why did the authors decide to use different abbreviations? This is not a huge issue, but interrupts the flow when reading. Also, t2m and tas are usually not the same in model output. The manuscript would benefit from clarification as to which of the two is used in this study – this is currently not clear.

*Thanks for this hint. We changed the names from psl → SLP, tas → nSAT (near surface air temperature) and pr → PR whenever the variable is meant. Table 1 introduces the model output variable names, which is why we kept psl, tas, pr etc. in there.*
*In CanESM2 and CRCM5 "tas" refers to near surface air temperature, and ERA-I variable "t2m" is 2-m temperature. We assumed that t2m is the ERA-I variable that is most similar to the model variable. We also placed an explanation in the manuscript:*

l. 114-115: "ERA-I variables t2m, tp and msl were chosen as they were assumed to most accurately represent the variables from the GCM and RCM models."

l. 120 The text says that there are two regions of interest, while table 2 specifies seven regions and the remaining manuscript references those seven regions. I suggest omitting the "two regions" phrase, as it is more confusing than helpful at this point.

*Originally, the "two regions" in this phrase refer to the NAO formation (1) and response (2) regions. We agree that the mention of seven analysis regions in Table 2 does not fit the "two regions" phrase, so we removed it.*

*Following a suggestion of Referee Comment 2, we replaced Table 2 with a labeled map (Fig. 1) indicating the size and position of the regions of interest:*

[Figure]

*Figure 1: Regions of interest. Abbreviations and domain sizes in terms of GCM grid cells (2.8°) are as follows: AH – Azores High (3×3); IL – Icelandic Low (3×3); NAR – large-scale North Atlantic region (28×16); CEUR – Central Europe (5×5); NE – northern Europe (1); BY – Bavaria (1); SE – southern Europe (1); ClimEx – domain used in ClimEx project (extent approximately 22×12 after resampling to GCM grid).*

l. 140  The authors use past tense to describe the present study here, and this appears to be the dominant choice of tense. Elsewhere, however, present tense is used (e.g. l. 120 "...there are two separated regions..."). This inconsistency can be found throughout the entire manuscript. To improve readability, I suggest the authors decide on one tense and stick to it throughout the manuscript.
*Thanks, we now use past tense.*

ll. 141-142  The word "representative" is lacking a reference here. The 30-year time horizon leads to an NAO distribution that is representative of what? Please elaborate briefly.
*As stated in the sentence before, major fluctuations of the natural climate system on several temporal scales are assumed to be included in the 30-year time horizon. Their potential influence on the NAO may thus be seen as represented within the sampled NAO time series. We rephrased the paragraph accordingly:*

l. 156-159: "This study focused on inter-annual analyses which were conducted for two time horizons covering 30 years each. The chosen period length was assumed to include major fluctuations, like internal climate variations or several solar cycles, which might affect NAO phases (Andrews et al., 2015). Thus their influence can be assumed to be represented by the sampled NAO time series."

ll. 144-145  This is an important caveat. I like that this is mentioned here, but missed it in the discussion section. I suggest taking it up again there to make sure this (perfectly acceptable) limitation of the study can be appreciated.
*We included a paragraph in the discussion section, which refers to this limitation.*

l. 498-502: "It has to be added that this study evaluated two 30-year blocks rather than continuous time series, treating the NAO–response relationship as stationary during these blocks such that the IMS of both periods represents generalized conditions for 1981–2010 and 2070–2099. According to Comas-Bru and McDermott (2014), potential non-stationarity in NAO–response relationships can at least partly be attributed to influences of the EA/SCA patterns on the NAO, and especially the geographical position of the North Atlantic SLP gradient."

ll. 150-154  I think this bit would be easier to understand if the order of the phrases was altered to first explain why March can be included and then say that DJFM is used for winter. Please consider making this change.
*Thanks, we changed the wording:*

l. 169-174: "Since the NAO is known to be strongest in winter (Hurrell and Deser, 2009) and the connection between station-based indices and NAO responses tends to be best in winter (see Pokorná and Huth, 2015, for months DJF), analyses were performed for this season only. Preliminary tests had shown that correlations and links between the NAO index and the climate variables were more distinct from noise, if March was included as well. That is why an extended winter season was used here (DJFM, see also Iles and Hegerl, 2017; Hurrell, 1995; Osborn, 2004)."

ll. 159      I suggest refraining from the statement that a station-based NAO index is "easy" to interpret – its reference is arbitrary (easy for whom?) and it is not a very scientific expression. Please rephrase.
*We agree that "easy to interpret" is not an appropriate expression in this context. We rephrased the paragraph:*

l. 176-181: "The NAO index was derived from ERA-I and CanESM2-LE data, resulting in 1 REF and 50 GCM realizations. The NAO is quantified in this study with an index which is closest to a station based or zonally averaged index. This allowed obtaining an index in a large data set (50 members during hist and fut time horizons) at justifiable computational time. Other than indices based on PCA, this index does not represent a "pure" NAO pattern, i.e. the variability of North Atlantic SLP without any other teleconnection patterns like the East Atlantic Pattern (EA) and the Scandinavian Pattern (SCA) (Moore et al., 2013). Instead, it directly represents the winter SLP gradient over the North Atlantic."

ll. 189-195      This section appears to already present results. Please consider moving it to the results section.
*This paragraph was included to explicitly mention the way internal variability was addressed in this study. It was not intended to present results. We included some more information to enhance its relevance.*

l. 228-237: "Internal variability was understood as being represented by the oscillations around the long-term mean of the time series of a given variable (Hawkins and Sutton, 2011). In this point of view, IMS of the LE originates from the superposition of all 50 realizations with their respective inter-annual variability. As the climatic evolution of all 50 members is equally likely by construction of the ensemble, this spread represents an envelope of possible sequences of weather events at any given time step or location. This allows to sample internal variability at single points in time as the range of the members' values.
Therefore, the NAO–response relationship was analyzed individually for each GCM and RCM member (as is done e.g. in Woollings et al., 2015).
Aggregations to ensemble means (like in Deser et al., 2017) and standard deviations (sd, see also Leduc et al., 2019; Déqué et al., 2007), the latter representing the IMS in maps, were only performed for illustrating purposes in order to avoid masking model internal variability (Zwiers and von Storch, 2004)."

l.200      In lines 97-98, the authors define REF as the ERA-Interim data set. Here, REF appears to refer to the NAO index within the ERA-I data set. Please define REF only once and unambiguously.
*Yes, reference/REF is defined to be anything derived from the ERA-I data set, but this sentence uses REF confusingly. We omitted the part "REF". Two sentences later, the NAO index derived from ERA-I is defined as being "a reference" for the rest of the study.*

l. 244: "First, a NAO index was calculated from the ERA-I reanalysis.
l. 246-247: "For further analyses it will therefore serve as a reference."

ll. 205-206      I am not sure I agree that figure 1a shows that REF (the blue bars) lies "comfortably" within the ensemble spread (grey & red). Particularly negative extremes, but to some degree also positive ones, seem to be underrepresented in the model. Can you please comment on this and possible implications for this study?
*The sentence in the discussion paper refers primarily to the x-axis of the histogram in Fig. 1a, not the frequency of occurrences: The index values of the ERA-I NAO index may be found within the CanESM2-LE, that is, between the minimum and maximum LE index values. It is true, that the distribution of ERA-I index values shows differences towards the distribution of the CanESM2-LE. These differences may partly be explained by different sample sizes ($n\_ERA-I = 30$, $n\_CanESM2-LE =1500$); the ERA-I sample is only one realization which is compared with the mean of 50 ensemble realizations, so deviations between the distributions may occur.*
*We changed (old) Fig. 1 (see Fig. 2), using CDFs rather than histograms as this removes the problem of binning and also represents the different sample sizes (see smooth curves for CanESM2 as opposed to steps for ERA-I)*

[Figure]

*Figure 2: Cumulative density functions (CDFs) of NAO index values. (a) distribution of all CanESM2-LE (n = 50 × 30 per period) and ERA-I (n = 30) NAO index values. Black: 1981–2010 CanESM2-LE, red: 2070–2099 CanESM2-LE, blue: 1981–2010 ERA-I.*

*In addition, we complemented the corresponding paragraph:*

l. 248-253: "The CanESM2-LE produces NAO index values which follow a distribution similar to the ERA-I data (centered over zero, slight surplus of low positive NAO values, see Fig. 3 (a)). The CanESM2-LE distribution appears smoother due to a larger sample size (n = 1500 for CanESM2-LE and n = 30 for ERA-I). Maximum and minimum index values (x-axis in Fig. 3 (a)) of some of the 50 members exceed those of the REF realization; thus, the REF realization lies well within the ensemble IMS. The future NAO index shows a similar distribution of values, but with slightly less positive and more negative values (red curve in Fig. 3 (a))."

l. 214          *"…original data into three subsets…"*
l. 214          Please consider changing "indifferent" to "neutral" or "average" here and throughout the document.
*We adopted both points, changing "Indifferent" to "neutral".*

l. 214          Are the "average psl conditions" referenced here the same as the "MSLP mean" in figure 2? If so, I highly recommend using coherent names (i.e. "mean" or "average" in both cases) to avoid confusion. I had to read this paragraph several times before I understood it.
*Yes, both refer to the same, i.e. neutral SLP conditions. We changed the wording in both text and header (see Fig. 3 below).*

ll. 216-217          Which difference is referenced here? Also, what do over- and underestimation refer to? If this is based on a comparison of figs. 2a and d, I cannot follow the argumentation – actually, it appears to me that the model overestimates mean SLP over the North Sea and underestimates SLP over Greenland. Can you please clarify?
*"Difference" refers to the mean SLP difference between CanESM2-LE and ERA-I (old Figs. 2a and 2d, respectively, see Fig. below). SLP over Greenland rises to about 1025 hPa in CanESM2-LE and about 1015 hPa in ERA-I data (hence overestimation in CanESM2-LE with respect to ERA-I; see yellow circles in Fig. 3); over the North Sea, SLP reaches 1000 hPa in the CanESM2-LE and 1010 hPa in ERA-I (hence underestimation in CanESM2-LE with respect to ERA-I; see red circles inserted in Fig. 3 below).*
*We clarified the wording and changed the coloring of the SLP maps to better visualize the differences.*

[Figure]

Figure 3: NAR winter mean SLP [hPa] composites in REF ((a)–(c)) and GCM ((d)–(i)) data showing long-term neutral conditions (left column), NAO positive (mid column) and negative anomalies (right column). (a)–(f): for 1981–2010. (g)–(i): 2070–2099 changes with respect to 1981–2010 in GCM data. White isolines: difference between positive and negative anomalies by a step of 2.50 hPa, as e.g. in Hurrell (1995), solid: positive, dashed: negative, bold line: zero. Stippling in subpanels (a)–(f): regions where the anomaly is smaller than the standard error of the composite samples. Black boxes: AH, IL and CEUR regions (see Fig. 1).

*The correspondent paragraph says now:*

l. 272-276: "Under neutral NAO conditions, the North Atlantic region is characterized by a pressure dipole. This structure is intensified and tilted clockwise in the CanESM2-LE ensemble mean (middle row of Fig. 4) compared to REF (top row). The mean SLP difference between the CanESM2-LE mean and REF reaches up to 10 hPa in both directions. SLP values are higher over Greenland and lower over the North Sea in the CanESM2-LE compared to ERA-I (compare subpanels (a), (d) in Fig. 4)."

l. 218     *"...phases also show less pronounced..."* Weren't the anomalies more pronounced in the model than in REF for the mean state? If so, please omit the "also".
*That is true, we removed the "also".*

ll. 239-240     *"...the spatial patterns of ERA-I and CRCM5/ERA-I differ **more** strongly than in Fig. 3,..."*
*We corrected the sentence.*

l. 241     What is the reference for the "more humid conditions"? The lack of a reference for relative statements is an issue that needs addressing throughout the manuscript.
*In this case, the reference is the neutral NAO state. We rephrased this sentence:*

l. 321-322: "In comparison to the neutral state, positive phases are also accompanied by more humid conditions in the north, and drier conditions in the south of the CEUR domain (see Fig. 8)."

l. 256          The NAO explains less variance than what?
l. 257          tas std decreases less than what?
l. 259          While I am sure the inconsiderable change of spatial patterns compares the historical to the projected period, I think it would help to give this information here again.
ll. 259-260     Could you please give a figure reference for the claims made here?

*In all three cases (l.s 256, 257, 259), we compared the historical and future time horizons. When rewriting (and shortening) the paragraph, we included the necessary references and the figure reference. However, the sentences regarding comments l.256 & 257 were removed from the manuscript as the information is already included in the previous sentences.*

l. 335-338: "Future NAO–climate relationships weaken in general compared to the historical ones for all variables. The spatial patterns of NAO-induced change do not change considerably between both periods. The response to the NAO, $\alpha_1$, is clearly reduced in nSAT mean as is nSAT std, and there is also a reduction in PR sum change (panels (g), (i) in Figs. 6–8)".

l. 264          Is there a particular area for which the transfer of internal variability from GCM to RCM is assessed?

*We assessed the "transfer" in the response regions – that is, spatially explicit in CEUR (see sd maps/subpanels (d), (f) in Figs. 6—8) and spatially aggregated in NE, BY, SE (see Fig. 10). We inserted a short note on this matter:*

l.341-342: "The representation of internal variability in the GCM and RCM regarding the responses to the NAO in CEUR and subset regions NE, BY, SE is assessed via differences in the IMS of the CRCM5-LE compared to the CanESM2-LE."

l. 277          If large tas deviations do not correspond to high or low $\alpha$, what do they correspond to?

*Thank you for this question; this sentence is not as clear and detailed as it should be. We included scatterplots regarding the relationship between LE ensemble means and sd, showing also several correlation measures (rather than just correlations as suggested in the final response; see Fig. 4).*

*Since in this case it is of no importance whether $\alpha_1$ is positive or negative –we are interested in whether a strong response is related with a large inter-member spread – only absolute $\alpha_1$ ensemble mean values were used.*

*The scatterplots indicate that particularly for nSAT std, the relationship may be seen as linear (the two clusters in the GCM data are probably due to the small domain size). nSAT mean and PR sum do not show a clear linear relationship in the GCM data.*

*RCM data shows more linearly oriented point clouds during the historical period for both nSAT variables, but a decrease in the correlation during the future period, as well as lower ensemble mean and sd values for both nSAT variables. PR sum though shows some higher ensemble mean and sd values.*

[Figure]

*Figure 4 : Relationship between LE mean and sd values for variables nSAT mean (a)–(d), nSAT std (e)–(h), PR sum (i)–(l) for hist and fut time frames. Upper right corner: r – Pearson correlation coefficient,  – Spearman rank correlation coefficient,  – Kendall's Tau.*

*We also updated the corresponding paragraph:*

l. 354-361: "Largest deviations for nSAT mean are found in continental regions of CEUR, but they do not simply correspond to high or low α1 (see also Fig. A3 (a)–(d)). Low IMS corresponds mostly to Alpine and sea regions. For nSAT mean, the signal-to-noise ratios (SNR) between ensemble mean and sd exceed 1 in most regions north of the Alps (see regions without stippling in Fig. 6). nSAT std shows SNR < 1 in the northern parts of the CanESM2-LE data (see Fig. 7 (c)) and in the Alpine region of the CRCM5-LE data (Fig. 7 (e)). This variable shows a strong linear relationship between LE mean and sd (Fig. A3 (e)–(h)). Regarding PR sum, RCM members vary most in regions with highest absolute α1 values and altitudes, but there is no clear dependence in GCM (Fig. A3 (i)–(l)). For PR sum, there is an east-west corridor of SNR values below 1 which accompanies rather low α1 values (see Fig. 8)."

l. 284    I find the presentation of this reference to figs. 3, 4 and 5, h & i ambiguous. Do you refer to panels h & i of all those plots, or just 5?
*Yes, you are right; the reference is ambiguous. Looking in figs. 3-5, we also noted that there is a mistake; it should be (j), not (i). We changed the reference to "panels (h), (j) in Figs. 3-5". It now reads:*

l. 361-363: "In addition to future changes in the NAO responses ensemble means, there is also a change in the spatial distribution of the IMS expressed as ensemble sd (see subpanels (h), (j) in Figs. 6—8)."

ll. 301-302    This sentence is difficult to understand due to the many parentheses and different references therein. I highly recommend splitting this sentence in at least two.
*We reordered the sentence:*

l. 385-387: "Apart from PR sum in the NE region (both time horizons), no significant difference between the spread amplitudes of GCM and RCM is visible (p ≤ 0.05, F-test)."

 I think the "matching subset region time series" warrant a more detailed explanation. As it stands, I am not sure what these are and how to interpret them. As a result I cannot follow the text. Please introduce this metric at least shortly.

*This is a good point. The idea was to compare the variability of nSAT mean, PR sum and nSAT std time series of the CRCM5 with the CanESM2 in the subset regions NE, BY, SE. Therefore, we correlated the time series of, e.g. nSAT mean, derived from the spatially aggregated subset region in CRMC5 with the time series derived from the CanESM2 subset region. These correlations were calculated member-wise, leading to 50 correlation coefficients per subset region. High (low) correlation coefficients indicate a strong (weak) co-variability of the CRMC5 and CanESM2 in the respective member. We added an explanation to the manuscript.*

l. 389-390: "To evaluate the co-variability of CanESM2 and CRCM5 data in the subset regions, time series of the response variables originating from both data sources were correlated member-wise (see Fyfe et al., 2017, for a similar approach)."

ll. 308-309 I am not sure I fully agree with this statement. While correlations indeed appear to be generally lower for pr sum (fig. 8b), 1/3 regions for tas mean (fig. 8a) and 2/3 regions for tas std (fig. 8c) show an increase towards the later period. I think the manuscript could benefit from a more detailed discussion here.

*These findings are certainly true. We included a more detailed and revised paragraph regarding this issue.*

l. 390-400: "As can be seen in Fig. 11, highest accordance on average is reached for nSAT mean in both periods, indicating that CanESM2-LE and CRCM5-LE show very similar temporal variability for this variable. The co-variability of GCM and RCM time series is weaker for PR sum and nSAT std than for nSAT mean in both periods. Also, the IMS is larger for PR sum and nSAT std than for nSAT mean. This finding suggests that there is a larger discrepancy in portraying PR sum and nSAT std in the RCM with respect to the GCM compared to nSAT mean. The correlations between CanESM2 and CRCM5 subset regions are in general significantly lower under future climate conditions compared to the historical ones, apart from nSAT mean in BY, PR sum in SE and nSAT std in BY (see text in Fig. 11). For nSAT std a shift of the distribution of r towards slightly larger values is visible. All variables exhibit a future IMS increase, though not all subset regions are affected (see e.g. nSAT mean BY or nSAT std SE in Fig. 11). This suggests that under future climate conditions a considerable reduction of GCM–RCM co-variability needs to be taken into account, at least for PR sum and (weaker) for nSAT mean. "

ll. 309-310 I do not quite understand the last sentence of the "results" section. As a result, I struggle to see what its consequences are. I recommend adding some more explanation here, as this might be a crucial point.

*The last sentence is not as precise as it should be. The results presented in (old) Fig. 8. suggest that there is a larger discrepancy in portraying PR sum and nSAT std in the RCM with respect to the GCM than for nSAT mean. We addressed this issue when rewriting the entire paragraph (see response to previous comment).*

ll. 314-315 What does it tell us that one realization shows a good correlation to REF? Why are the two so highly correlated? I am not sure why this is mentioned here. As in the introduction, this (apparently) irrelevant information might cause the reader to loose track of what is important. Please consider omitting this sentence or, if you deem it relevant enough, elaborate to illustrate its relevance.

*This realization was mentioned to show that the ensemble may incidentally produce very "realistic" looking realizations. However, we agree that it might seem irrelevant and distracting, so we removed the sentence.*

l. 316 It is not clear about which strong psl gradient the authors are writing here.

*Yes, this information is missing here. We refer to the SLP gradient over the North Atlantic within the CanESM2 under neutral SLP conditions as seen in (old) Fig. 2 (d). The sentence was updated with the corresponding information:*

l. 415-416: "The strong SLP gradient under neutral NAO conditions over the North Atlantic noted in the CanESM2-LE though suggests an overestimation of the local atmospheric circulation with too strong westerlies."

ll. 318-319 NAO+ and NAO- are weaker within CanESM2-LE than which reference?

*The reference (which is indeed missing) is the ERA-I data set. However, when rewriting the discussion, this sentence was removed as the information is already provided in the results section.*

 The very limited sample size of n=7 (or rather n=3 and n=4) in REF is an important issue that is worrisome. It should be discussed further! How robust are the results presented here? What could maybe be learned about observations from the model?

*We agree that the small sample size is problematic. We conducted an uncertainty assessment of the samples for positive and negative NAO composites (referring to Fig. 2, panels (b)-(c) and (e)-(f)). Therefore, we estimated the standard error of the arithmetic mean on each grid cell for the ERA-I data and compared it with the CanESM2 samples (which are considerably larger). We included stippling in Fig. 2 (a-f, only visible in b-c, e-f) to show where the signal is larger than the standard error.*
*The results section was updated accordingly:*

l. 276-285: "Long-term neutral states of both data sources show robust signals in the entire NAR region (i.e., no stippling). This suggests that the different patterns in GCM and REF data are not singularly artefacts arising from different sample sizes. The GCM multi-member composites of positive and negative phases show less pronounced SLP anomalies than the REF data. This difference between GCM and REF may be due to the fact that REF composites were derived from n = 3 negative and n = 4 positive years whereas the GCM data provided n = 264 negative and n = 263 positive years during 1981–2010. Regions with strong sampling uncertainties, i.e. where the standard error is larger than the anomaly, are indicated with stippling in panels (a)–(f). These regions are mostly found in the transition region between the wider ERA-I AH and IL nodes, whereas the SLP anomalies at the NAO centers of action show less uncertainty. The GCM patterns are more robustly assessed (i.e. less prone to sampling uncertainty) as can be seen by the very small area with stippling in which the sign of the anomaly may not be assessed robustly."

*However, it may be difficult to learn about observations from the model. Learning from the model about observations would imply that the model internal variability can be seen as "correct" as the observed internal variability which is not easy to estimate since there is only a single realization of observations.*

 At this point, I somewhat expected a discussion on the influence of other teleconnection patterns. I think the authors should at least provide some indication (from the literature) about how large these teleconnections' influence on this study can be expected to be.

*Following this suggestion, we included a short survey on the influence of the East Atlantic Pattern and the Scandinavian Pattern, as we based our NAO index on the SLP gradient over the North Atlantic, which occasionally is affected by these teleconnection patterns (see Moore et al. 2013 and Comas-Bru and McDermott 2014).*

l. 417-424: "Since the NAO index was obtained from raw SLP data, it contains the contribution of the NAO, but possibly also of micro-climatic noise or other teleconnection patterns like the East Atlantic (EA) and the Scandinavian Pattern (SCA) which interact with the NAO and exert a notable control on the North Atlantic SLP gradient (Moore et al., 2013). Moore et al. (2013) investigated the contributions of the North Atlantic teleconnections NAO, EA and SCA in reanalysis data by separating them with empirical orthogonal functions. The authors found that the NAO accounts for about one third of winter SLP variability, and the second and third leading modes for roughly 20 % and 15 %, respectively (see also Comas-Bru and McDermott, 2014). Thus the results presented here may be seen as representing the superposition of these atmospheric modes."

 The latter is not as clear in the chose domain as what?
*We rephrased the sentence:*

l. 410-411: "The first is found in the here presented results, the latter is not clearly visible in the chosen domain."

 I think the observation is missing a reference in this sentence: Is it NAO+ or NAO-? And are these observations derived from reanalysis or the literature or a model? As it stands, this is quite ambiguous.
*We agree that this sentence is ambiguous. It is meant to refer to the fact that the Jetstream position it altered during the NAO+/NAO- phases and therefore associated air mass advection is displaced (see e.g. Woollings et al. 2015). However, as we do not further refer to the Jetstream in the text and thus the sentence does not add to the argumentation in the discussion, we removed it in the revised manuscript.*

 Omit the comma between "region" and "which".

*Thanks, we removed it.*

 This is an intriguing thought. What are its consequences/implications? Please consider to elaborate a bit.

*The GCM reproduces strongest variability in (geographically) other regions than ERA-I, but in the RCM the positions are "correct"; so for example, we may also see added RCM value for regional scale analysis in this. We included a more detailed paragraph:*

l. 456-469: "Apart from the coarser pattern resolution, there is also a shift in the spatial climate patterns in the CanESM2-LE within the CEUR domain with respect to ERA-I data which is not found in the CRCM5-LE: for example, typical continental climate features, such as high nSAT variability (as indicated by Fig. 6), are shifted southwards in the CanESM2-LE with respect to CRCM5-LE data (or ERA-I). This shift may be explained by the fact that due to coarser spatial resolution the GCM topography shows land grid cells where the Mediterranean or the Baltic Sea extend in ERA-I and CRCM5; thus, in the GCM, the continent Europe also occupies a region which is sea in ERA-I. Assuming that the land–sea distribution affects the climate evolution, the GCM also experiences a geographical shift of climatic characteristics (such as continental properties) compared with the ERA-I and RCM data within the study domain. Another example is the dividing line for NAO–PR sum relations (see Fig. 8) which shows a displacement in the GCM compared to the RCM. This displacement is related to the GCM orography which deviates due to the coarser spatial resolution in shape, position and height from the RCM orography. These findings suggest that similar responses of GCM and RCM to the NAO may not be visible at the same geographical location (i.e. coordinates), but under similar geographical conditions (exposition, altitude, distance to sea). Continuing this thought, the RCM reproducing the spatial climatic patterns in the "correct" location is another expression of the RCM added value for regional or local scale analyses. However, for general statements on this issue, analyses on a larger domain would be necessary."

 What does it mean for the findings presented here that the GCM overestimates T and pr? Does this limit the conclusions that can be drawn?

*This information is given as background information. We included the correspondent information in the data section. The overestimation of average nSAT and PR does not affect the findings regarding the correlation coefficients since these are based on the changes/variability, rather than on background temperature/precipitation. However, the large discrepancies among local α1 between the GCM and RCM maps show a clear resemblance with the background nSAT and PR fields.*

l. 118-124: "Figure A1 shows that the CRCM5 tends to underestimate (overestimate) mean winter nSAT mean in the northern (southern) part of the domain, regardless of the driving data (see first column for ERA-I and third column for CanESM2), whereas winter PR sums are overestimated in nearly the entire domain with strongest values in the south-eastern part. The GCM overestimates (underestimates) nSAT mean north (south) of the Alps. PR sum is underestimated in the entire domain apart from the western side of the Alps in the GCM. However, as this study will focus on changes in nSAT and PR induced by the NAO (see Section 2.2.4), biases are of no large relevance in general, but may show some influence when it comes to regions with particularly high PR sum values."

l. 327-332: "In the CRCM5-LE, single spots in mountainous regions (e.g. in the Dinaric Alps) show extremely high PR sum α1 values (up to ±220 mm per unit index change) where long-term mean PR sums are also very high. This stresses the more detailed production of geographical features, but also the tendency to evolve local extreme values in the high-resolution RCM (see similar results for local daily extreme precipitation in Leduc et al., 2019) which may even be noted in the (spatially aggregated) bias towards the GCM (see Fig. A1 (f))."

 Since the patterns are "only" very similar, I find the statement "atmospheric dynamics are correctly implemented" a bit too strong. Please consider rephrasing to, e.g., "...can be regarded as correctly implemented".

*Thanks, we rephrased the statement accordingly.*

l. 478-480: "However, as the ensemble mean (GCM and RCM) reproduces patterns very similar to the observed ones, the atmospheric dynamics behind can be regarded as correctly reproduced in all members."

 As stated before (comment lines 205-206), I do not agree that the observations lie comfortably within the model spread, so I also have an issue with the statement "...the same climate statistics". Please either explain where I went wrong or rephrase.

*Thanks for your concerns. We agree that "the same climate statistics" sounds too strong. As shown in (old) Fig. 6, the CanESM2 ensemble generally encompasses the REF realization regarding several statistics, e.g. inter-annual variability or number and mean values of positive/negative phases. When rewriting the discussion section, we removed this sentence, but the information about comparable characteristics is mentioned in the results and conclusions section, e.g.:*

l. 267-268: "The members also show no systematic correlation with the REF NAO index despite similar statistics (see also Fig. 9)."

l. 538: "The ensemble also shows comparable climate statistics with the REF time series and patterns."

 Maybe rephrase to *"...with highest change in CRCM5-LE, but not **necessarily** in CasESM2-LE."*?

*Thanks for this suggestion. However, this sentence was removed as its information is already implicit in section 3.2.1:*

l. 354-361: "Largest deviations for nSAT mean are found in continental regions of CEUR, but they do not simply correspond to high or low  $\alpha1$ (see also Fig. A3 (a)–(d)). Low IMS corresponds mostly to Alpine and sea regions. For nSAT mean, the signal-to-noise ratios (SNR) between ensemble mean and sd exceed 1 in most regions north of the Alps (see regions without stippling in Fig. 6). nSAT std shows SNR < 1 in the northern parts of the CanESM2-LE data (see Fig. 7 (c)) and in the Alpine region of the CRCM5-LE data (Fig. 7 (e)). This variable shows a strong linear relationship between LE mean and sd (Fig. A3 (e)–(h)). Regarding PR sum, RCM members vary most in regions with highest absolute $\alpha1$ values and altitudes, but there is no clear dependence in GCM (Fig. A3 (i)–(l)). For PR sum, there is an east-west corridor of SNR values below 1 which accompanies rather low correlation values (see Fig. 8)."

 Less tas and pr variation is explained by NAO than by what?

*"Less" is referring to a comparison between historical and future time periods. We rephrased:*

l. 527-529: "As less nSAT and PR variance is explained by the NAO in the future climate projections than in the historical period, the influence of this climate mode on CEUR climate may be seen as potentially reduced."

Conclusions   I think the reference to the questions raised in the introduction could be made clearer. While the references are there, I think it would make this part clearer if it was structured in bullet points, like the questions raised in the introduction. Please consider making this change.

*Thanks for this idea. We put the answers to key questions (a)-(d) in bullet points.*

l. 535-552: "
  (a) Both large ensembles within the ClimEx project climate model chain are able to depict a robust NAO pattern under current forcing conditions. Each member represents a distinct climate evolution while sharing comparable statistics with all other 49 realizations and producing NAO and response patterns that are more robust than patterns of single realizations. The ensemble also shows comparable climate statistics with the REF time series and patterns. **The clearly visible connection of the NAO with nSAT mean and PR sum follows well-known patterns. The influence of the NAO on nSAT variability, as expressed by the analyses on nSAT std, is also remarkable**.
  (b) The RCM is able to reproduce the large-scale SLP pattern and realistic response patterns in the analyzed domain. Clearly more topographic features are visible in the CRCM5-LE than in the CanESM2-LE which **suggests** added value of the RCM regarding the evaluation of small-scale NAO impacts. Deviations of nSAT and PR responses between members vary spatially within the domain and are found mostly in regions with strongest NAO responses.
  (c) Internal variability of the NAO pattern is expressed very well within the 50 member single-model ensemble, and easily spans the observations regarding various indicators. The range of NAO responses is represented consistently between the driving GCM and the nested RCM. The spread is shifted towards stronger NAO–nSAT/PR relations in the RCM compared to the GCM in both time horizons.
  (d) Concerning climate change, several changes go hand in hand: the winter index variability is reduced, the overall winter variability of nSAT and PR and also the fraction of NAO-explained nSAT is reduced,

the relationship between NAO and response variables is weakened, the RMS* error regarding the large-scale SLP pattern between GCM and RCM slightly increases, and the co-variability of CanESM2 and CRCM5 subset regions for all weather variables is reduced.

ll. 397-399       This is a long sentence that is hard to understand because it takes up two different points. Please consider splitting the sentence in two.
*Thank you; we considered this suggestion. Please have a look at the response to the previous comment (bullet point (a), sentences in bold).*

l. 404          I find the word "proves" very strong. I agree that the clearly visible topographic features are nice to look at and encouraging for the model presented here, but I disagree with the notion that the mere notice of more pronounced topographic features "proves" the added value of anything. High resolution does not always equal added value. Please rephrase.
*Thanks for this concern. We agree that "proves" sounds rather strong. We rephrased to "suggests" (also in bold in response to the comment regarding the conclusions, bullet point (b)).*

Fig. 2 caption     *"(g)-(i): 2070-2099 changes* **with respect to** *1981-2010"*
*We adopted the suggestion.*

Figs. 3-5 caption  What are the correlations show in blue isolines? What is correlated to what? Also, this is a confusing figure, partly due to the ambiguous headers for the subpanels (which are identical for, e.g., c and g). Please think about a more intuitive way to convey this very interesting information.
*The blue isolines corresponded to lines of equal correlations between the NAO index and the nSAT mean/nSAT std/PR sum time series on the grid cells by increments of 0.1. We agree that the bare presentation of blue isolines is rather confusing. We changed the increments to 0.25 (in order to picture less lines), and indicate the correlation strengths by different grey scales (and a legend). We think that figures 6—8 gain more clarity in doing so. Also, headers and captions were adjusted. See the following Fig. 5 as an example (the same changes were applied to the plots for nSAT std and PR sum).*
*We also included the information from former appendix Figs. A3-4: regions with signal-to-noise ratio < 1 are indicated by stippling.*

[Figure]

Figure 5: Spatial patterns of change in nSAT mean (in [K]) for a unit change in the NAO index for ERA-I, CRCM5/ERA-I, CanESM2-LE and CRCM5-LE in 1981–2010 ((a)–(f)) and the difference of 2070–2099 with respect to 1981–2010 ((g)–(j)). Both 50-member ensembles are represented with ensemble mean and sd representing the IMS. Grey lines in the ensemble mean maps represent the Pearson correlation between nSAT mean and the NAO index at an increment of 0:25; grey shadings see legend in upper left panel. Grey stippling in the ensemble mean maps show regions were SNR < 1, SNR being the signal-to-noise ratio between the 30 year ensemble mean and sd of GCM and RCM LEs in both time periods.

Fig. 6          Some of the indices named in the upper left corner have slightly different names than those found on the x-axis. It could help the clarity of the (otherwise very nice and interesting!) figure if those names were the same. Please consider changing the figure accordingly.

*Thank you. We corrected the names:*

[Figure]

Figure 6: Several index statistics of all 50 CanESM2-LE members expressed as multiples of the respective ERA-I value (REF value set to 1.0): teleconnectivity (Pearson correlation between AH and IL time series), index variability (expressed as standard deviation in time of index time series), mean value of all positive (negative) phases and count of all positive (negative) phases in a single realization. Positive (negative) years are defined by an absolute index value exceeding 1. Text in upper left corner: significantly ($p \leq 0.05$, using an unpaired Mann-Whitney/U-test) different outcomes in the fut time frame.

**Fig. 8**          Please explain a, b and c in the caption. Also, I do not quite understand what is displayed. What is a "similarity of matching regions"?

*These figures display the temporal co-variability of the corresponding CanESM2 and CRCM5 members in the three subset regions (NE, BY, SE) for nSAT mean (a), PR sum (b) and nSAT std (c). Thus "matching" refers to the same member in the GCM and RCM. We included a detailed description of the metric in the text (see also response to comment line l. 306), and changed the caption accordingly:*

[Figure]

*Figure 7: Temporal co-variability of CanESM2 and CRCM5 subset regions in all 50 members. Each boxplot represents 50 Pearson correlation coefficients between the time series of variables nSAT mean (a), PR sum (b) and nSAT std (c) in the subset regions of CanESM2 members and the corresponding CRCM5 members. Time periods used for correlations: 1981–2010 (hist, black), 2070–2099 (fut, red). For regions NE,BY, SE see Fig. 2. Text denotes combinations of which the differences are significant at p ≤ 0.05 using an unpaired Mann-Whitney/U-test.*

**Fig. A2 caption**    Please explain the subpanels in the caption.

*Figure A2 shows the ratio of tas mean $\alpha_1$ and winter tas std for the data sets employed in the study: (a) CRCM5/ERA-I and (b) ERA-I under historical conditions, and CanESM2-LE ((c)-(d)) andCRCMR5-LE ((e)-(f)) under historical and future conditions. We extended the caption accordingly:*

"Ratio of nSAT $\alpha1$ and winter mean daily standard deviation of nSAT for CRCM5/ERA-I (a) and ERA-I (b) under historical conditions and CanESM2-LE mean ((c)–(d)) and CRMC5-LE mean ((e)–(f)) under historical and future climate conditions. The panels show the fraction of nSAT 1 on winter mean daily standard deviation of nSAT."

References in this response:

Andrews, M., Knight, J., Gray, L. (2015): A simulated lagged response of the North Atlantic Oscillation to the solar cycle over the period 1960–2009, Environmental Research Letters, 10.

Benedict, J., Lee, S., Feldstein, S. (2004): Synoptic View of the North Atlantic Oscillation, Journal of the Atmospheric Sciences, 61, 121–144.

Comas-Bru, L., McDermott, F. (2014): Impacts of the EA and SCA patterns on the European twentieth century NAO-winter climate relationship. Quaterly Journal of the Royal Meteorological Society, 140, 354-363.

Déqué, M., Rowell, D., Lüthi, D., Giorgi, F., Christensen, J., Rockel, B., Jacob, D., Kjellström, E., de Castro, M., van den Hurk, B. (2007): An intercomparison of regional climate simulations for Europe: assessing uncertainties in model projections, Climatic Change, 81, 53–70.

Deser, C., Hurrell, J., Phillips, A. (2017): The role of the North Atlantic Oscillation in European climate projections, Climate Dynamics, 49, 3141–3157.

Fyfe, J., Derksen, C., Mudryk, L., Flato, G., Santer, B., Swart, N., Molotch, N., Zhang, X.,Wan, H., Arora, V., Scinocca, J., Jiao, Y. (2017): Large near-term projected snowpack loss over the western United States, Nature Communications, 8.

Giorgi, F., Jones, C. Asrar, G. R. (2009): Addressing climate information needs at the regional level: the CORDEX framework. WMO Bulletin, 58 (3), 175-183.

Hawkins, E., Sutton, R. (2011): The Potential to narrow uncertainty in projections of regional precipitation Change, Climate Dynamics, 37, 407–418.

Hurrell, J. W. (1995): Decadal Trends in the North Atlantic Oscillation: Regional Temperatures and Precipitation, Science, 269, 676–679.

Hurrell, J. W., Deser, C. (2009): North Atlantic climate variability: The role of the North Atlantic Oscillation, Journal of Marine Systems, 78, 28–41.

Iles, C., Hegerl, G. (2017): Role of the North Atlantic Oscillation in decadal temperature trends, Environmental Research Letters, 12.

Leduc, M., Mailhot, A., Frigon, A., Martel, J.-L., Ludwig, R., Brietzke, G., Giguère, M., Brissette, F., Turcotte, R., Braun, M., Scinocca, J. (2019): The ClimEx Project: A 50-Member Ensemble of Climate Change Projections at 12-km Resolution over Europe and Northeastern North America with the Canadian Regional Climate Model (CRCM5), Journal of Applied Meteorology and Climatology, 58, 663–693.

Moore, G., Renfrew, I., Pickart, R. (2013): Multidecadal Mobility of the North Atlantic Oscillation. Journal of Climate, 26, 2453-2466.

Osborn, T. (2004): Simulating the winter North Atlantic Oscillation: the roles of internal variability and greenhouse gas forcing, Climate Dynamics, 22, 605–623

Pokorná, L., Huth, R. (2015): Climate impacts of the NAO are sensitive to how the NAO is defined, Theoretical and Applied Climatology, 119, 639–652.

von Trentini, F., Leduc, M., Ludwig, R.(2019): Assessing natural variability in RCM signals: comparison of a multi model EURO-CORDEX ensemble with a 50-member single model large ensemble, Climate Dynamics, 53, 1963–1979.

Woollings, T., Franzke, C., Hodson, D., Dong, B., Barnes, E., Raible, C., and Pinto, J. (2015): Contrasting interannual and multidecadal NAO variability, Climate Dynamics, 45.

Zwiers, F., von Storch, H. (2004): On the role of statistics in climate research, International Journal of Climatology, 24, 665–680.

**Responses to Referee Comment No. 2**

In this study a regional climate model (CRCM5) is employed to dynamically downscale a single global climate model (CanESM2) large ensemble of climate change simulations to investigate the nature of downscaled responses to the modeled North Atlantic Oscillation (NAO) and its influence on future European climate. By employing a large ensemble, the authors are able to evaluate future downscaled responses associated NAO inter-annual variability in addition to mean changes. The authors set out four key questions related to, documenting the properties and fidelity of the modeled NAO in both the GCM and RCM; the associated screen temperature and precipitation responses in both models; and how such properties change under future external forcings (following the future CMIP5 pathway RCP8.5).

This is an interesting paper and ultimately worthy of publication. The authors present the problem from the perspective of downscaling teleconnections that exist in the driving data (ie the NAO). This is a subtle but critically important shift in focus for the dynamical downscaling community. The proper communication of teleconnection patterns/relationships from driving data to the RCM is essential for credible downscaled results. The use of a large GM/RCM ensemble pair positions the authors to say something definitive about this problem and offer guidance to the community.

The four key questions represent a clear and sensible plan for the paper. However, I found it difficult at times to cleanly connect a particular analysis performed by the authors with an answer to some of these questions. Specifically, I do not think that the authors addressed the first part of their question 3, "Do GCM NAO impulses propagate correctly into the RCM realizations" (l. 71). Perhaps a better way of stating this is, does the RCM faithfully represent the NAO pattern present in the driving data? This is a critical question in the authors' "model chain" (l. 65) that needs to be addressed before one moves on to evaluate the NAO responses. That is, if the largescale NAO pattern is not faithfully represented in the RCM domain in some location, the downscaled responses in that location would be less credible. The increased resolution and potentially improved physical processes present in the RCM themselves cannot correct the large-scale NAO pattern within the RCM domain. As the authors discuss, the NAO pattern is governed by "planetary wavebreaking in the polar front" (Benedict et al., 2004), which is intern influenced by external factors such as sea-ice, snow cover, sea-surface temperatures, ENSO, stratospheric circulation variability, solar variability, volcanic eruptions and the Quasi-Biennial Oscillation (eg Hall et al. 2014 https://doi.org/10.1002/joc.4121).

Given that the European domain is relatively small, and the experimental design employs spectral nudging in the RCM, the NAO pattern, and its interannual variability, should on balance be reasonably represented in the RCM. For the authors' stated plan, however, this needs to be verified. Given that the authors employ a large ensemble in their study, they are in the unique position to definitively address this issue and provide an example to the community of the type of analysis that is required to support the credibility of downscaled results in such complex problems. It is my recommendation that, prior to publication, the manuscript undergo major revision to address this issue and to improve its overall clarity. My detailed comments follow.

*Thank you very much for this generally positive assessment of the study scope, but also for your concerns regarding key question 3. This question originally targeted the question whether the combination of NAO indices from the GCM and response variables from the RCM produces realistic looking NAO responses in the RCM. The suggested formulation changes its meaning towards the nesting of the NAO/SLP pattern itself. However, in light of the fact that indeed the assessment of large-scale SLP patterns in the RCM data is relevant but missing so far, this change of formulation is justifiable.*

*We adopted the suggestion in the major comment (see our point-by-point responses) and the ideas regarding different groups of correlations among member index time series.*

*Overall, we tried to optimize the structure of the paper, following the key questions raised at the end of the introduction in all following sections.*

Major Comment:

RCM reproduction of NAO teleconnection in driving data

As part of the authors' model chain, it is essential to verify that the large-scale year-to year variations of the NAO pattern in surface pressure are faithfully reproduced (each year) in CRCM5 when driven by both ERA-I and CanESM2. Inspired by Fig. 2, the sort of analysis required would be as follows:

- interpolate monthly-mean timeseries of sea-level pressure (SLP) in the driving dataset onto the RCM grid (such interpolation is already done for the driving-data winds used for spectral nudging). Call this field SLP_Drive.

- take the difference of the RCM and driving data monthly-mean SLP on the RCM grid SLP_RCM - SLP_Drive, and then smooth the result retaining large scales that are representative of the driving data resolution:

$D\_m(i,j,t,n) = [SLP\_RCM - SLP\_Drive]\_LRG$

Here, i,j are lateral spatial coordinates of the RCM grid, t is time in units of years, n is ensemble member, and the subscript m corresponds to month (1-12). The smoothing operation, represented by the operator [ ]_LRG, can be performed with the same double-cosine transform used for the spectral nudging.

- derive a normalized root-mean-square difference map for extended winter, over the two 30-year periods displayed in Fig 2, over all ensemble members:

$RMS\ (i,j) = Ave\_{(m=12,1-3)}\{Ave\_n \{ SQRT[Ave\_t \{D\_m(i,j,t,n)^2 \}] / Var\_Drive\_m(i,j,n)\}\}$

where, Ave_x is a simple averaging operators for the quantity x and Var_Drive_m is the variance in time of the driving data for each month and each ensemble member:

$Var\_Drive\_m(i,j,n) = Ave\_t \{[ SLP\_Drive\_m (i,j,t,n) - Ave\_t \{SLP\_Drive\_m (i,j,t,n) \} ]^2\}.$

Normalization by Var_Drive_m is important as it indicates the size of an rms difference relative to the interannual variability in the NAO pattern at that location. Such an RMS map would provide a sensible measure of the difference in the driving data and RCM SLP patterns associated with the NAO, which need to be faithfully reproduced in each year. If RMS « 1 at a given location, then the large-scale NAO pattern is well represented there and one can conclude that the downscaling is consistently being performed on the "correct" large-scale flow. The larger RMS is, towards O(1) values, the more suspect the downscaled responses are at that location (ie a large-scale flow disconnected from the NAO in the driving data was being downscaled in these regions).
One should also do a significance test and indicate this by, say, filling in contours by color in only those regions that are significant at the 5% level. Given the size of the GCM/RCM ensemble, this should be quite robust (ie much of the canvas should be colored) and definitive statements could be made. This test would seem to be most well posed for the case of observational driving of the RCM (ie ERA-I driving of CRCM5 over the historical period 1981-2010). The large scales in that data are well observed and, because they came from the real system, they were influenced realistically by all processes and scales. Significant deviations in RMS(i,j) for ERA-I (ie RMS_ERA-I) would necessarily indicate a degradation of the NAO teleconnection in those regions of the CRCM5 domain.
If regions of NAO deviation in RMS_ERA-I were consistent with regions of NAO deviation in RMS_CanESM2 (in the historical and even the future periods), then this would indicate a systematic issue with the reproduction of the NAO pattern in the European domain in these locations and care should be taken in the interpretation of the downscaled responses in this, and possibly other RCM studies using the same domain.
*Thanks for this very detailed suggestion! It is true that the original analysis did not include an assessment of the large-scale RCM SLP pattern. We adopted this suggestion with some slight modifications, the first one being that we interpolated the RCM data (and also the ERA-I driving data) to the GCM grid. This was done in order to not create additional errors during the interpolation onto the high resolution RCM grid. By aggregating the data, we also filtered the small scales, retaining only the large-scale patterns. We also decided to take the square-root of VarDrive. In this way, the RMS is dimensionless and may be interpreted as a root-mean squared difference. The RMS error was calculated on the entire ClimEx domain. Please see the next paragraph, which describes some insights gained from this measure:*

l. 297-308: "Figure 5 maps the RMS* of the difference between driving data and RCM SLP during 1981–2010 for driving data ERA-I (a) and CanESM2 (b) and 2070–2099 (c). A value of RMS* ≥ 1 indicates that the root-mean squared error between the RCM and driving data is larger than the temporal variability in the driving data. In this case, the large-scale SLP pattern may not be seen as being correctly represented in the RCM data. The large-scale SLP pattern over the entire ClimEx domain, which also includes the CEUR, NE, BY and SE domains, is reasonably well represented: with RMS* < 1 in most parts of the entire ClimEx domain for both driving data sets and both periods (significant at p ≤ 0.05 using a t-test with a false detection rate < 0.1 to account for

multiple hypothesis testing, see Wilks, 2016). All data sets show an RMS* increase towards the south, indicating that in these regions the control exerted by the lateral boundary conditions on the CRCM5 internal solution appears to be weaker. The RMS* is larger in the CanESM2/CRCM5 combination than in the ERA-I/CRCM5 combination, and slightly increases in the future period in the southern parts (see Fig. 5 (c)). In the CEUR domain (indicated as red box in Fig. 5), however, errors are low in general and therefore the NAO pattern of the driving data may be assumed to be correctly incorporated there."

[Figure]

Figure 8: RMS of monthly SLP differences between driving data and CRCM5 members, calculated following Eq. (2). Colouring: RMS < 1 significant at p ≤ 0.05 with a false detection rate smaller than 0.1 (see Wilks, 2017). (a) for driving data set ERA-I (1981–2010), (b) for driving data set CanESM2 (1981–2010), (c) for driving data set CanESM2 (2070–2099). Red box: position of CEUR domain.

Minor Comments:

l.2              "natural variability". Later it seems, "internal variability" (l. 16) is used to refer to the same phenomenon. It would be helpful to be consistent throughout.
*This is true. We note that terms like "natural variability", "internal variability" and "noise" were used inconsistently in the study. We fixed this issue, using "internal variability" throughout the text.*

l. 5-6.          "its transfer from the driving model CanESM2 into the driven model CRCM5." Perhaps better wording might be "its representation in the driven model CRCM5 relative to the driving model CanESM2."
*Thank you. However, when rewriting the abstract, this sentence was modified strongly. It now reads:*

l. 5-7: "The overall goal of the study is to assess whether the range of NAO internal variability is represented consistently between the driving global climate model (GCM; the CanESM2) and the nested regional climate model (RCM; the CRCM5)."

l.11             "(b) impulses from the NAO in the CanESM2-LE produce" The use of the word impulses implies causality, which may be true for the one-way nesting/spectral nudging methodology but is not for the NAO itself. To avoid confusion perhaps say, "(b) reproduction of the CanESM2-LE NAO flow patterns in the CRCM5-LE produce"
*We changed the wording in this sentence:*

l.11-12: "Reproductions of the NAO flow patterns in the CanESM2-LE trigger responses in the high-resolution CRCM5-LE that are comparable with reference reanalysis data."

l. 21            "is to apply slight differences in" -> "is to perturb"
ll.21-22         "with similar long-term climate statistics" This refers to a response rather than an experimental setup. I think it might be more correct to say "under identical external forcings"
*Thanks, we adopted the wording suggestions:*

l. 54:-56: "One way to trigger internal variability in GCM simulations is to perturb the initial conditions of the model, leading to several realizations of weather sequences under identical external forcing which also allow to derive a robust distribution of NAO index values."

l. 44            "its dynamics in a future climate" -> "its fidelity in a future climate"
*We changed the sentence accordingly:*

**21/27**

l. 42-43: "While the typical NAO pattern and its impacts are usually correctly reproduced in global climate models (GCMs) (Stephenson et al., 2006; Ulbrich and Christoph, 1999; Reintges et al., 2017), its fidelity in a future climate remains uncertain."

l.61        "is transferred correctly from the driving GCM into the driven RCM". Inter-member spread is not "transferred" from the driving model to the RCM. It would be clearer to say, "is represented consistently between the driving GCM and the driven RCM". Also, from my major comment, representation of NAO inter-member spread is a necessary condition from credible downscaled responses.
*Thanks for your explanation. We rephrased the paragraph:*

l. 67-69: "This study also targets the question, how global circulation variability, in this case the NAO teleconnection, affects local climate characteristics when downscaled using an RCM. It specifically aims at evaluating whether the range of internal variability is represented consistently between the driving GCM and the driven RCM."

ll. 65-66       "finding robust NAO patterns which exceed the uncertainty due to internal variability in the ensemble." The phrase, "exceed the uncertainty due to internal variability" is confusing in this context. Perhaps say, "finding robust NAO patterns by significantly reducing sampling uncertainty associated with internal variability"
*The suggested formulation is certainly clearer than the original one. We adopted the suggestion.*

l.64-66: "The combination of the driving GCM and nested RCM large ensembles (LE) allows for analyzing the spread of NAO states and responses within one model chain, thus establishing the range of internal variability of the NAO, and finding robust NAO patterns by significantly reducing uncertainty associated with internal variability in the ensemble."

l. 71        "Do GCM NAO impulses propagate correctly into the RCM realizations" perhaps better stated as, "Does the RCM correctly represent the NAO pattern present in the driving data" (ie my major comment)
*We agree that (old) key question (c) is better stated in this way as the suggested wording also encompasses the additional analyses regarding the large-scale SLP pattern. We changed the order of the key questions, such that new key question (b) now reads:*

l. 74-75: "Nesting approach: Does the RCM correctly incorporate the NAO pattern present in the driving data and produce realistic response patterns?"

ll. 68-74.       These are excellent focal points/topics for the paper. It would be very helpful if these were better referred back to in the analysis, discussion, and summary sections so the reader can more easily keep track of which of these you are addressing and what progress you have made on each.
*Thanks! We tried to structure the following sections accordingly. In the results section, the presentation of the results appeared to be easier when using a slightly different structure, i.e. the NAO index and spatial patterns (referring to key question (a) and partly (b)) is the first block, whereas the second block refers to the multi-model ensemble and internal variability (mostly key question (c)). (d) was found to be best presented alongside with the "hist" results.*
*In the discussion section however, results were integrated into the same structure as given by the key questions.*

ll.101-103.      two names are presented for each of three variables (eg msl/psl, t2m/tas, and tp/pr). I did not see a reason for this. If there is a reason it should be stated. If there isn't, then it would be clearer if just one name was presented for each and used throughout the paper.
*Thanks for this note. Please have also a look at the responses to Referee Comment 1 where we address a similar issue. The two names refer to different model variable output names (e.g. msl, t2m, tp were derived from ERA-I, psl, tas, pr from CanESM2 and CRCM5). We changed the analysis variable names in the following way psl → SLP, tas mean/std → nSAT mean/std, pr sum → PR sum. A short explanation was placed in the manuscript:*

l. 114-115: "ERA-I variables t2m, tp and msl were chosen as they were assumed to most accurately represent the variables from the GCM and RCM models."

ll.120-139.     It would be very helpful here to provide a schematic, say of the range/extent displayed in Fig.2, where the RCM domain is indicated and where all of the regions discussed in this section were labeled . Not until I got to Fig 2 did the layout of things become clearer to me. Even then I had to look up Leduc (2019) to understand the relative positioning of the RCM domain.

*We replaced Table 2 with a map showing all domains employed in the study. We think this is a more intuitive way to illustrate the position and extent of the domains than listing the boundary coordinates.*

[Figure]

*Figure 9: Regions of interest. Abbreviations and domain sizes in terms of GCM grid cells (2.8°) are as follows: AH – Azores High (3×3); IL – Icelandic Low (3×3); NAR – large-scale North Atlantic region (28×16); CEUR – Central Europe (5×5); NE – northern Europe (1); BY – Bavaria (1); SE – southern Europe (1); ClimEx – domain used in ClimEx project (extent approximately 22×12 after resampling to GCM grid).*

l. 200 Fig.1     This figure is very faint and it is very hard to distinguish between the three cases being presented here. The authors should work on making these results clearer by using more vivid colours and/or fills.

*Please have a look at the response to the next comment.*

l. 208     "Pairwise correlations between the members". As discussed in Leduc et al. (2019), The CanESM2-LE was spawned in 1950 from 5 independent historical realizations (separated by 150 years of coupled integration each - including 50 years of preindustrial simulation between the launch of each ensemble member). As such, each of the 5 groups of 10 are highly independent of each other. The question of independence applies to the members within each group of 10 which has only 30years of coupled integration to develop independence prior to the 1981-2010 analysis period. Wouldn't a better check of independence be to form two correlation groups? The first would involve pairwise correlations between each member and the 40 other members from the 4 other groups that were spawned from a different CanESM2 realization in 1950. This first group would form a control assumed to be highly independent. The second group would involve pairwise correlations between each member and the 9 other members of the same group spawned from the same CanESM2 realization in 1950. Plots like figure 1b for this latter group could be compared to similar plots of the control group to assess the independence of the ensemble members most likely to have residual correlations during the 1981-2010 period.

*This is a very nice idea. We performed an analysis following these steps. In order to better discriminate the different groups (and periods) we also switched from histograms to CDFs. Names of the two groups are SOIC – "same ocean initial conditions" (looking at members from the same family), and MOIC – "mixed ocean initial conditions" (looking at members from different ocean families) following Leduc et al. 2019. Also, we think that the colored lines are easier to read than histograms.*

*Referring to the independence of the members, we also included a new figure in the manuscript (see Fig. 11 below), showing the spreads among ten 5-member groups (see Leduc et al. 2019) for a daily NAO index in SOIC and MOIC groups. Groups from SOIC "start" with no standard deviations among the members. The spreads among members show no systematic differences after about a month after initialization.*

*A similar figure with winter NAO indices shows no differences in the entire period between the spreads among SOIC and MOIC members (Fig. 12 below).*

[Figure]

*Figure 10: Cumulative density functions (CDFs) of NAO index values. (a) distribution of all CanESM2-LE (n = 50 × 30 per period) and ERA-I (n = 30) NAO index values. (b) pairwise correlations among member NAO index time series from the same ocean families (SOIC – same ocean initial conditions, dotted lines, n = 225), from different ocean families (MOIC – mixed ocean initial conditions, solid lines, n = 1000) and between ERA-I and all CanESM2 members (n = 50). Black: 1981–2010 CanESM2-LE, red: 2070–2099 CanESM2-LE, blue: 1981–2010 ERA-I.*

*The text says:*

l.254-269: "For further analyses on the IMS as a measure of internal variability, the independence of the 50 ensemble members is of high importance. To investigate independence among the ensemble members in both 30 year time frames, it seems favourable to analyse pairwise member correlations. Although zero correlations do not automatically imply independence, clear correlations among embers would contradict the assumption of independence. In order to take into account the two perturbations during the production of the LE (1850 for 5 ocean families, 1950 for perturbations leading to 10 members per ocean family), these correlations were split in two groups like in Leduc et al. (2019): (i) correlations among the 10 members from the same ocean family (same ocean initial conditions in 1950, SOIC, n = 225, see dotted lines in Fig. 3(b)) and (ii) correlations between each member and the 40 members from the 4 other ocean families (mixed ocean initial conditions, MOIC, n = 1000, see solid lines in Fig. 3 (b)).

These correlations approximately follow a normal distribution with mu = 0. There is a slight surmount of low positive correlations in the SOIC group compared to the MOIC group which is (not significantly) stronger in the fut time horizon (see red and black dotted lines in Fig. 265 3 (b)). In general, the members are thus not seen as being dependent. As will be discussed below, the SLP pattern over the North Atlantic changes slightly in the future period. So the direct comparison between historical and future SOIC and MOIC correlations remains difficult. The members also show no systematic correlation with the REF NAO index despite similar statistics (see also Fig. 9). Thus, the ERA-I and GCM indices can be seen as not dependent realizations drawn from the same distribution."

*Regarding member independence, we placed a comment saying:*

l. 89-91: "Regarding the atmospheric circulation, Fig. 1 shows that owing to the chaotic nature of the atmospheric system the daily NAO 90 index seems to lose dependence from the initial conditions within the course of one month after initialization (see Leduc et al., 2019, for a similar presentation of member independence)."

[Figure]

*Figure 11: Inter-member standard deviation of a daily NAO index in the CanESM2-LE starting on 1 Jan 1950 as a function of time. The inter-member standard deviation was derived from ten groups of five members with the same ocean initial conditions (SOIC) and ten groups of five members with mixed ocean initial conditions (MOIC, following an approach in Leduc et al.,2019).*

[Figure]

*Figure 12: Like Fig. 11, but with winter (DJFM) NAO indices for the entire simulation period (1950-2100). This figure is not included in the paper.*

l. 211      "They are not systematically related to the ERA-I (the nu201creferencenu201d) realization." Why would they be? I don't understand the reasoning behind this correlation. If you are looking for a control group, a much larger group could be formed by the suggestion immediately preceding this point.

*When correlating the ERA-I realization with the 50 CanESM2 members we were not so much looking for a control group. The idea was to evaluate whether ERA-I may show dependence with the CanESM2 members (i.e. non-zero correlations). We rephrased the sentence:*

l. 267-268: "The members also show no systematic correlation with the REF NAO index despite similar statistics (see also Fig. 9)."

l. 214      "positive, negative and indifferent index values" -> "positive, negative and neutral index values"
l. 223      "it backs the choice" -> "it supports the choice"
*We changed "indifferent NAO" values to "neutral" values throughout the manuscript, and also adopted the second suggestion.*

ll.312-390      Discussion section. The references and discussion here are quite detailed and require constant back-and-forth reference to the earlier sections. For example, the opening statement of the second paragraph states, "The strong psl gradient suggests an overestimation of the local atmospheric circulation with too strong westerlies over the North Atlantic in the background state within the CanESM2-LE." What gradient? Where? The reader has to stop to review the previous sections to determine the context of this statement. This extends to the use of quantities that were defined in previous sections. For example, "Concerning NAO responses, they are most reliable in regions where r is significant (i.e. |r| > 0.361 for p nu2264 0.05,...". "r" may

have been define earlier but the reader must stop here to find where that was to understand this context. (Also "Historical nu03b11 values" l. 327.) This discussion needs to be elevated somewhat out of the details of the previous section, summarize those outcomes and their implications, and connect back to the 4 key issues outlined in the introduction.

*We agree that there is a lot "back and forth" which is related to the fact that we tried to respect the strict separation of the results and discussion sections. In order to improve readability though and following some comments of Referee Comment 1, we restructured the discussion section with respect to the four key questions (see section headers).*

l. 321            "less prone to incidental fluctuations of single realizations" -> "less prone to sampling uncertainty"
*Thanks, we adopted the correction.*

ll. 323-325        "On the other hand, lower correlation values (|r| < 0.361) suggest that climate variability at the local scale evolves differently from the global teleconnection. In these cases, the NAO is not the most important contributor and nu03b5Y in Eq. (2) is dominant. Since the index was obtained from raw psl data, it contains the NAO contribution, but possibly also of other teleconnection patterns and noise." There is also the possibility that the large-scale NAO pattern in these regions was not reproduced correctly in the RCM. See my major comment.
ll 341-343        "Another possible explanation could be that the control exerted by CanESM2 through the CRCM5 lateral boundary conditions (LBC) is insufficient, but this is unlikely given the relatively small CRCM5 domain". Adopting the suggestion in my major comment would explicitly address this key issue.
*We noted that in general the error is well below 1 in our CEUR domain, but especially southern regions exhibit a higher error compared to the rest of the entire ClimEx domain. This was also included in our discussion section:*

l. 450-453: "Nevertheless, the influence of the lateral boundary conditions appears to vary over the CRCM5 domain, being a bit weaker in the southern part. It is worth noting that this feature is less pronounced when CRCM5 is driven by ERA-I as compared with CanESM2, highlighting the importance to investigate further the interactions between global atmospheric circulation, surface forcings (e.g. topography and land-sea contrasts) and local feedbacks."

References in this response:

Leduc, M., Mailhot, A., Frigon, A., Martel, J.-L., Ludwig, R., Brietzke, G., Giguère, M., Brissette, F., Turcotte, R., Braun, M., and Scinocca, J. (2019): The ClimEx Project: A 50-Member Ensemble of Climate Change Projections at 12-km Resolution over Europe and Northeastern North America with the Canadian Regional Climate Model (CRCM5), Journal of Applied Meteorology and Climatology, 58, 663–693.

Reintges, A., Latif, M., Park, W. (2017): Sub-decadal North Atlantic Oscillation variability in observations and the Kiel Climate Model, Climate Dynamics, 48, 3475–3487.

Stephenson, D., Pavan, V., Collins, M., Junge, M., Quadrelli, R. (2006): North Atlantic Oscillation response to transient greenhouse gas forcing and the impact on European winter climate: a CMIP2 multi-model assessment, Climate Dynamics, 27, 401–420.

Ulbrich, U., Christoph, M.(1999): A shift of the NAO and increasing storm track activity over Europe due to anthropogenic greenhouse gas forcing, Climate Dynamics, 15, 551–559.

Wilks, D. S. (2016): "The stippling shows statistically significant grid points": How Research Results are Routinely Overstated and Overinterpreted, and What to Do about I, Bulletin of the American Meteorological Society, 97, 2263-2273.

**Responses to Editor Comment**

…
I had the impression that discrepancies between the model and reanalysis data seem to be interpreted in two different ways. While you argue that the difference in figure 1 can be explained by the smaller sample size in the reanalysis (response to reviewer 1, ll. 205-206), in a different response to reviewer 1 (ll. 216-217) you state that the higher MSLP over Greenland in CanESM5 compared to observations points to an overestimation in the model. Would the larger number of realisations in the model be an alternative explanation for this discrepancy as well?

*Thank you for your question. In both cases, the mean SLP and the anomalies, the size of the ensemble (or composite sample) may lead to differences. When looking at the anomaly composites though, this effect may be more relevant as the ERA-I anomaly composites have sample sizes n = 3 and n = 4 (negative and positive, respectively), whereas the long term mean ERA-I map has sample size n = 30 and may be thus somewhat more robust.*

Please also consider the comments by reviewer 1 on lines 320 and 378 and your responses in this context and make sure to explain your interpretation and reasoning to the readers.

*We tried to include the explanations to the referee comments where possible. Please have a look on the manuscript paragraph regarding line 320 below as an example:*

l. 276-284: "Long-term neutral states of both data sources show robust signals in the entire NAR region (i.e., no stippling). This suggests that the different patterns in GCM and REF data are not singularly artefacts arising from different sample sizes. The GCM multi-member composites of positive and negative phases show less pronounced SLP anomalies than the REF data. This difference between GCM and REF may be due to the fact that REF composites were derived from n = 3 negative and n = 4 positive years whereas the GCM data provided n = 264 negative and n=263 positive years during 1981–2010. Regions with strong sampling uncertainties, i.e., where the standard error is larger than the anomaly, are indicated with stippling in panels (a)–(f). These regions are mostly found in the transition region between the wider ERA-I AH and IL nodes, whereas the SLP anomalies at the NAO centers of action show less uncertainty. The GCM patterns are more robustly assessed (i.e., less prone to sampling uncertainty) as can be seen by the very small area with stippling in which the sign of the anomaly may not be assessed robustly."

Please make sure that definitions related to internal variability are explained and used consistently to avoid ambiguity. Currently, some phrases could be unclear (as pointed out by both reviewers). Some example are:
line 2 'natural variability'
lines 191/192: 'internal model noise', 'spread of internal variability'
…

*This is true; the use of definitions related to internal variability was often quite ambiguous. In order to clarify, we now use "internal variability" throughout the text and avoid the use of ambiguous wording like "internal model noise" etc. where possible. Additionally, we dropped the name "std.dev50" and stick to "inter-member spread" (IMS) when referring to the ensemble spread (maximum to minimum) and use "ensemble sd" specifically when referring to the maps of ensemble standard deviations.*

[revised manuscript text omitted]

---

## Author Response (AR2)

**Responses to Reviewer Comments (Major Revision, 2^nd round)**

Dear reviewers, dear editor,

We would like to thank you for a second round of constructive advice and suggestions. During the revision process we tried to incorporate these where possible. Please find our point-by-point responses to the issues raised in the Referee Comments below.

The structure of this document is as follows:

1) Responses to Referee Comment No. 1

2) Responses to Referee Comment No. 2

3) Revised manuscript with track changes (with respect to previous version)

We answer the general remarks stated in the introducing paragraphs of each report before the point-by-point responses.

Changes in the revised manuscript that do not apply to specific comments include:

- Fig. 4: We removed the boxes indicating the AH/IL and CEUR regions to improve readability. The information on their locations is provided in Fig. 2.

- Figs. 6—8: Since the grey lines, representing the correlations between nSAT/PR and the NAO index, are difficult to read in most regions, we changed their colors to red shadings. This also helps to separate the lines from the stippling.

- Figs. A2—A4: The subpanels in Fig. A2 were re-arranged and the subpanel headers of Figs. A3 and A4 were corrected (variable names).

- Table 1: since psl of both RCM data sets was used for the calculation of RMSD, this variable is now included in this table (for some reason, the track changes did not capture it. It is included in the revised manuscript).

In the following, comments raised by the referees are marked in blue, responses in black (explanations in italics). Line numbers refer to the revised manuscript.

With kind regards,

Andrea Böhnisch on behalf of all co-authors

11 May 2020

**Reviewer No. 1**

This is my second review of the manuscript "Using a nested single-model large ensemble to assess the internal variability of the North Atlantic Oscillation and its climatic implications for Europe" by A. Böhnisch et al. I appreciate the amount of work the authors put into improving the manuscript. The authors adequately addressed most reviewer comments (the exception will be discussed below). In my opinion, they have indeed produced a much improved version of the manuscript: it is for the most part very clear, easy to follow, and continues to present an interesting piece of research. I enjoyed this review as I learned a lot. I would like to thank the authors for that.

I suggest publishing the paper in Earth System Dynamics after some minor revisions that I outline below have been addressed.

While the text and reasoning are much easier to follow in the current version of the manuscript than they were in the previous version, some (easily amendable) issues remain. Firstly, the authors still do not consistently use present or past tense (my previous review, comment on l. 140). While I can see they made an effort for a consistent use of past tense, this is not applied everywhere (e.g. l. 105 "Model data is compared..."; l. 176 "The NAO is quantified..."). In fact, most of the results section is written in present tense. I suggest using one or the other, and doing so consistently. My personal preference is usage of present tense for the study at hand and past tense for all cited literature. This will strongly improve readability of the paper. Second, the authors use different abbreviations for "standard deviation": sometimes it is "std" (e.g. l. 129), other times "sd" is used (e.g. l. 354). I suggest choosing one ("std"?) and sticking to it.

Use of tenses:

When re-reading the manuscript, we decided to switch to present tense in section 2 (data & methods) whenever we refer to our present study, while past tense now marks results from cited literature. In doing so, we intend to keep coherence with the results section. This follows the reviewer's suggestion and will hopefully avoid any misunderstanding regarding the presentation of our results and findings.

Abbreviations for standard deviations:

Originally, both abbreviations referred to different aspects: "std" as in "nSAT std" is a standard deviation along the time axis of temperature. "sd" in turn is the across-member standard deviation. However, for the sake of clearness we may refrain from one of the abbreviations and stick to the other one (i.e., sd). We therefore changed nSAT std to nSAT sd.

**Detailed comments**

ll. 13-14 How do the results stress the importance of SMILEs for estimating internal variability?

*We included an explaining phrase in this sentence and changed the word "importance" to "value":*

ll. 13-14: „The results stress the value of single-model ensembles for the evaluation of internal variability by pointing out the large differences of NAO—response relationships among individual members."

l. 40 I think replacing "though" with ", however," would sound nicer. This is a issue several times in the document. Please consider making these changes.

*Thank you. We adopted the suggestion within the revision.*

l. 64 Readers that are not familiar with regional modeling would benefit from a short explanation of the term "nesting" here, I think.

*We included a short parenthesis:*

ll. 73-74: "Combining the driving GCM and nested RCM (i.e., driven by lateral boundary conditions of the GCM) large ensembles (LE) allows for analyzing the spread of NAO states and responses within one model chain."

l. 67 The word "also" suggests that there were other questions asked before. As far as I can tell, there were no explicit questions up until here. I suggest omitting "also" or asking previous questions more explicitly. In light of the reference to "research questions" in line 71, I highly suggest making these research questions absolutely explicit somewhere earlier in the text.

*Thank you. We now explicitly mention the research question:*

ll. 77-79: "The present study targets the research question, how global circulation variability, in this case the NAO teleconnection, affects local climate characteristics when downscaled using an RCM. It specifically aims at evaluating whether the range of internal variability is represented consistently between the driving GCM-LE and the driven high resolution RCM-LE."

ll. 156-166 I suggest presenting the examined 30-year long time horizons (ll. 160-166) first and then discussing their implications (ll.156-159). The current order raises several questions about the time horizons first, that are then answered in the next paragraph. By switching the order of the paragraphs, these questions could be avoided.

**and**

ll. 167-174 Here I have the same comment as for ll. 156-166. Please consider switching the order of the paragraphs. Otherwise the reader might wonder, e.g., how winter is defined. This is not necessary and solely an artifact of the order in which the text presents itself.

*Agreed. We rearranged in both cases the paragraphs.*

Equation 2 Why does the RMS have an asterisk? In the text, the "*" is confusing: I looked for a footnote several times.

*The asterisk was included to indicate that this RMS is a modified RMS. But we agree that, since the metric is defined in a formula, it is not necessary and may be confusing. The asterisk was thus removed in the revised manuscript. We use the term "RMSD" now.*

ll. 228-237 Recently, several references appeared in the literature using large ensemble to examine transient internal climate variability. (At least) these should be cited here: Kay et al. (2015), Sigmon & Fyfe (2016), Maher et al. (2018), Maher et al. (2019).

*Thank you for these suggestions! We conducted a literature review and included references that seemed fitting (e.g. in the introduction, partly also in section 2.2.5 Addressing internal variability).*

l. 249 I cannot see the "slightly low positive NAO value" surplus in figure 3a that the authors describe here. This statement does not seem essential anyway, so I suggest omitting it.

*Agreed, we may omit it. This statement refers to the slight shift of curves towards positive values, i.e. crossing the horizontal 0.5-line at an index value larger than zero. The revised text says:*

ll. 267-269: „The CanESM2-LE produces NAO index values which follow a distribution comparable to the ERA-I data (similar to a normal distribution with $\mu = 0$, $\sigma = 1$, Fig. 3 (a)), but the CanESM2-LE

distribution appears smoother due to a larger sample size (n = 1500 for CanESM2-LE and n = 30 for ERA-I)."

All figures As it stands, stippling is really hard to see in the maps. Please redo the maps with more prominent stippling. I had trouble finding the stipples in the current version although the text told me they were there!

*Thank you for this remark. We included more prominent stippling and clearly indicated in the figure captions the relevant subpanels. In order to better separate it from the isolines, we also changed their colors. See e.g. Fig. 6:*

[Figure]

*Figure 1:* Spatial patterns of change in nSAT mean ($\alpha_1$ in [K]) for a unit change in the NAO index for ERA-I, CRCM5/ERA-I, CanESM2-LE and CRCM5-LE in 1981–2010 ((a)–(f)) and the change in 2070–2099 with respect to 1981–2010 ((g)–(j)). Both 50-member ensembles are represented with ensemble mean ((c)–(d), (g)–(h)) and standard deviation (sd, (e)–(f), (i)–(j)) representing the inter-member spread. Reddish lines in the ensemble mean maps represent the Pearson correlation between nSAT mean and the NAO index at an increment of 0.25; red shadings see legend in upper right panel. Grey stippling in the ensemble mean maps show regions where SNR < 1, SNR being the signal to-noise ratio between the 30 year ensemble mean and sd of GCM and RCM LEs in both time periods. Stippling will be explained in moredetail in Section 3.2.2.

ll. 277, 282 The explanation of what stippling means comes too late here. Please first explain what stippling means, then reference it.

*Thank you. We mention the stippling earlier in the text when the figure is presented.*

l. 290 Please omit "is very promising as it". This statement is not necessary.

*Agreed. We removed it.*

Figure 5 The panel headings for (b) and (c) say "GCM", but I suspect what is shown is from the regional model. Please correct this.

*This figure shows the RMSD of the large-scale SLP pattern between the regional model and its driving data: (a) between CRCM5 and ERA-I as driving data during 1981-2010 winters, (b) between CRCM5 and CanESM2 as driving data during 1981-2010 winters, and (c) like (b), but for 2070-2099 winters. As the panel headings indeed are misleading, we changed them to explicitly mention the driving data/CRCM5 combination.*

[Figure]

*Figure 2:* RMSD of monthly SLP differences between driving data and CRCM5 members, calculated following Eq. (2). Colouring: RMSD <= 1 significant at p <= 0.05 with a false detection rate smaller than 0.1 (see Wilks, 2016). (a) for driving data ERA-I (1981–2010, one realization), (b) for driving data CanESM2-LE (1981–2010, 50 members), (c) for driving data CanESM2-LE (2070–2099, 50 members). Red box: position of CEUR domain.

ll. 368ff Please use one p-value throughout the manuscript and stick to it! Using changing p-values introduces significant ambiguity into the text. For example, in line 369, what does "sometimes significant" mean? Which significance level does that refer to? Switching p-values for statistical significance gives the impression of p-hacking, and I am sure that the authors want to avoid this impression (and these results are significant and interesting enough as they are).

*p-values: Thanks for your remark. In this paragraph, we wanted to show whether the variances between historical and future periods are significantly different within the future and historical ensembles. In the conclusion, we argue that they are **not** since we cannot reject our null hypothesis of no differences – apart from one case with $p <= 0.05$ ($p = 0.0443$) and another one with $p <= 0.1$ ($p = 0.0546$). We agree that the deviating p-value is rather confusing and should be abandoned. So the text now only mentions the one counter-example which is significant at the level chosen for the text (i.e., $p <= 0.05$).*

*"sometimes": This is misleading wording. In this paragraph, it is as in the remainder of the study $p <= 0.05$. We rephrased to show what we actually wanted to say:*

ll. 403-407: "This general finding does not change in the projected future climate: most boxes and whiskers keep their size, only GCM nSAT in the NE region is characterized by a larger range in the future (significant at $p ≤ 0.05$, using an F-test for comparison of variances). Some of the ensemble mean values exhibit a significant shift towards lower r values in the future for both models for nSAT mean and PR sum (see text insertions CanESM2(hist, fut) and CRCM5(hist, fut))."

l. 374 "it is now advised" should be omitted. Who gave that advise?

*Agreed, we rephrased this sentence.*

l. 411: "Having analyzed GCM and RCM separately so far, the next step is to compare both ensembles".

ll. 399-340 Replace "needs to be taken into account" with "needs to be expected"?

*We rephrased, but we suggest: "needs to be considered" (l.437).*

ll. 491-502 Isn't this already part 4.4?

*Yes, it can be argued that this paragraph is actually part 4.4. Originally, the focus of section 4.3 was set on internal variability which is why we first put this paragraph in there. However, in order to provide a more complete image of the future conditions and changes, we include this paragraph in section 4.4 in the revised manuscript.*

l. 505 This is one of the examples of surprising switch to past tense, when present tense was used before.

*Adjusted. We decided to change to present tense for analyses performed in this study rather than past tense and keep it whenever results of our study are mentioned.*

l. 554 What is the "climate module"? I do not think it was mentioned in the manuscript so far.

*Thank you! This is a leftover of a previous revision. The climate module originally refers to one of the modules within the ClimEx project. Since we do not mention it in the manuscript any more, the term was changed to: "of this GCM-RCM combination" (l. 590).*

**Reference:**

Wilks, D. S.: "The Stippling Shows Statistically Significant Grid Points": How Research Results are Routinely Overstated and Overinterpreted, and What to Do about It, Bulletin of the American Meteorological Society, 97, 2263–2273, https://doi.org/10.1175/BAMS-D-15-00267.1, 2016.

**Reviewer No. 2**

In this study the regional climate model, CRCM5, is employed to dynamically downscale a large ensemble of climate change simulations to investigate the nature of downscaled responses to the modeled North Atlantic Oscillation (NAO) and its influence on future European climate. By employing a large ensemble, the authors are able to evaluate future downscaled responses associated NAO inter-annual variability in addition to mean changes. The authors set out four key questions related to, documenting the properties and fidelity of the modeled NAO in both the GCM and RCM; the associated screen temperature and precipitation responses in both models; and how such properties change under future external forcings.

In my initial review of this manuscript, I asked the authors to provide a more detailed investigation of the RCM's ability to faithfully reproduce the NAO signal in the driving data and I provided a detailed suggestion on how that might be done. The authors have done an excellent job of addressing this issue in the revised manuscript. However, in reading through the new manuscript I continue to find it difficult to read due to back-and-forth referencing (including far too many acronyms), which I mentioned in my minor comments of the first draft.

In particular, the authors have front-loaded Section 2 with discussion and derivation of diagnostics which are then used in later sections of the paper. However, the discussion in those later sections is conducted as if that earlier material was just presented. It was not, and the reader must constantly stop and hunt down the meaning of acronyms and variables. This makes it very difficult to read the paper. It could easily be resolved by bringing the motivation, derivation, and application of diagnostics and ideas together. I have identified a few places but it is problematic in more than these examples. I don't want to hold the authors to another major revision as I did not raise this so strongly in my first review. As a consequence, I offer my comments as suggestions to the authors and leave it to them to follow through. I would highly recommend that the authors attempt to improve the flow of this paper, however, as I feel it is a very good piece of work and I wouldn't want to see that obfuscated by the manner in which it was presented. With this understanding, I therefore recommend publishing this work with minor modification. My detailed comments follow.

**Abbreviations/acronyms:**

We went through the abbreviations in the manuscript to decide which may be abandoned. This was the case for "IMS" (inter-member spread), "LBC" (lateral boundary conditions) and "REF" (reference). On other occasions, we inserted a reminder if the abbreviation/acronym reappears several paragraphs/pages later again (this applies e.g. for the $\alpha_1$ parameter, i.e. the change of temperature/precipitation associated with unit index change of the NAO). Where possible, we also tried to rephrase/rearrange phrases with excessive abbreviation use to avoid chains of acronyms.

**Structure:**

We understand the concerns regarding the difficulties when reading the paper due to the separation of methods and results/discussion. Nevertheless, we decided to stick to that structure and include, where applicable, reminders of the concepts in the results and discussion sections. This allows us to use the methods section for answering the question "how is this done?" for each of the key questions.  However, we did remove the detailed discussion of concepts in the methods section to reduce the mentioned front-loading and included it in the results/discussion sections.

**Minor Comments:**

1) p.1 l.11 "Reproductions" -> "Reproduction"

*We rephrased this sentence, such that it now reads:*

ll. 10-12: "NAO flow pattern reproductions in the CanESM2-LE trigger responses in the high-resolution CRCM5-LE that are comparable with reanalysis data."

2) p.2 l.23 "Atmospheric modes" -> "Such large-scale atmospheric modes"

*Thank you, we rephrased accordingly (l.24-25).*

3) p.2 ll.54-56 "One way to trigger internal variability in GCM simulations is to perturb the initial conditions of the model, leading to several realizations of weather sequences under identical external forcing which also allow to derive a robust distribution of NAO index values." perhaps reword to, "One way to sample realizations is to perform an initial-value ensemble in which multiple simulations are performed with identical external forcings but perturbed initial conditions. Such an initial-value ensemble would allow a more robust distribution of NAO index values to be sampled."

*In this case, we opt for a combination of the suggestion and the original sentence:*

ll. 55-62: "One way to sample realizations is to perturb the initial conditions of the model, leading to multiple simulations with identical external forcing which only differ due to internal variability. […] Such initial condition ensembles also allow a more robust distribution of atmospheric modes to be sampled, as was done e.g. for El Niño/Southern Oscillation in Maher et al. (2018)."

4) p3. l.78 "until 2099" -> "that extend until 2099" ***and*** 5) p.4 l.94 "regarding" -> "of"

*Thanks, we adopted both wording suggestions (l. 88, l. 104).*

6) p.5 l.97 At this point I didn't recall what IMS represented. Even though it was defined on the previous page under key question c, I had to stop and go back to find it. It is only used once in this paragraph and not used again for 5 pages. The economy of saving three words is not worth the break in flow here. This is a problem that exists all over the manuscript and it does significant harm to the readability of the paper. I would encourage the authors to greatly reduce the use of such acronyms - particularly when used so intermittently.

*We removed the acronym IMS (as well as REF) which indeed seem superfluous. Several others, such as GCM, RCM, NAO or the models' and variables names were kept, as we think that they are intuitive to the reader and make the text more readable. When it comes to the region names, we added the word "region" to the acronyms to break long "acronym chains".*

7) p.5 ll.100-104 Care should be taken here. Just because a globally integrated quantity from an initial-value ensemble of one model spans a similar range as a multi-model ensemble does not mean they are interchangeable. Different models have different physics packages and so could have very different regional behavior and that fact is potentially masked by a globally integrated diagnostic.

*Thank you for this remark. We rephrased the paragraph regarding this aspect. We did not mean to indicate that both spreads were interchangeable. They are not due to different sources of uncertainties in both ensembles (internal variability vs. model response uncertainty + internal variability). We included a short discussion on the subject of spatially integrated versus regional/local signals:*

**8/14**

ll. 108-117: "Comparing the internal variability of the CRCM5 members with the inter-member spread of a subset of the multi-model EURO-CORDEX (Coordinated Regional climate Downscaling Experiment) ensemble regarding regionally integrated European winter temperature and precipitation, von Trentini et al. (2019) showed that both ensemble spreads are of comparable magnitude. The CORDEX ensemble consists of several GCM-RCM combinations set up in a coordinated modelling framework and aims at evaluating uncertainty due to model configuration (Giorgi et al., 2009). The comparison of the single-model and multi-model spreads suggests that a large fraction of the CORDEX ensemble spread regarding temperature and precipitation can be explained by internal variability, despite the fact that it was not explicitly sampled within the CORDEX framework (where most models provided a single simulation, von Trentini et al., 2019). At smaller regional scales, however, single-model and multi-model spreads may show considerable and in parts temporally changing differences which may partly be induced by model response uncertainties (von Trentini et al., 2019)."

8) p.5 l.118 "Figure 1" -> "In the Appendix, Figure 1"

*We included the suggestion: "In the Appendix, Fig. A1, …" (l. 131).*

9) p.5 ll.120-124 Perhaps reword "The GCM... PR sum values" to "Displaying opposite bias to CRCM5, the GCM overestimates (underestimates) mean winter nSAT in the norther (southern) part of the domain, whereas winter PR sum is underestimated in the eastern half of the domain and overestimated on the western side of the Alps. As this study will focus on responses in nSAT and PR induced by the NAO (see Section 2.2.4), aside from regions with particularly high PR sum values, it is found that such NAO responses are generally insensitive to these biases."

*Thanks for this suggestion. We included it with additional figure references:*

ll. 133-138: "Displaying opposite bias to CRCM5, the CanESM2 overestimates (underestimates) mean winter nSAT in the northern (southern) part of the domain (Fig. A1 (b)), whereas winter PR sum is underestimated in the eastern half of the domain and overestimated on the western side of the Alps (Fig. A1 (e)). As this study will focus on responses of nSAT and PR induced by the NAO (see Section 2.2.4), aside from regions with particularly high PR sum values, it is found that such NAO responses are generally insensitive to these biases."

10) p.5 ll.127-128 "...dispersion. So the following analyses were not confined to..." -> "...variability. So, in addition to analyses of"

*We adopted this suggestion (l.141).*

11) p.8 ll.199-200 "As a next step, the monthly difference between driving data and the RCM data was taken for each time step and member" -> "As a next step, time series of the difference between monthly mean driving data and the RCM data was taken for each member"

*When revising the methods section, we changed this paragraph. In accordance with a comment raised by the second reviewer, we also dropped the asterisk: RMS\* → RMSD. The paragraph now reads:*

ll.211-213: "As a next step, a root-mean-square difference (RMSD) of the difference time series between monthly mean driving and RCM data over the hist and fut time periods is obtained across all members and winter months:"

12) p.8 l.205 "between driving" -> "between monthly mean driving"

*Thanks, we adopted this clarifying suggestion (l. 216).*

13) p.8 ll.208-209 "allows to derive a measure relative to the inter-annual variability of the SLP pattern on a given location. Low RMS* values indicate a low error." -> "provides a measure relative to the inter-annual variability of the SLP pattern in a given location. Low RMS* values in a particular region indicate a low error and so a good reproduction of the SLP variations in that region of the RCM."

*Thank you, we accepted this suggestion apart from the last sentence. Instead, we included an according explanation in the results section (by following your suggestion in point 21).*

ll. 219-220: "The normalization by the square root of the temporal variance of the driving data provides a measure relative to the inter-annual variability of the SLP pattern in a given location."

14) p.9 l.228 "IMS" see my point 6)

*We aimed at solving this problem, e.g. by reminding the reader what a given abbreviation means. "IMS" is, as indicated above, removed now.*

15) p.9 ll.227-234 This is poorly worded. The authors are mixing ideas of stationary and non-stationary systems to discuss issues related internal variability. For stationary dynamical systems, one can define/identify internal variability by looking a the statistical properties of a time series relative to its long-term time average. For a non-stationary system, one can define/identify internal variability by looking a the statistical properties of an initial-value ensemble of time series relative to its ensemble mean at any specific time. The degrees of freedom that go into each evaluation depends on the length of the time series in the former and the number of ensemble members in the latter.

*Thank you for your explanation. We revised this paragraph, including some additional references suggested by the second reviewer.*

16) p.10 l.248, l.251 The terms ERA-I data and REF realization are the same I believe. This is very confusing as the equivalence between the two was drawn 12 pages earlier! Why REF? Why not OBS. This would be much more obviously connected to ERA-I. This again falls back to my point 6). Also l.254 I couldn't remember what IMS was again here.

*The abbreviations REF, GCM and RCM refer to the models; ERA-I, CanESM2 and CRCM5 are the models'/model data names. We refrained from the abbreviation REF and changed to ERA-I or reference. We did not use OBS since using "reanalyses" and "observations" synonymously may confuse readers.*

17) p.10 l.260 l.261 My point 6) again. I had to hunt a long time to find where SOIC, MOIC and "n" were defined. The first two weren't even in the text - they were in the caption to Fig. 1! Again, this breaks the flow of the paper and makes it very difficult to read and understand. Are these acronyms really necessary? For "n" perhaps remind the reader what it is when using it here - even a text search of a PDF isn't much help with a single letter!

*SOIC and MOIC are explained in the text in line 260 (first revised manuscript version): "(i) correlations among the 10 members from the same ocean family (same ocean initial conditions in 1950, SOIC " […] "(ii) correlations between each member and the 40 members from the 4 other ocean family (mixed ocean initial conditions, MOIC".*

*To clarify, we rearranged the order of the words, such that this phrase now reads:*

ll. 284-286: "(i) correlations among the 10 members from the same group (same ocean initial conditions – SOIC, n = 225 cases, dotted lines in Fig. 3(b)) and (ii) correlations between each member

and the 40 members from the 4 other groups (mixed ocean initial conditions – MOIC, n = 1000 cases, solid lines in Fig. 3 (b))"

*n as symbol for sample size was introduced in line 249/250 of the first revised manuscript version: "… due to a larger sample size (n = 1500 for CanESM2-LE and n = 30 for ERA-I)." We followed your suggestion, including a reminder.*

18) p.10 ll.254-260 "For further ... in Leduc et al. (2019):" In addition to comments 16 and 17 above, I found this description very hard to follow. In its place I offer the following. "The independence of the 50-member ensemble is critical to interpreting the inter-member spread as a proxy for internal variability. In evaluating this, it is important to recall that the 50-member LE was constructed in two parts. First, independent atmosphere/ocean states in 1850 were used to launch 5 historical simulations and integrated forward until 1950. Second, in 1950, each of these 5 ensemble members were used to launch 10 individual simulations by applying a small perturbation to the atmosphere and integrated forward until 2099 thereby producing the 50-member large ensemble. As a consequence, for this study, members between each of the 5 groups of 10 are expected to be highly independent while members within each group of 10 are perfectly correlated in 1950 and progressively increase their independence beyond their 1950 starting point. To evaluate whether the 10 members within each of the 5 groups had become sufficiently independent by the two 30-year periods of interest (1981-200 and 2070-2099), correlations were applied to two groups following Leduc et al. (2019):"

*Thank you for this detailed suggestion. We adopted this paragraph with some minor adjustments:*

ll. 274-284: "For the following analyses independence of the 50-member ensemble is critical to interpreting the inter-member spread as a proxy for internal variability. In evaluating this, it is important to recall that the 50-member CanESM2-LE was constructed in two steps (Fyfe et al., 2017; Leduc et al., 2019): First, independent atmosphere/ocean states in 1850 were used to launch 5 historical simulations integrated forward until 1950. Second, in 1950, each of these 5 ensemble members were used to launch 10 individual simulations by applying a small perturbation to the atmosphere and integrated forward until 2099, thereby producing the 50-member large ensemble.

As a consequence, for this study, members between each of the 5 groups of 10 are expected to be independent. However, members within each group of 10 are highly correlated in 1950 and progressively increase their independence beyond their 1950 starting point. To evaluate whether the 10 members within each of the 5 groups have become sufficiently independent by the two 30-year periods of interest (1981-2010 and 2070-2099), correlations among member time series are applied to two groups following Leduc et al. (2019):"

19) Fig.3 Perhaps plot the normal distribution in this figure for reference.

*Thank you, we followed this suggestion (see green line in Fig. 3):*

[Figure]

[Figure]

*Figure 3:* Cumulative density functions (CDFs) of NAO index values. (a) distribution of all CanESM2-LE (n = 50 x 30 per period) and ERA-I (n = 30) NAO index values. (b) pairwise correlations among member NAO index time series from the same ocean families (SOIC – same ocean initial conditions, dotted lines, n = 225), from different ocean families (MOIC – mixed ocean initial conditions, solid lines, n = 1000) and between ERA-I and all CanESM2 members (n = 50). Black: 1981–2010 CanESM2-LE, red: 2070–2099 CanESM2-LE, blue: 1981–2010 ERA-I, green: normal distribution with μ = 0 and σ = 1 in (a) and σ = 0.2 in (b).

20) p.12 l. 297 "RMS*" -> "RMS* (eq. 2)" it's been many pages since it was discussed. Which brings up an important point. Why not motivate the need for this diagnostic, then discuss the derivation of RMS* at the place where the diagnostic is used - here. This goes for most of what is presented in Section 2 and then is used in later sections of the manuscript. It would make the paper far more readable. The constant back-and-forth looking for things being discussed is quite disruptive. The authors are making it very challenging for themselves by this choice of presentation and they are not doing a good job of meeting this challenge.

*Thanks for your concerns. As mentioned in the beginning of this document, we decided to keep the separation of methods and results. In this case, we included an introducing sentence which may also be seen as a reminder of concept, before explaining the RMSD:*

l. 324: "This is achieved by analysing the deviations of RCM and driving data SLP variability.“

21) p.12 ll.298-300. "A value of RMS∗ ≥ 1 indicates that the root-mean squared error between the RCM and driving data is larger than the temporal variability in the driving data. In this case, the large-scale SLP pattern may not be seen as being correctly represented in the RCM data." To make this point clearer, perhaps reword to, "An O(1) value of RMS* would indicate a poor reproduction of the SLP signal in the RCM because the RMS difference between the RCM and driving data SLP is of the same order as the variability of the SLP in the driving data itself. Values of RMS*<<1, on the other hand, would indicate a good reproduction of the SLP signal in the RCM because it suggests that the RCM is tracking the variability in the driving data."

*Thanks, we adopted this suggestion with some minor changes:*

ll. 326-329: "An O(1) value of RMSD would indicate a poor reproduction of the SLP signal in the RCM because the RMSD between the RCM and driving data SLP is of the same order as the variability of the SLP in the driving data itself. Values of RMSD << 1, on the other hand, would indicate a good reproduction of the SLP signal in the RCM because it suggests that the RCM is tracking the variability in the driving data.”

22) p.12 ll.300-302 With 21) above you could now change, "The large-scale SLP pattern over the entire ClimEx domain, which also includes the CEUR, NE, BY and SE domains, is reasonably well represented: with RMS∗ < 1 in most parts of..." to "With this understanding it can be seen that the

large-scale SLP pattern over the entire ClimEx domain, which also includes the CEUR, NE, BY and SE domains, is reasonably well represented in most parts of..."

*Thanks, we also adopted this suggestion, also with some minor modifications:*

ll. 329-331: "With this understanding it can be seen that the large-scale SLP pattern is reasonably well represented in most parts of the entire ClimEx domain for both driving data sets and both periods…"

23) p.13 ll.311-313 "nSAT", "PR", "\alpha_1" see my point 6) and 20). I've run out of steam highlighting these and I leave it to the authors to identify and correct the many that remain beyond this point in the paper

*As stated before, we tried to reduce our use of acronyms or abbreviations. However, in this case, since nSAT and PR are the names of our variables, we did not remove them. But we did include a reminder on what "\alpha_1" means.*

ll. 340-342: "nSAT and PR spatial responses as revealed in the ERA-I data are generally reproduced under current climate conditions in the CanESM2-LE and CRCM5-LE (see Figs. 6–8). Highest magnitudes of the NAO-responses (i.e., the slope of the regression line, $\alpha_1$, introduced in Eq. (4)) occur in the CRCM5/ERA-I run for all variables."

24) p.13 l.311 "... the ERA-I data are reproduced in their general properties under current ..." -> "... the ERA-I data are generally reproduced under current ..."

*Thank you for this suggestion, we adopted it (l. 340).*

25) Figs. 6-7 One quantity is displayed in (a)-(f) and a different quantity is displayed in (g)-(j). The titles on the plots in (c)-(f) are identical to those in (g)-(j). This is confusing. Also, it would be helpful if the mean (columns 1 and 3 in rows 2 and 3) and sd (columns 2 and 4 in rows 2 and 3) plots were placed beside each other to facilitate comparison of identical quantities between the GCM and RCM.

*We included an axis label to clarify that the first two rows show total values during 1981-2010, whereas the third row shows the 2070-2099 anomalies. Additionally, the panels were rearranged such that the first and second columns show the ensemble mean values for GCM and RCM while the third and fourth column present the ensemble sd.*

**References:**

[revised manuscript text omitted]